# Uniformity First: Uniformity-aware Test-time Adaptation of Vision-language Models against Image Corruption

## Abstract

Pre-trained vision-language models, such as contrastive language-image pre-training (CLIP), have demonstrated a remarkable generalizability, enabling a wide range of applications, including zero-shot classification. However, vision-language models still struggle to handle *distribution shifts*, where input samples have large gaps from training ones. We found that CLIP is especially vulnerable to image corruption, a type of realistic distribution shift caused by sensor conditions such as weather, light, or noise. Collecting a new dataset from a test distribution for fine-tuning is highly costly since image corruption occurs unexpectedly and has a wide variety of types. Thus, we investigate *test-time adaptation (TTA)* of zero-shot classification, which enables on-the-fly adaptation to the test distribution with unlabeled test data. Existing TTA methods for CLIP mainly focus on modifying image and text embeddings or predictions to address distribution shifts. Although these methods can adapt to domain shifts, such as out-of-distribution or different renditions in input images, they fail to adapt to distribution shifts beyond domain shifts, e.g., image corruption. We found that *uniformity* of image embeddings, which is related to the amount of information, is a key factor that differentiates domain shifts and other distribution shifts. To enable adaptation to image corruption, we propose a novel method called *un*iformity-aware *info*rmation-balanced TTA (UnInfo). To address distribution shifts, we introduce uniformity-aware confidence maximization, information-aware loss balancing, and knowledge distillation from the exponential moving average (EMA) teacher. Through experiments, we demonstrate that our UnInfo improves accuracy under image corruption by retaining information in terms of uniformity.

## 1 Introduction

Vision-language models (VLMs) pre-trained on large-scale datasets, such as contrastive language-image pre-training (CLIP) (Radford et al., 2021), have demonstrated remarkable generalizability and rich feature representations. Specifically, pre-trained VLMs have enabled various applications such as zero-shot transfer (Radford et al., 2021; Ge et al., 2023; Wang et al., 2023b), image/video retrieval (Baldrati et al., 2022; Fang et al., 2021), and image generation (Patashnik et al., 2021; Ramesh et al., 2022). VLMs owe their success to their rich feature representations that unify vision and language modalities and public availability of the pre-trained weights, such as OpenCLIP (Ilharco et al., 2021; Cherti et al., 2023; Schuhmann et al., 2022). However, despite their generalizability, VLMs still face a challenge in adapting to distribution shifts, i.e., making predictions on test datasets with large gaps from the training dataset (Zhang et al., 2022b; Huang et al., 2024; Chen et al., 2023; Shu et al., 2022; Zhou et al., 2024; Karmanov et al., 2024; Zhang et al., 2024b; Zanella & Ben Ayed, 2024b; Wang et al., 2024b; Qian & Hu, 2024).

A naive way of adapting VLMs is to collect a dataset from the test distribution and fine-tune the entire model parameters or the head classifier. However, labeled data from the test distribution may not be available because the distribution shift is unknown before deployment (Wang et al., 2021; Adachi et al., 2023).

To adapt to distribution shifts instantly after being deployed in the test distribution, *test-time adaptation (TTA)* (Liang et al., 2024; Dong et al., 2025), a paradigm aiming to adapt models during testing using

only unlabeled test data, has attracted attention. In the context of VLMs, recent studies (Shu et al., 2022; Zhou et al., 2024; Karmanov et al., 2024; Zhang et al., 2024b; Wang et al., 2024b; Qian & Hu, 2024; Zanella & Ben Ayed, 2024b) have intensively studied the TTA for zero-shot classification, which is one of the most common applications of VLMs. When the domain changes, typical text prompts can be suboptimal. For example, in an art or illustration domain, text prompts such as "`a photo of a [class name]`" is not appropriate (Shu et al., 2022; Zhou et al., 2022b;a). In other words, there is a modality gap between text prompts and images in the embedding space, which is crucial for VLMs' generalization (Liang et al., 2022; Khattak et al., 2023; Qian et al., 2024; Yamaguchi et al., 2025; Eslami & de Melo, 2025), under domain shifts. Thus, existing TTA methods for VLMs are mainly designed for domain shifts (also called natural distribution shifts), such as changes in rendition or out-of-distribution (OOD) (Hendrycks et al., 2021a; Recht et al., 2019; Hendrycks et al., 2021b; Wang et al., 2019) by modifying image and/or text embeddings during testing, which can be viewed as addressing the modality gap.

While existing TTA methods successfully adapt to the domain shifts, we found that they are prone to overfitting to domain shifts and degrade the performance on other types of distribution shifts, e.g., *image corruption* (Hendrycks & Dietterich, 2019; Sójka et al., 2023). When an image recognition system is deployed in the real world, the model faces various perturbations even in the same domain. This is because of changes in weather, light conditions, noise, cameras, etc., which are crucial in a wide range of applications, such as autonomous driving or surveillance cameras (Dai & Gool, 2018; Volk et al., 2019; Eastwood et al., 2022; Adachi et al., 2023; 2024; Enomoto et al., 2024). Such perturbations occur unexpectedly even within a single domain. In the existing literature on ordinary classification models, image corruption deteriorates the model's accuracy (Hendrycks & Dietterich, 2019; Qin et al., 2022).

We examined the robustness of VLMs against various types of image corruption (Hendrycks & Dietterich, 2019; Mintun et al., 2021) in terms of zero-shot classification performance using CLIP-family models. Through the experiment, we found that they significantly degrade the performance on corrupted images and that existing TTA methods can fail to improve performance. Moreover, we analyzed image embeddings and CLIP's knowledge about image corruption to reveal the difference between domain shift and image corruption. As a result, we experimentally found that image corruption also causes the modality gap between image and text embeddings, but the mechanism differs from domain shifts. Under image corruption, the modality gap occurs by image embeddings being "corrupted" in terms of *uniformity*, a measure related to the amount of input information retained in the embedding space (Wang & Isola, 2020; Wang et al., 2023a). In other words, the amount of input information retained in the image embeddings becomes small by image corruption. Nevertheless, most of existing CLIP TTA methods for domain shifts address the modality gap by updating embedding vectors in a post-hoc manner without updating encoders, which implicitly assumes that embedding vectors are discriminative under domain shifts. However, under image corruption, image embeddings retain less information, i.e., they are less discriminative. This is the primary reason why existing CLIP TTA methods fail to maintain their performance on image corruption. Furthermore, we found that CLIP models cannot sufficiently encode words related to image quality; thus, simple prompting techniques, such as ensembles or incorporating corruption names into prompts, cannot recover the performance degradation (Sec. 3). From these observations, existing TTA methods or simple prompting techniques targeting domain shifts suffer from image corruption, highlighting the necessity of a novel TTA method suitable for a wider range of distribution shifts.

To enable the TTA of CLIP under various types of image corruption, we propose a novel method called ***uniformity-aware info**rmation-balanced test-time adaptation (UnInfo)*. UnInfo addresses the fundamental challenge of image embeddings' corruption by updating the image encoder with a low-rank adapter (LoRA) (Hu et al., 2022). To realize effective adaptation, UnInfo consists of three components: (i) uniformity-aware confidence maximization, (ii) information-aware loss balancing, and (iii) knowledge distillation from the exponential moving average (EMA) teacher. Uniformity-aware confidence maximization seeks to maximize prediction confidence in terms of entropy, while incorporating uniformity to prevent embeddings from losing input information. The information-aware loss balancing adaptively controls the balance between confidence maximization and uniformity enhancement on the basis of mutual information so that uniformity is first leveraged and then confidence is addressed. This balancing plays a critical role, specifically when confidence is unreliable because of severe image embedding corruption. The knowledge distillation from the

EMA teacher stabilizes the encoder update by tracking the EMA of LoRA parameters and regularizing the student's prediction to be close to the teacher's.

Through extensive experiments, our UnInfo improved the test zero-shot accuracy on various types of image corruption by incorporating uniformity and balancing priority between uniformity and entropy.

## 2 Zero-shot Classification with CLIP

Given a pre-trained CLIP composed of a text encoder $f_{\theta_{\text{txt}}}^{\text{txt}} : \mathcal{T} \to \mathbb{R}^d$ and image encoder $f_{\theta_{\text{img}}}^{\text{img}} : \mathcal{X} \to \mathbb{R}^d$, we first encode the text prompts to obtain text embeddings, which are used for the prototype of each class in the embedding space $\mathbb{R}^d$, where $\mathcal{T}$ and $\mathcal{X}$ are text and image input spaces, and $\theta_{\text{txt}}$ and $\theta_{\text{img}}$ are pre-trained weights of the encoders. The text prompts typically consist of a template and class names, like "`a photo of a [class name]`," denoted by $\mathbf{p}_c$ for class $c$. We denote the corresponding text embeddings as $\{\mathbf{t}_c = f_{\theta_{\text{txt}}}^{\text{txt}}(\mathbf{p}_c)\}_{c=1}^C$, where $C$ is the total number of classes. We assume the text embeddings are normalized, i.e., $\|\mathbf{t}_c\|_2 = 1$.

For a test image $\mathbf{x} \in \mathcal{X}$, we compute the image embedding $\mathbf{z} = f_{\theta_{\text{img}}}^{\text{img}}(\mathbf{x})$, where $\|\mathbf{z}\|_2 = 1$. The similarity between the image and text embeddings $\mathbf{z}^\top \mathbf{t}_c$ is regarded as the logit for class $c$. The zero-shot prediction probability is obtained by

$$\hat{p}_c = \text{softmax}(\mathbf{z}^\top [\mathbf{t}_1, \ldots, \mathbf{t}_C]/\tau)_c, \tag{1}$$

where $\tau > 0$ is the temperature parameter. The final prediction is made by taking the argmax of $\hat{p}_c$.

## 3 Preliminary Experiment

First, we empirically demonstrate the vulnerability of CLIP under image corruption in terms of zero-shot classification accuracy. Moreover, CLIPs cannot properly encode images under such distribution shifts, i.e., the performance degradation cannot be recovered by simple prompting techniques because CLIPs cannot sufficiently represent concepts related to image quality.

**Setup.** We evaluated a ViT-B/16 CLIP pre-trained on the LAION dataset (Schuhmann et al., 2022) downloaded via OpenCLIP (Ilharco et al., 2021). We used the ImageNet-C (Hendrycks & Dietterich, 2019) dataset, which includes 15 types of image corruption simulating the image corruption (see Sec. 5.1 for dataset details). We performed zero-shot classification with the text prompt "`a photo of a [class name]`" (Normal prompt). We also used the ensemble of 80 text prompts, e.g., "`a bad photo of a [class name]`," "`a photo of many [class name]`," and so on, which is widely adopted as one baseline (Radford et al., 2021) (Ensemble), to see the effectiveness of prompt engineering on image corruption. To check the expressiveness of CLIP representation, we also examined text prompts that included descriptions of corruption. Specifically, we used the prompts "`a photo of a [class name] corrupted by [corruption name]`." For each corruption, we generated ten synonyms of the `corruption name` with GPT-4o (Hurst et al., 2024) and ensembled the text prompts (Corruption prompt). Details of the prompt ensemble are provided in Sec. C in the appendix. We tested on ImageNet (Deng et al., 2009), ImageNet-A/R (Hendrycks et al., 2021b;a) (domain shifts), and ImageNet-C (Hendrycks & Dietterich, 2019) to observe the effect of image corruption.

**Evaluation metrics.** We evaluated accuracy, entropy, uniformity loss, and the modality gap between text and image embeddings by the earth mover's distance (EMD). The entropy measures the uncertainty of predictions (a lower value indicates high confidence), and the uniformity loss measures how image embeddings are uniformly distributed on the unit hypersphere, which is related to the amount of input information preserved in the image embedding (Oord et al., 2018) (a lower value indicates more information is preserved). The modality gap measures distributional distance between image and text embeddings, which is related to the generalizability of CLIP (Liang et al., 2022; Khattak et al., 2023; Qian et al., 2024; Yamaguchi et al.,

Table 1: Zero-shot classification metrics of OpenCLIP ViT-B/16 with simple prompting techniques on ImageNet family and ImageNet-C.

| Metric | ImageNet | ImageNet-A | ImageNet-R | Domain shift Mean* | Defocus blur | Glass blur | Motion blur | Zoom blur | Contrast | Elastic transform | Jpeg compression | Pixelate | Gaussian noise | Impulse noise | Shot noise | Brightness | Fog | Frost | Snow | Corruption Mean |
|---|---|---|---|---|---|---|---|---|---|---|---|---|---|---|---|---|---|---|---|---|
| Accuracy (Normal prompt, ↑) | 66.97 | 32.91 | 74.36 | 58.08 | 28.18 | 11.81 | 19.13 | 17.65 | 17.90 | 13.37 | 36.77 | 36.92 | 6.02 | 6.12 | 7.88 | 54.75 | 34.39 | 27.32 | 27.77 | 23.06 |
| Accuracy (Ensemble, ↑) | 67.59 | 33.15 | 76.51 | 59.08 | 29.09 | 12.56 | 20.56 | 19.12 | 18.55 | 14.34 | 37.71 | 37.89 | 6.36 | 6.54 | 8.27 | 55.78 | 35.99 | 28.37 | 28.34 | 23.97 |
| Accuracy (Corruption prompt, ↑) | - | - | - | - | 26.53 | 12.15 | 18.39 | 17.99 | 17.65 | 13.96 | 36.29 | 36.35 | 6.02 | 6.10 | 7.80 | 53.93 | 33.43 | 27.34 | 26.55 | 22.70 |
| Entropy (↓) | 0.748 | 1.220 | 0.595 | 0.854 | 2.011 | 2.703 | 2.317 | 2.245 | 3.247 | 2.297 | 1.632 | 1.615 | 3.773 | 3.714 | 3.671 | 1.061 | 1.681 | 1.905 | 2.048 | 2.395 |
| Uniformity loss (↓) | 0.513 | 0.538 | 0.500 | 0.517 | 0.682 | 0.735 | 0.722 | 0.715 | 0.744 | 0.706 | 0.630 | 0.641 | 0.855 | 0.853 | 0.839 | 0.601 | 0.665 | 0.655 | 0.686 | 0.715 |
| Modality gap (EMD, ↓) | 1.291 | 1.333 | 1.348 | 1.324 | 1.298 | 1.326 | 1.325 | 1.327 | 1.337 | 1.341 | 1.293 | 1.296 | 1.299 | 1.297 | 1.296 | 1.293 | 1.307 | 1.315 | 1.308 | 1.311 |

*The mean over the vanilla ImageNet, A, and R.

Table 2: Zero-shot corruption type classification accuracy (%).

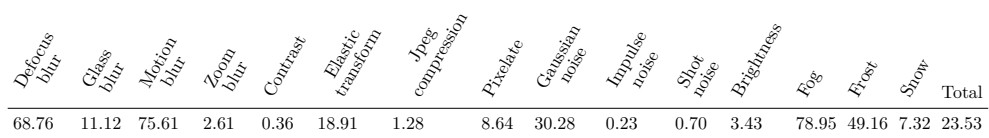

| Defocus blur | Glass blur | Motion blur | Zoom blur | Contrast | Elastic transform | Jpeg compression | Pixelate | Gaussian noise | Impulse noise | Shot noise | Brightness | Fog | Frost | Snow | Total |
|---|---|---|---|---|---|---|---|---|---|---|---|---|---|---|---|
| 68.76 | 11.12 | 75.61 | 2.61 | 0.36 | 18.91 | 1.28 | 8.64 | 30.28 | 0.23 | 0.70 | 3.43 | 78.95 | 49.16 | 7.32 | 23.53 |

2025; Eslami & de Melo, 2025). The entropy and uniformity loss are defined as follows:

$$\text{Entropy} := \sum_{c=1}^{C} -\hat{p}_c \log \hat{p}_c, \tag{2}$$

$$\text{Uniformity loss} := \mathbb{E}_{\mathbf{z}_1, \mathbf{z}_2} \left[ \exp(-\|\mathbf{z}_1 - \mathbf{z}_2\|_2^2) \right], \tag{3}$$

$$\text{Modality gap} := \text{EMD}(\{\mathbf{z}_1, \ldots, \mathbf{z}_N\}, \{\mathbf{t}_1, \ldots, \mathbf{t}_C\}). \tag{4}$$

Details of EMD are provided in Sec. D of the appendix.

**Results.** Tab. 1 shows the results. On the domain shifts, the prompt ensemble had 1%pt accuracy improvement on average compared to the normal prompt. On the other hand, ImageNet-C significantly degraded accuracy, and the ensemble did not result in significant improvement. Moreover, including corruption information in the prompt (Corruption prompt) resulted in accuracy degradation. Notably, the entropy and uniformity loss significantly increased compared to that of the domain shifts on all corruption types, while there is no significant difference in the modality gap. In other words, under image corruption, less information is preserved in the image embeddings, which makes the zero-shot prediction uncertain. Thus, image corruption can distort the semantical alignment between image and text embeddings, but the characteristic of the modality gap differs from that of the domain shift. This is because ViTs or CNNs, which are typically used for image encoders, are vulnerable to such image corruption (Hendrycks & Dietterich, 2019; Qin et al., 2022). Unlike domain shifts, image-corruption data have clean counterparts. In other words, image corruption can be viewed as a Markov chain $X \to X' \to Z$, where $X$ and $X'$ are the original image and its corrupted one, and $Z$ is the embedding. By the data processing inequality, $\mathcal{I}(X; X') \geq \mathcal{I}(X; Z)$, where $\mathcal{I}(\cdot; \cdot)$ is mutual information. By rewriting mutual information with entropy $\mathcal{H}(\cdot)$, we have $\mathcal{H}(X|X') + \mathcal{H}(Z) \leq \mathcal{H}(Z|X)$. When corruption becomes severe, it becomes more difficult to recover the original image $X$ from the corrupted one $X'$, i.e., $\mathcal{H}(X|X')$ increases. Thus, assuming that the fluctuation of $\mathcal{H}(Z|X)$ is small, $\mathcal{H}(Z)$ is likely to decrease, i.e., $Z$ becomes less diverse and distributes less uniformly. Intuitively, as image corruption becomes severe, the semantic content in images becomes more indistinguishable in the embedding space; image embeddings become more similar to each other, resulting in greater uniformity loss (i.e., less information). In such a situation, recovering semantic information from the image embeddings is impossible. Thus, enhancing uniformity first is necessary to maintain semantic information.

Next, to check whether the CLIP recognizes image corruption types, we performed zero-shot classification of the 15 corruption types of input images instead of object categories using the text prompts "`a photo corrupted by [corruption name]`." Tab. 2 shows the results. Most corruption types had poor accuracy, which suggests that CLIP cannot recognize image corruption types. Even in the cases of corruption types

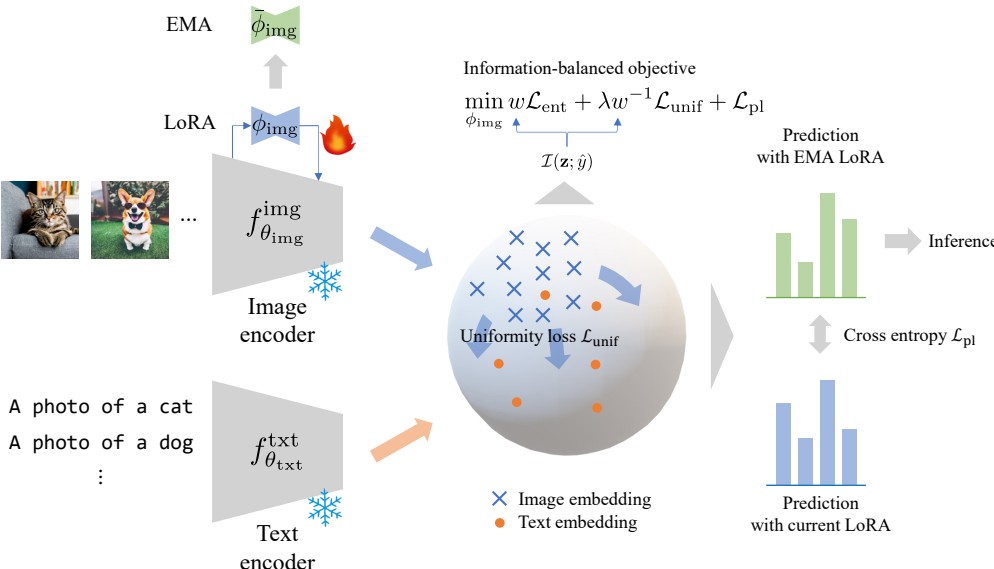

Figure 1: Overview of Uniformity-aware Information-balanced Test-time Adaptation (UnInfo).

with high accuracies (such as defocus blur, motion blur, and fog), the contribution of including the corruption information for object classification is limited or, even worse, as shown in Tab. 1.

From these observations, image corruption causes the modality gap as well as domain shifts, but has fundamentally different properties from domain shifts in terms of the uniformity and entropy, which suggests that the CLIP cannot properly encode images corrupted by the image corruption.

## 4 Uniformity-aware Information-balanced Test-time Adaptation

We introduce our proposed method, *Uniformity-aware Information-balanced Test-time Adaptation (UnInfo)*. Fig. 1 illustrates the overview of UnInfo. The goal of TTA is to perform zero-shot classification on each incoming batch of test images $\{\mathbf{x}_i\}_{i=1}^{B}$ with updating the model parameters to make accurate predictions, where $B$ is the batch size. Since distribution shifts occur on input images and text embeddings are fixed in zero-shot classification, we specifically update $\theta_{\text{img}}$ in our method.

### 4.1 Uniformity-aware Confidence Maximization

Our method's basic approach is to enhance the confidence of predictions via minimizing entropy, which is widely adopted in existing TTA methods for general classification models (Wang et al., 2021; Zhou & Levine, 2021; Niu et al., 2022; Adachi et al., 2023; Enomoto et al., 2024; Zhang et al., 2022a; Adachi et al., 2024). In zero-shot classification with CLIP, the entropy loss is computed by using the prediction probability $\hat{p}_c$ on the basis of similarity, defined in Eq. (1):

$$\mathcal{L}_{\text{ent}} = \frac{1}{B} \sum_{i=1}^{B} \sum_{c=1}^{C} -\hat{p}_{i,c} \log \hat{p}_{i,c}. \tag{5}$$

Minimizing entropy is expected to improve the model performance because entropy is a promising proxy for classification accuracy when the images are appropriately embedded and input information is preserved, e.g., domain shifts. However, the amount of input information preserved in the image embeddings decreases under image corruption as the entropy and uniformity loss increase in Tab. 1, as discussed in Sec. 3. In such cases, the improvement by solely minimizing the entropy is limited since it attempts to leverage less information. In other words, images are less distinguishable in the embedding space. For enhancing information retained

in image embeddings, we aim to improve the distinguishability of image embeddings from each other by minimizing the uniformity loss (Wang & Isola, 2020):

$$\mathcal{L}_{\text{unif}} = \log \frac{1}{B^2} \sum_{i=1}^{B} \sum_{j=1}^{B} \exp\left(-\|\mathbf{z}_i - \mathbf{z}_j\|_2^2\right). \tag{6}$$

We minimize the entropy and uniformity losses simultaneously for refining prediction confidence and making image embeddings distinguishable (i.e., enhancing retained information):

$$\mathcal{L} = \mathcal{L}_{\text{ent}} + \lambda \mathcal{L}_{\text{unif}}, \tag{7}$$

where $\lambda > 0$ is a hyperparameter, and $\mathbf{z}_i, \mathbf{z}_j$ are image embeddings of a batch of input images.

### 4.2 Information-aware Loss Balancing

As described in Secs. 3 and 4.1, minimizing the uniformity loss helps retain input information of image embeddings. However, we found that the importance of uniformity and entropy can dynamically change during TTA. For example, entropy should be prioritized when zero-shot classification works well, e.g., when image corruption is not severe. In fact, uniformity differs depends on corruption types, which is demonstrated in the preliminary experiment (Tab. 1). On the other hand, uniformity loss should be leveraged first, and then entropy should be addressed when zero-shot classification goes to a degenerated solution, e.g., classifying all images into a single class under severe image corruption.

To recognize the current regime and adaptively assign weights to the entropy and uniformity loss, we propose *information-aware loss balancing*. We employ the mutual information between the image embedding $\mathbf{z}$ and prediction $\hat{y}$, denoted by $\mathcal{I}(\mathbf{z}; \hat{y})$, to detect whether classification works without supervision. We assign the weights to the two losses as follows:

$$\mathcal{L} = w\mathcal{L}_{\text{ent}} + \lambda w^{-1}\mathcal{L}_{\text{unif}}, \quad w = \exp(\mathcal{I}(\mathbf{z}; \hat{y}) - \mathcal{I}_0), \tag{8}$$

where $\mathcal{I}_0$ is a hyperparameter to determine the threshold between the two regimes. The mutual information $\mathcal{I}(\mathbf{z}; \hat{y})$ is widely used in representation learning for measuring the quality of features and is computed as follows (Bridle et al., 1991; Krause et al., 2010; Shi & Sha, 2012; Hu et al., 2017; Liang et al., 2020):

$$\mathcal{I}(\mathbf{z}; \hat{y}) = \mathcal{H}(\hat{y}) - \mathcal{H}(\hat{y}|\mathbf{z}) = \sum_{c=1}^{C} -\bar{p}_c \log \bar{p}_c - \frac{1}{B} \sum_{i=1}^{B} \sum_{c=1}^{C} -\hat{p}_{i,c} \log \hat{p}_{i,c}, \tag{9}$$

where $\mathcal{H}(\cdot)$ is entropy and $\bar{p}_c = (1/B) \sum_{i=1}^{B} \hat{p}_{i,c}$. Although $\mathcal{I}(\mathbf{z}; \hat{y})$ is not always a precise estimation of the true mutual information since the batch size $B$ (e.g., 64) can be far smaller than the number of classes $C$ (e.g., 1000), it works on this purpose to verify whether the current classification goes well. Intuitively, $\mathcal{I}(\mathbf{z}; \hat{y})$ takes small values when the image embedding $\mathbf{z}$ is less diverse and predictions are not confident. In such a case, a larger weight is assigned to $\mathcal{L}_{\text{unif}}$ to make the image embeddings diverse and retain more information. In the opposite case, $\mathcal{L}_{\text{ent}}$ is leveraged to make predictions more confident. Note that gradients are not propagated to the weight $w$ since the balance is not an objective to be optimized.

### 4.3 Update with Low-rank Adapters

Although we aim to update the image encoder $f_{\theta_{\text{img}}}^{\text{img}}$ to minimize the proposed loss in Eq. (8), updating the whole $\theta_{\text{img}}$ naively leads to catastrophic model forgetting (Lai et al., 2023; Vesdapunt et al., 2024). To avoid this, we fix the original pre-trained parameter $\theta_{\text{img}}$ and add the LoRA (Hu et al., 2022) to the linear weights in the attention layers, inspired by Zanella & Ben Ayed (2024a). Specifically, given an attention layer at the $l$-th layer

$$\mathbf{h}^{l+1} = \text{softmax}\left((W_{\text{Q}}^l \mathbf{h}^l)(W_{\text{K}}^l \mathbf{h}^l)^\top / \sqrt{d^l}\right) W_{\text{V}}^l \mathbf{h}^l, \tag{10}$$

we attach low-rank matrices to $W_{\cdot}^l$:

$$W_{\cdot}^l \rightarrow W_{\cdot}^l + A_{\cdot}^l B_{\cdot}^l, \tag{11}$$

where $A_{\cdot}^l \in \mathbb{R}^{d^l \times d_{\text{LoRA}}}$ and $B_{\cdot}^l \in \mathbb{R}^{d_{\text{LoRA}} \times d^l}$ ($d_{\text{LoRA}} < d^l$). We denote the LoRA parameters as $\phi_{\text{img}}$.

### 4.4 Knowledge Distillation from EMA Teacher

For stabilizing the adaptation, we track the EMA of $\phi_{\text{img}}$ following previous studies (Wang et al., 2022; Gao et al., 2022; Döbler et al., 2023; Wang et al., 2024a). That is, we update $\bar{\phi}_{\text{img}}$ in every iteration:

$$\bar{\phi}_{\text{img}} \leftarrow m\phi_{\text{img}} + (1 - m)\bar{\phi}_{\text{img}}, \tag{12}$$

where $m \in (0, 1)$ is the momentum parameter. We adopt predictions made with $\bar{\phi}_{\text{img}}$ for inference. Using $\bar{\phi}_{\text{img}}$ as the teacher, we penalize the current LoRA parameters $\phi_{\text{img}}$ (student) to make predictions close to those of the teacher. Specifically, we take the cross entropy between the teacher and student outputs (Wang et al., 2022; Gao et al., 2022):

$$\mathcal{L}_{\text{pl}} = \frac{1}{B} \sum_{i=1}^{B} \sum_{c=1}^{C} -\hat{q}_{i,c} \log \hat{p}_{i,c}, \tag{13}$$

where $\hat{p}_{i,c}$ is the student's output defined in Eq. (1), and $\hat{q}_{i,c}$ is the teacher's output computed in the same way with $\hat{p}_{i,c}$ but using the EMA parameter $\bar{\phi}_{\text{img}}$.

To sum up Secs. 4.1 to 4.4, our objective is as follows:

$$\min_{\phi_{\text{img}}} w\mathcal{L}_{\text{ent}} + \lambda w^{-1}\mathcal{L}_{\text{unif}} + \mathcal{L}_{\text{pl}}. \tag{14}$$

## 5 Experiment

We conducted experiments on TTA under image corruption.

### 5.1 Datasets

**ImageNet-C (Hendrycks & Dietterich, 2019):** This dataset is constructed to evaluate the robustness of vision models. It consists of the corrupted version of the validation set of ImageNet (Deng et al., 2009). ImageNet-C includes 15 types of corruption, such as blur or digital noise. Each corruption type has five severity levels. We used the images corrupted at the highest severity level.

**ImageNet-C-bar (Mintun et al., 2021):** This dataset is constructed to evaluate the robustness of the vision models on a broader range of corruption types. Like ImageNet-C, it also consists of the corrupted version of the ImageNet validation set. ImageNet-C-bar includes 10 corruption types that are algorithmically selected to be dissimilar from ImageNet-C. We used the images corrupted at the highest severity level, as in the ImageNet-C case.

### 5.2 Implementation

We used the ViT-B/16 CLIP pre-trained on the LAION-2B dataset (Schuhmann et al., 2022). We downloaded the pre-trained weights via HuggingFace[1]. We set the temperature of the zero-shot prediction $\tau = 0.01$. For our UnInfo, we used the AdamW optimizer (Loshchilov & Hutter, 2019) with learning rate = 0.001, weight decay = 0.01, and batch size $B = 64$. We set the weight of the uniformity loss $\lambda = 1$ and the threshold of the information-aware loss balancing weight $\mathcal{I}_0 = 3$. We chose these hyperparameters by using a few corruption types in ImageNet-C (brightness, defocus blur, elastic transform, and Gaussian noise) and used the selected hyperparameters for the others as default. $\mathcal{I}_0$ was chosen on the basis of the mutual information computed on clean data (ImageNet). For the LoRA in UnInfo, we used the implementation provided by Zanella & Ben Ayed (2024a)[2]. We set the LoRA hyperparameters, such as $\alpha$ and the rank $d_{\text{LoRA}}$, to the default values and the momentum of EMA $m = 0.001$. The LoRA matrices $A^l$ and $B^l$ are initialized by the Kaiming uniform initialization (He et al., 2015) and zero, respectively, so that image embeddings are unaffected by random $\phi_{\text{img}}$ at the initial state.

---

[1] https://huggingface.co/laion/CLIP-ViT-B-16-laion2B-s34B-b88K
[2] https://github.com/MaxZanella/CLIP-LoRA

Table 3: Test accuracy (%) on ImageNet-C. The numbers (1), (5), and (10) presented with the method names are the shot numbers per class $n$ used for the few-shot adaptation methods.

| Method | Defocus blur | Glass blur | Motion blur | Zoom blur | Contrast | Elastic transform | Jpeg compression | Pixelate | Gaussian noise | Impulse noise | Shot noise | Brightness | Fog | Frost | Snow | Mean |
|---|---|---|---|---|---|---|---|---|---|---|---|---|---|---|---|---|
| No-adapt | 28.31 | 11.89 | 19.16 | 17.61 | 17.87 | 13.22 | 36.79 | 37.01 | 6.08 | 6.17 | 7.85 | 54.89 | 34.55 | 27.35 | 27.67 | 23.09 |
| Linear probing (1) | $11.53_{\pm0.59}$ | $6.15_{\pm0.20}$ | $8.79_{\pm0.19}$ | $9.02_{\pm0.41}$ | $5.87_{\pm0.13}$ | $8.17_{\pm0.39}$ | $15.57_{\pm0.35}$ | $16.36_{\pm0.46}$ | $2.79_{\pm0.15}$ | $2.76_{\pm0.14}$ | $3.41_{\pm0.16}$ | $27.05_{\pm0.58}$ | $16.43_{\pm0.53}$ | $9.88_{\pm0.19}$ | $11.87_{\pm0.41}$ | $10.38_{\pm0.02}$ |
| Linear probing (5) | $19.55_{\pm0.18}$ | $10.53_{\pm0.10}$ | $15.16_{\pm0.24}$ | $15.25_{\pm0.11}$ | $10.04_{\pm0.41}$ | $14.47_{\pm0.05}$ | $25.68_{\pm0.08}$ | $27.02_{\pm0.09}$ | $4.76_{\pm0.15}$ | $4.95_{\pm0.22}$ | $5.62_{\pm0.16}$ | $43.28_{\pm0.34}$ | $26.10_{\pm0.15}$ | $17.31_{\pm0.12}$ | $19.48_{\pm0.11}$ | $17.28_{\pm0.04}$ |
| Linear probing (10) | $23.84_{\pm0.14}$ | $12.72_{\pm0.13}$ | $17.97_{\pm0.05}$ | $18.57_{\pm0.32}$ | $12.45_{\pm0.11}$ | $17.69_{\pm0.10}$ | $30.50_{\pm0.24}$ | $32.32_{\pm0.18}$ | $5.78_{\pm0.19}$ | $6.04_{\pm0.13}$ | $6.83_{\pm0.21}$ | $49.90_{\pm0.22}$ | $31.15_{\pm0.32}$ | $21.49_{\pm0.29}$ | $23.63_{\pm0.33}$ | $20.73_{\pm0.06}$ |
| Tip-adapter (Zhang et al., 2022b) (1) | $19.00_{\pm0.31}$ | $9.09_{\pm0.22}$ | $13.92_{\pm0.32}$ | $14.04_{\pm0.36}$ | $9.53_{\pm0.15}$ | $12.60_{\pm0.31}$ | $26.98_{\pm0.12}$ | $27.19_{\pm0.29}$ | $4.06_{\pm0.12}$ | $4.02_{\pm0.13}$ | $5.15_{\pm0.21}$ | $44.92_{\pm0.25}$ | $25.99_{\pm0.35}$ | $18.10_{\pm0.21}$ | $19.28_{\pm0.41}$ | $16.92_{\pm0.07}$ |
| Tip-adapter (5) | $23.43_{\pm0.27}$ | $12.27_{\pm0.03}$ | $17.61_{\pm0.14}$ | $17.76_{\pm0.20}$ | $11.56_{\pm0.30}$ | $16.82_{\pm0.17}$ | $30.87_{\pm0.21}$ | $32.09_{\pm0.20}$ | $4.88_{\pm0.24}$ | $4.86_{\pm0.12}$ | $6.03_{\pm0.19}$ | $49.88_{\pm0.39}$ | $30.59_{\pm0.18}$ | $21.43_{\pm0.21}$ | $22.82_{\pm0.15}$ | $20.19_{\pm0.06}$ |
| Tip-adapter (10) | $26.11_{\pm0.11}$ | $13.99_{\pm0.22}$ | $19.85_{\pm0.36}$ | $20.23_{\pm0.23}$ | $12.99_{\pm0.15}$ | $19.26_{\pm0.22}$ | $33.03_{\pm0.11}$ | $34.60_{\pm0.21}$ | $5.52_{\pm0.06}$ | $5.82_{\pm0.08}$ | $6.82_{\pm0.15}$ | $52.40_{\pm0.24}$ | $33.15_{\pm0.19}$ | $24.01_{\pm0.32}$ | $24.83_{\pm0.19}$ | $22.17_{\pm0.05}$ |
| TPT (Shu et al., 2022) | $29.66_{\pm0.05}$ | $12.87_{\pm0.03}$ | $21.11_{\pm0.02}$ | $20.54_{\pm0.00}$ | $20.11_{\pm0.08}$ | $15.21_{\pm0.03}$ | $39.27_{\pm0.01}$ | $41.14_{\pm0.03}$ | $6.48_{\pm0.03}$ | $6.74_{\pm0.02}$ | $8.50_{\pm0.01}$ | $57.35_{\pm0.04}$ | $37.05_{\pm0.07}$ | $29.81_{\pm0.06}$ | $30.23_{\pm0.04}$ | $25.07_{\pm0.01}$ |
| ZERO (Farina et al., 2024) | $26.85_{\pm0.05}$ | $8.86_{\pm0.06}$ | $18.11_{\pm0.04}$ | $19.89_{\pm0.04}$ | $16.46_{\pm0.02}$ | $12.38_{\pm0.00}$ | $35.09_{\pm0.05}$ | $37.44_{\pm0.04}$ | $3.69_{\pm0.02}$ | $5.33_{\pm0.02}$ | $4.43_{\pm0.03}$ | $53.35_{\pm0.08}$ | $33.50_{\pm0.01}$ | $26.56_{\pm0.10}$ | $27.42_{\pm0.01}$ | $21.96_{\pm0.02}$ |
| MTA (Zanella & Ben Ayed, 2024b) | $27.79_{\pm0.10}$ | $11.29_{\pm0.04}$ | $19.25_{\pm0.04}$ | $18.88_{\pm0.02}$ | $21.18_{\pm0.05}$ | $13.92_{\pm0.03}$ | $37.23_{\pm0.04}$ | $38.95_{\pm0.04}$ | $2.41_{\pm0.01}$ | $2.87_{\pm0.00}$ | $2.96_{\pm0.01}$ | $53.56_{\pm0.02}$ | $34.32_{\pm0.05}$ | $28.02_{\pm0.07}$ | $28.66_{\pm0.04}$ | $22.75_{\pm0.02}$ |
| TDA (Karmanov et al., 2024) | $30.13_{\pm0.04}$ | $14.59_{\pm0.08}$ | $22.10_{\pm0.03}$ | $21.09_{\pm0.06}$ | $19.59_{\pm0.06}$ | $17.15_{\pm0.08}$ | $38.58_{\pm0.03}$ | $39.53_{\pm0.03}$ | $7.23_{\pm0.01}$ | $7.45_{\pm0.03}$ | $9.34_{\pm0.07}$ | $56.99_{\pm0.04}$ | $38.09_{\pm0.06}$ | $30.24_{\pm0.05}$ | $31.02_{\pm0.02}$ | $25.54_{\pm0.03}$ |
| DMN (Zhang et al., 2024b) | $29.81_{\pm0.01}$ | $12.97_{\pm0.00}$ | $20.89_{\pm0.00}$ | $19.52_{\pm0.00}$ | $18.69_{\pm0.00}$ | $14.36_{\pm0.00}$ | $38.93_{\pm0.00}$ | $39.69_{\pm0.00}$ | $6.65_{\pm0.00}$ | $6.89_{\pm0.00}$ | $8.58_{\pm0.00}$ | $57.65_{\pm0.00}$ | $36.57_{\pm0.00}$ | $28.98_{\pm0.00}$ | $29.66_{\pm0.00}$ | $24.65_{\pm0.00}$ |
| Mint (Bao et al., 2025) | $24.85_{\pm0.08}$ | $\mathbf{28.49_{\pm0.06}}$ | $\mathbf{25.24_{\pm0.21}}$ | $\mathbf{28.45_{\pm0.02}}$ | $\mathbf{24.16_{\pm0.12}}$ | $28.39_{\pm0.09}$ | $\mathbf{43.88_{\pm0.03}}$ | $\mathbf{43.60_{\pm0.16}}$ | $8.05_{\pm0.06}$ | $7.94_{\pm0.14}$ | $7.71_{\pm0.12}$ | $\mathbf{58.08_{\pm0.06}}$ | $\mathbf{43.61_{\pm0.12}}$ | $31.07_{\pm0.10}$ | $\mathbf{33.50_{\pm0.02}}$ | $\mathbf{29.14_{\pm0.01}}$ |
| BATCLIP (Maharana et al., 2025) | $22.48_{\pm0.34}$ | $24.47_{\pm0.16}$ | $24.72_{\pm0.12}$ | $26.64_{\pm0.11}$ | $23.69_{\pm0.53}$ | $\mathbf{31.79_{\pm0.30}}$ | $33.49_{\pm0.17}$ | $35.93_{\pm0.16}$ | $\mathbf{12.56_{\pm0.14}}$ | $\mathbf{13.01_{\pm0.20}}$ | $\mathbf{14.08_{\pm0.02}}$ | $50.03_{\pm0.04}$ | $37.91_{\pm0.17}$ | $28.33_{\pm0.26}$ | $30.21_{\pm0.04}$ | $27.29_{\pm0.03}$ |
| UnInfo (ours) | $\mathbf{31.51_{\pm0.19}}$ | $16.76_{\pm1.62}$ | $23.47_{\pm0.74}$ | $20.40_{\pm0.80}$ | $22.81_{\pm0.27}$ | $16.59_{\pm0.40}$ | $42.03_{\pm0.23}$ | $42.38_{\pm0.26}$ | $7.56_{\pm3.50}$ | $10.60_{\pm0.41}$ | $11.36_{\pm0.86}$ | $57.75_{\pm0.11}$ | $39.16_{\pm0.34}$ | $\mathbf{31.65_{\pm0.48}}$ | $32.40_{\pm0.13}$ | $27.10_{\pm0.12}$ |

## 5.3 Baseline

We compared our UnInfo with existing TTA methods for CLIP zero-shot classification and few-shot adaptation methods. For few-shot methods, we split $n \in \{1, 5, 10\}$ test samples per class, used them for adaptation, and then tested the model on the rest test samples.

**No-adapt**: Just performs zero-shot classification without adaptation.

**LP** (Linear probing): Trains a linear classifier head with few-shot labeled samples from the test set.

**Tip-adapter** (Zhang et al., 2022b): Modifies predictions by using cached features and logits of few-shot labeled samples from the test set.

**TPT** (Test-time prompt tuning) (Shu et al., 2022): Updates the text token embeddings corresponding to the words of a text template, e.g., "`a photo of a`," to minimize the marginal entropy over augmented views of an input image.

**TDA** (Training-free dynamic adapter) (Karmanov et al., 2024): Constructs positive and negative caches on the basis of prediction confidence on incoming test images and modifies predictions of subsequent inputs.

**ZERO** (Farina et al., 2024): Performs voting within predictions of augmented views of an input image.

**MTA** (MeanShift for test-time augmentation) (Zanella & Ben Ayed, 2024b): Selects reliable image embeddings among augmented views and modifies the image embedding.

**DMN** (Dual memory network) (Zhang et al., 2024b): caches test image embeddings in a memory and utilizes similar cached embeddings for the current prediction.

**Mint** (Bao et al., 2025): Updates the image encoder to optimize inter- and intra-class variances to maintain separability of image embeddings.

**BATCLIP** (Maharana et al., 2025): Updates both image and text encoders to refine confidence and improve separability while maintaining image-text matching.

## 5.4 Results

### 5.4.1 Adaptation Performance

We evaluated each corruption type's test classification accuracy on ImageNet-C and C-bar. We ran TTA three times with different random seeds for each method and corruption type, and report the mean score. We adopt a batched online evaluation protocol, in which model updates and batch evaluations are performed alternately (i.e., each sample is accessed once). Tabs. 3 and 4 show the results. Our UnInfo consistently improved accuracy compared to No-adapt regardless of corruption types. Intriguingly, the baselines, even the few-shot methods that use labeled test samples for adaptation, sometimes underperformed No-adapt. This is because the baseline methods aim to refine text and/or image embeddings in a post-hoc manner during testing. While corrupted images are not appropriately encoded, as discussed in Sec. 3, the encoders themselves remain fixed. In contrast, UnInfo successfully adapts to image corruption by updating the image encoder with LoRA. Although recent CLIP-TTA methods, Mint (Bao et al., 2025) and BATCLIP (Maharana et al., 2025), outperformed UnInfo on average in Tab. 3, they performed worse than No-adapt on some corruption types. In contrast, UnInfo stably worked on all corruption types.

Table 4: Test accuracy (%) on ImageNet-C-bar. The numbers (1), (5), and (10) presented with the method names are the shot numbers per class $n$ used for the few-shot adaptation methods.

| Method | Blue noise sample | Brownish noise | Caustic refraction | Checkerboard cutout | Cocentric sine waves | Inverse sparkles | Perlin noise | Plasma noise | Single frequency greyscale | Sparkles | Mean |
|---|---|---|---|---|---|---|---|---|---|---|---|
| No-adapt | 21.87 | 46.66 | 39.55 | 45.51 | 10.05 | 19.57 | 51.21 | 20.69 | 16.10 | 50.00 | 32.12 |
| Linear probing (1) | $9.05_{\pm0.08}$ | $24.22_{\pm0.39}$ | $17.77_{\pm0.17}$ | $21.99_{\pm0.39}$ | $3.74_{\pm0.05}$ | $8.59_{\pm0.22}$ | $24.87_{\pm0.28}$ | $10.04_{\pm0.30}$ | $4.23_{\pm0.11}$ | $26.19_{\pm0.45}$ | $15.07_{\pm0.08}$ |
| Linear probing (5) | $15.45_{\pm0.17}$ | $38.05_{\pm0.34}$ | $29.12_{\pm0.08}$ | $35.58_{\pm0.26}$ | $6.89_{\pm0.19}$ | $14.28_{\pm0.48}$ | $39.24_{\pm0.33}$ | $15.43_{\pm0.22}$ | $6.82_{\pm0.17}$ | $40.55_{\pm0.17}$ | $24.14_{\pm0.08}$ |
| Linear probing (10) | $18.58_{\pm0.08}$ | $44.03_{\pm0.18}$ | $34.96_{\pm0.36}$ | $41.54_{\pm0.03}$ | $8.83_{\pm0.12}$ | $17.36_{\pm0.25}$ | $45.82_{\pm0.19}$ | $18.41_{\pm0.14}$ | $9.05_{\pm0.18}$ | $47.20_{\pm0.18}$ | $28.58_{\pm0.02}$ |
| Tip-adapter (Zhang et al., 2022b) (1) | $14.95_{\pm0.19}$ | $37.35_{\pm0.17}$ | $29.99_{\pm0.15}$ | $35.58_{\pm0.31}$ | $7.12_{\pm0.12}$ | $14.82_{\pm0.10}$ | $40.06_{\pm0.14}$ | $15.46_{\pm0.11}$ | $10.06_{\pm0.05}$ | $41.29_{\pm0.18}$ | $24.67_{\pm0.05}$ |
| Tip-adapter (5) | $18.41_{\pm0.08}$ | $43.08_{\pm0.26}$ | $34.86_{\pm0.27}$ | $40.79_{\pm0.09}$ | $8.37_{\pm0.14}$ | $17.58_{\pm0.15}$ | $45.58_{\pm0.12}$ | $18.15_{\pm0.09}$ | $10.24_{\pm0.16}$ | $46.44_{\pm0.13}$ | $28.35_{\pm0.08}$ |
| Tip-adapter (10) | $20.22_{\pm0.38}$ | $45.93_{\pm0.20}$ | $37.59_{\pm0.19}$ | $43.65_{\pm0.14}$ | $9.77_{\pm0.08}$ | $19.62_{\pm0.25}$ | $48.52_{\pm0.26}$ | $20.07_{\pm0.05}$ | $11.51_{\pm0.07}$ | $49.26_{\pm0.23}$ | $30.61_{\pm0.08}$ |
| TPT (Shu et al., 2022) | $24.63_{\pm0.07}$ | $49.86_{\pm0.04}$ | $42.53_{\pm0.12}$ | $46.36_{\pm0.01}$ | $10.96_{\pm0.07}$ | $21.93_{\pm0.03}$ | $54.31_{\pm0.03}$ | $23.16_{\pm0.03}$ | $17.43_{\pm0.08}$ | $52.20_{\pm0.02}$ | $34.34_{\pm0.01}$ |
| ZERO (Farina et al., 2024) | $25.65_{\pm0.06}$ | $45.60_{\pm0.01}$ | $41.14_{\pm0.04}$ | $45.30_{\pm0.04}$ | $10.59_{\pm0.02}$ | $22.79_{\pm0.02}$ | $49.76_{\pm0.02}$ | $20.42_{\pm0.06}$ | $\underline{19.81}_{\pm0.07}$ | $45.81_{\pm0.10}$ | $32.69_{\pm0.02}$ |
| MTA (Zanella & Ben Ayed, 2024b) | $23.75_{\pm0.02}$ | $45.13_{\pm0.03}$ | $40.28_{\pm0.12}$ | $45.05_{\pm0.08}$ | $9.93_{\pm0.01}$ | $20.53_{\pm0.04}$ | $50.50_{\pm0.01}$ | $20.04_{\pm0.02}$ | $19.32_{\pm0.06}$ | $45.89_{\pm0.02}$ | $32.04_{\pm0.02}$ |
| TDA (Karmanov et al., 2024) | $24.53_{\pm0.04}$ | $50.30_{\pm0.02}$ | $\underline{42.90}_{\pm0.02}$ | $49.85_{\pm0.02}$ | $12.69_{\pm0.03}$ | $22.56_{\pm0.04}$ | $53.89_{\pm0.09}$ | $24.75_{\pm0.09}$ | $17.77_{\pm0.01}$ | $55.00_{\pm0.10}$ | $35.42_{\pm0.01}$ |
| DMN (Zhang et al., 2024b) | $23.21_{\pm0.00}$ | $50.42_{\pm0.00}$ | $\underline{41.37}_{\pm0.00}$ | $47.08_{\pm0.00}$ | $10.95_{\pm0.00}$ | $20.98_{\pm0.00}$ | $54.15_{\pm0.00}$ | $22.95_{\pm0.00}$ | $16.91_{\pm0.00}$ | $\underline{53.09}_{\pm0.00}$ | $34.11_{\pm0.00}$ |
| Mint (Bao et al., 2025) | $\mathbf{29.22}_{\pm0.07}$ | $\mathbf{53.98}_{\pm0.07}$ | $39.49_{\pm0.17}$ | $\mathbf{53.12}_{\pm0.05}$ | $\underline{15.48}_{\pm0.05}$ | $\mathbf{24.31}_{\pm0.25}$ | $54.87_{\pm0.06}$ | $\mathbf{32.48}_{\pm0.05}$ | $\mathbf{20.13}_{\pm0.07}$ | $\mathbf{57.56}_{\pm0.06}$ | $\mathbf{38.06}_{\pm0.04}$ |
| BATCLIP (Maharana et al., 2025) | $26.92_{\pm0.17}$ | $47.44_{\pm0.25}$ | $36.60_{\pm0.25}$ | $46.63_{\pm0.10}$ | $\mathbf{16.28}_{\pm0.15}$ | $23.52_{\pm0.09}$ | $50.21_{\pm0.12}$ | $28.91_{\pm0.11}$ | $18.91_{\pm0.10}$ | $50.69_{\pm0.20}$ | $34.61_{\pm0.11}$ |
| UnInfo (ours) | $\underline{26.78}_{\pm0.64}$ | $51.21_{\pm0.32}$ | $\mathbf{43.73}_{\pm0.40}$ | $50.03_{\pm1.89}$ | $12.49_{\pm0.28}$ | $23.58_{\pm0.69}$ | $\mathbf{55.22}_{\pm0.23}$ | $\underline{24.88}_{\pm0.22}$ | $19.67_{\pm0.35}$ | $53.74_{\pm0.71}$ | $36.13_{\pm0.16}$ |

Table 5: Ablation of UnInfo on ImageNet-C.

| Method | Defocus blur | Glass blur | Motion blur | Zoom blur | Contrast | Elastic transform | Jpeg compression | Pixelate | Gaussian noise | Impulse noise | Shot noise | Brightness | Fog | Frost | Snow | Mean |
|---|---|---|---|---|---|---|---|---|---|---|---|---|---|---|---|---|
| $\mathcal{L}_{\mathrm{ent}}$ | $0.21_{\pm0.01}$ | $0.15_{\pm0.01}$ | $0.15_{\pm0.01}$ | $0.17_{\pm0.01}$ | $0.19_{\pm0.02}$ | $0.14_{\pm0.00}$ | $0.32_{\pm0.01}$ | $0.30_{\pm0.01}$ | $0.11_{\pm0.00}$ | $0.11_{\pm0.01}$ | $0.11_{\pm0.00}$ | $0.70_{\pm0.08}$ | $0.24_{\pm0.01}$ | $0.19_{\pm0.03}$ | $0.22_{\pm0.02}$ | $0.22_{\pm0.01}$ |
| $\mathcal{L}_{\mathrm{ent}} + \mathcal{L}_{\mathrm{pl}}$ | $30.63_{\pm0.18}$ | $15.56_{\pm1.78}$ | $22.61_{\pm0.77}$ | $19.54_{\pm0.26}$ | $21.83_{\pm0.18}$ | $15.99_{\pm0.15}$ | $41.03_{\pm0.18}$ | $40.82_{\pm0.47}$ | $3.31_{\pm3.08}$ | $6.68_{\pm0.58}$ | $8.06_{\pm0.56}$ | $57.17_{\pm0.09}$ | $37.78_{\pm0.28}$ | $30.84_{\pm0.18}$ | $31.29_{\pm0.29}$ | $25.54_{\pm0.35}$ |
| $\mathcal{L}_{\mathrm{mi}}$ | $0.66_{\pm0.08}$ | $0.70_{\pm0.07}$ | $0.66_{\pm0.05}$ | $1.01_{\pm0.09}$ | $0.58_{\pm0.01}$ | $1.02_{\pm0.16}$ | $1.26_{\pm0.10}$ | $1.35_{\pm0.15}$ | $0.22_{\pm0.04}$ | $0.19_{\pm0.04}$ | $0.21_{\pm0.05}$ | $2.46_{\pm0.30}$ | $1.73_{\pm0.13}$ | $1.01_{\pm0.14}$ | $0.98_{\pm0.13}$ | $0.94_{\pm0.02}$ |
| $\mathcal{L}_{\mathrm{mi}} + \mathcal{L}_{\mathrm{pl}}$ | $30.29_{\pm0.25}$ | $\mathbf{17.77}_{\pm1.07}$ | $\mathbf{23.75}_{\pm0.49}$ | $\mathbf{21.35}_{\pm0.36}$ | $22.21_{\pm0.04}$ | $\mathbf{17.61}_{\pm0.91}$ | $40.60_{\pm0.19}$ | $41.08_{\pm0.34}$ | $\underline{7.26}_{\pm4.37}$ | $\mathbf{11.19}_{\pm0.21}$ | $\mathbf{12.29}_{\pm0.33}$ | $56.52_{\pm0.23}$ | $37.80_{\pm0.54}$ | $30.64_{\pm0.61}$ | $\mathbf{32.16}_{\pm0.03}$ | $26.84_{\pm0.27}$ |
| $\mathcal{L}_{\mathrm{ent}} + \mathcal{L}_{\mathrm{unif}} + \mathcal{L}_{\mathrm{pl}}$ | $31.08_{\pm0.07}$ | $16.48_{\pm1.62}$ | $23.40_{\pm0.48}$ | $20.32_{\pm0.42}$ | $22.49_{\pm0.12}$ | $16.67_{\pm0.23}$ | $41.26_{\pm0.35}$ | $41.26_{\pm0.45}$ | $3.90_{\pm3.70}$ | $7.87_{\pm0.59}$ | $7.17_{\pm3.64}$ | $57.24_{\pm0.04}$ | $38.74_{\pm0.20}$ | $31.24_{\pm0.21}$ | $32.01_{\pm0.42}$ | $26.07_{\pm0.26}$ |
| $\mathcal{L}_{\mathrm{ent}} + \mathcal{L}_{\mathrm{unif}} + \mathcal{L}_{\mathrm{pl}}$+Balancing (UnInfo) | $\mathbf{31.51}_{\pm0.19}$ | $16.76_{\pm1.62}$ | $23.47_{\pm0.74}$ | $20.40_{\pm0.80}$ | $\mathbf{22.81}_{\pm0.27}$ | $16.59_{\pm0.40}$ | $\mathbf{42.03}_{\pm0.23}$ | $\mathbf{42.38}_{\pm0.26}$ | $\mathbf{7.56}_{\pm3.50}$ | $10.60_{\pm0.41}$ | $11.36_{\pm0.86}$ | $\mathbf{57.75}_{\pm0.11}$ | $\mathbf{39.16}_{\pm0.34}$ | $\mathbf{31.65}_{\pm0.48}$ | $\mathbf{32.40}_{\pm0.13}$ | $\mathbf{27.10}_{\pm0.12}$ |

## 5.4.2 Ablation Study

Here, we examined the effect of each component in UnInfo: uniformity-aware confidence maximization, information-aware loss balancing, and knowledge distillation from the EMA teacher. Tabs. 5 and 6 show the results. Solely minimizing the entropy loss $\mathcal{L}_{\mathrm{ent}}$ resulted in catastrophically poor accuracy in all cases. This is because image corruption affects uniformity, as observed in Sec. 3; entropy can assign unreasonably high confidence to wrong classes, resulting in overfitting quickly. In contrast, incorporating the knowledge distillation loss $\mathcal{L}_{\mathrm{pl}}$ drastically improved the stability. The uniformity loss $\mathcal{L}_{\mathrm{unif}}$ further improved accuracy compared to $\mathcal{L}_{\mathrm{ent}} + \mathcal{L}_{\mathrm{pl}}$. However, its improvement is sometimes marginal because it overlooks the balance between entropy and uniformity, as Sec. 4.2 describes. Thus, adding the balancing further improved accuracy by properly enhancing entropy or uniformity. Specifically, we observed significant improvements in difficult corruption types that produce high uniformity loss, such as noise corruption. In such cases, uniformity should be recovered first before minimizing entropy loss. UnInfo successfully controls the priority of the losses.

On the other hand, instead of uniformity and balancing, one may directly use mutual information in Eq. (9) as a loss function $\mathcal{L}_{\mathrm{mi}} = -\mathcal{I}(\mathbf{z}; \hat{y})$ that can enhance the embeddings' diversity while refining the prediction confidence. However, solely minimizing $\mathcal{L}_{\mathrm{mi}}$ resulted in collapse because it is optimal for $\mathcal{L}_{\mathrm{mi}}$ to make each image embedding in a batch identical to a text embedding, regardless of ground truth, making prediction diverse and confident, but suboptimal for the image embeddings' diversity. Although combining with the

Table 6: Ablation of UnInfo on ImageNet-C-bar.

| Method | Blue noise sample | Brownish noise | Caustic refraction | Checkerboard cutout | Cocentric sine waves | Inverse sparkles | Perlin noise | Plasma noise | Single frequency greyscale | Sparkles | Mean |
|---|---|---|---|---|---|---|---|---|---|---|---|
| $\mathcal{L}_{\mathrm{ent}}$ | $0.18_{\pm0.02}$ | $0.44_{\pm0.07}$ | $0.21_{\pm0.01}$ | $0.58_{\pm0.16}$ | $0.12_{\pm0.01}$ | $0.13_{\pm0.00}$ | $0.52_{\pm0.07}$ | $0.15_{\pm0.01}$ | $0.14_{\pm0.01}$ | $0.41_{\pm0.20}$ | $0.29_{\pm0.01}$ |
| $\mathcal{L}_{\mathrm{ent}} + \mathcal{L}_{\mathrm{pl}}$ | $26.20_{\pm0.37}$ | $49.67_{\pm0.17}$ | $42.81_{\pm0.45}$ | $48.87_{\pm1.72}$ | $11.67_{\pm0.34}$ | $22.78_{\pm0.63}$ | $53.94_{\pm0.28}$ | $23.85_{\pm0.55}$ | $19.48_{\pm0.24}$ | $49.63_{\pm3.89}$ | $34.89_{\pm0.23}$ |
| $\mathcal{L}_{\mathrm{mi}}$ | $0.57_{\pm0.10}$ | $2.32_{\pm0.28}$ | $1.56_{\pm0.26}$ | $2.49_{\pm0.17}$ | $0.48_{\pm0.05}$ | $0.90_{\pm0.03}$ | $2.65_{\pm0.19}$ | $0.87_{\pm0.15}$ | $0.47_{\pm0.03}$ | $3.14_{\pm0.10}$ | $1.54_{\pm0.03}$ |
| $\mathcal{L}_{\mathrm{mi}} + \mathcal{L}_{\mathrm{pl}}$ | $26.45_{\pm1.00}$ | $48.96_{\pm0.53}$ | $42.69_{\pm0.16}$ | $\mathbf{51.20}_{\pm0.21}$ | $\mathbf{13.52}_{\pm0.47}$ | $22.78_{\pm0.32}$ | $53.78_{\pm0.21}$ | $\underline{24.41}_{\pm0.39}$ | $19.43_{\pm0.11}$ | $\mathbf{55.34}_{\pm0.18}$ | $35.86_{\pm0.21}$ |
| $\mathcal{L}_{\mathrm{ent}} + \mathcal{L}_{\mathrm{unif}} + \mathcal{L}_{\mathrm{pl}}$ | $26.75_{\pm0.34}$ | $50.28_{\pm0.39}$ | $43.14_{\pm0.04}$ | $49.59_{\pm0.91}$ | $12.08_{\pm0.23}$ | $23.36_{\pm0.56}$ | $54.47_{\pm0.36}$ | $24.27_{\pm0.41}$ | $\mathbf{19.69}_{\pm0.20}$ | $54.27_{\pm1.50}$ | $35.79_{\pm0.21}$ |
| $\mathcal{L}_{\mathrm{ent}} + \mathcal{L}_{\mathrm{unif}} + \mathcal{L}_{\mathrm{pl}}$+Balancing (UnInfo) | $\mathbf{26.78}_{\pm0.64}$ | $\mathbf{51.21}_{\pm0.32}$ | $\mathbf{43.73}_{\pm0.40}$ | $50.03_{\pm1.89}$ | $12.49_{\pm0.28}$ | $\mathbf{23.58}_{\pm0.69}$ | $\mathbf{55.22}_{\pm0.23}$ | $\mathbf{24.88}_{\pm0.22}$ | $19.67_{\pm0.35}$ | $53.74_{\pm0.71}$ | $\mathbf{36.13}_{\pm0.16}$ |

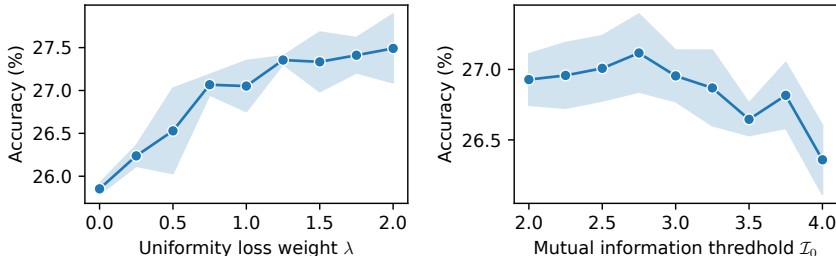

Figure 2: Sensitivity analysis of the uniformity loss weight $\lambda$ (left), and the mutual information threshold $\mathcal{I}_0$ used in the loss balancing (right). The mean and standard deviation of the test accuracy calculated over the ImageNet-C corruptions are plotted.

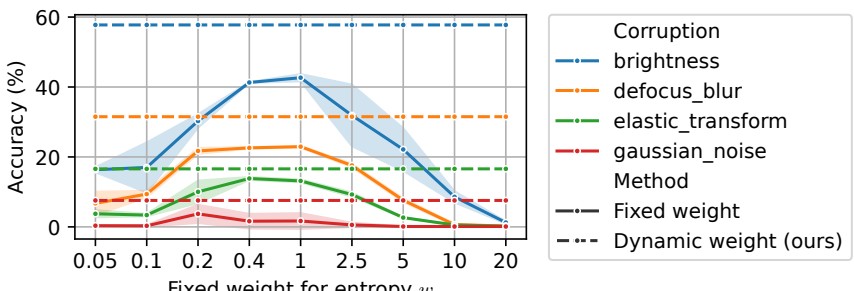

Figure 3: Sensitivity analysis of the balancing weight $w$ with being fixed. The dashed lines represent accuracies when $w$ is dynamically updated with our balancing mechanism described in Sec. 4.2.

EMA teacher ($\mathcal{L}_{\mathrm{mi}}+\mathcal{L}_{\mathrm{pl}}$) significantly improved the accuracy and sometimes outperformed UnInfo, enhancing uniformity and adaptive balancing had better results on average.

### 5.4.3 Sensitivity Analysis

Fig. 2 plots the sensitivity analysis of the hyperparameters of UnInfo. We changed the weight of the uniformity loss $\lambda$ in Eq. (8) and the threshold of the mutual information $\mathcal{I}_0$ for the loss balancing. We ran UnInfo on the ImageNet-C corruptions and reported the average test accuracy. Increasing $\lambda$ produces better accuracies, mainly in $0.0 \leq \lambda \leq 0.75$, suggesting the efficacy of the uniformity loss, and further increasing $\lambda$ beyond 1.0 results in slightly higher accuracy. On the other hand, varying $\mathcal{I}_0$ hits the best accuracy when $\mathcal{I}_0 = 2.75$ but slightly affects the accuracy within $2.0 \leq \mathcal{I}_0 \leq 3.25$. However, increasing $\mathcal{I}_0$ too much deteriorates the accuracy because it controls the bias of the balance between entropy and uniformity. When $\mathcal{I}_0$ is too high, the uniformity loss is constantly overweighted, and the entropy is no longer optimized.

For checking the effectiveness of the dynamic weight by information-aware loss balancing, we fixed the weight $w$ in Eq. (8). Fig. 3 shows the accuracy with fixed value of $w$ being changed. Although leveraging both entropy and uniformity ($0.2 \leq w \leq 1$) produces higher accuracy regardless of corruption types, dynamic $w$ had significant improvements compared to the best accuracies of fixed $w$.

### 5.4.4 Qualitative Analysis

Fig. 4 plots the evolution of the dynamic weights $w$ and $w^{-1}$ in Eq. (8). For easy image corruption types on which No-adapt produced relatively high accuracy, such as brightness in (a), the weight for the entropy $w$ quickly increased, and the weight for uniformity loss $w^{-1}$ was suppressed. This is because image embeddings under the brightness corruption retain information for classification; solely addressing entropy can improve accuracy. The defocus blur in (b) and elastic transform in (c) also showed similar evolutions in which the

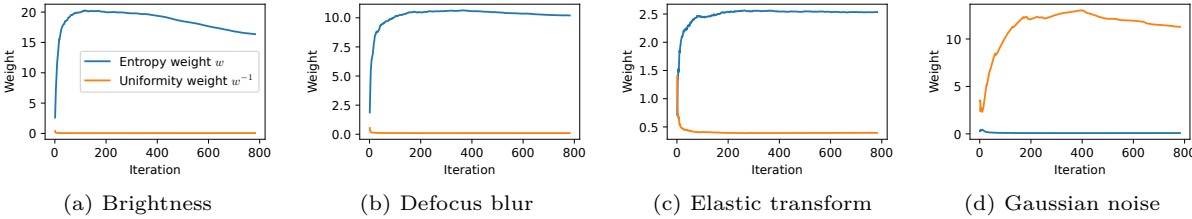

(a) Brightness      (b) Defocus blur      (c) Elastic transform      (d) Gaussian noise

Figure 4: Evolution of the information-aware loss balancing weights. The weights are adaptively assigned to the entropy and uniformity losses by the difficulty of distribution shifts. A larger weight is assigned to the entropy for easy distribution shifts such as brightness in (a). In contrast, the uniformity loss is prioritized for difficult distribution shifts such as Gaussian noise in (d).

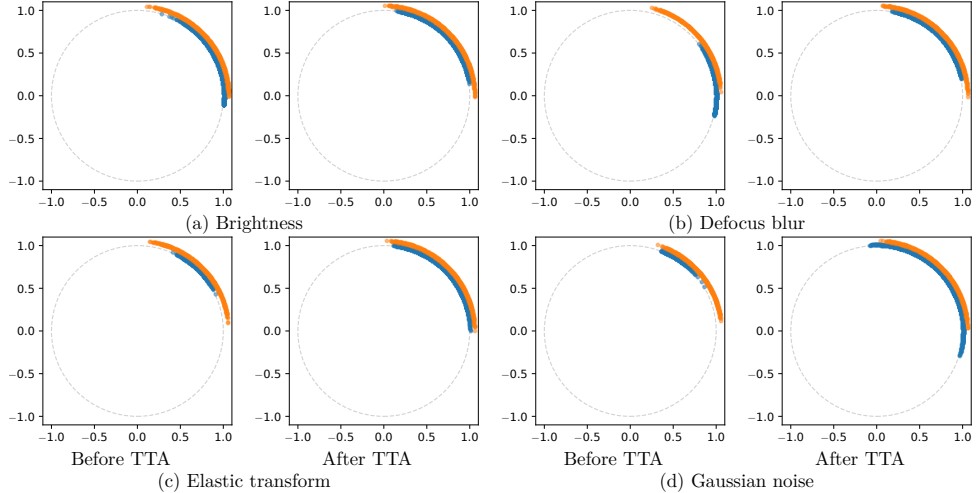

Figure 5: Spherical PCA (Liu et al., 2019) visualization of image (blue dots) and text (orange dots) embeddings before and after TTA with UnInfo on ImageNet-C. The image embeddings are distributed in a broader range of the circle after TTA, which suggests that the uniformity is improved. Moreover, the distribution of the image embeddings is aligned with the text embeddings, i.e., the image embeddings are more classification-friendly.

entropy quickly increases. However, the maximum value of the weight for entropy differs, and the weight for the uniformity loss is retained. This suggests the uniformity loss needs to be minimized along with entropy to retain information for these corruptions. On the other hand, the Gaussian noise in (d) showed a different evolution: the weight is larger for uniformity loss than for the entropy in the initial phase, indicating that retaining information of image embeddings is prioritized.

Next, we visualized embeddings to observe how the uniformity is improved. As the CLIP's embeddings are normalized and distributed on a unit hypersphere, we used the spherical PCA (Liu et al., 2019), which projects data points on a high-dimensional unit hypersphere onto a low-dimensional hypersphere (a 2D circle here). Fig. 5 visualizes the image embeddings before and after TTA with UnInfo on several corruptions of ImageNet-C, along with the text embeddings. After TTA, the image embeddings are distributed in a broader range of the circle than those before TTA on all corruptions, which suggests that the uniformity is improved. Moreover, the distributions of image and text embeddings are aligned after TTA, which suggests that the image embeddings become more classification-friendly. Uniformity improved most significantly in the Gaussian noise corruption in (d) because uniformity loss was prioritized by the information-aware loss balancing, as shown in Fig. 4 (d), as we intended. On the other hand, uniformity improved less significantly in the brightness corruption in (a) than in the other corruptions since the entropy is highly prioritized for the brightness corruption in Fig. 4 (a). In other words, uniformity is less important for this corruption

Table 7: Mean adaptation throughput and GPU memory usage.

| Method | Throughput (images/sec.) | GPU memory (MiB) |
|---|---|---|
| No-adapt | $299.6_{\pm 1.0}$ | 1348 |
| TPT (Shu et al., 2022) | $1.8_{\pm 0.0}$ | 14409 |
| ZERO (Farina et al., 2024) | $4.2_{\pm 0.0}$ | 2319 |
| MTA (Zanella & Ben Ayed, 2024b) | $1.3_{\pm 0.0}$ | 2331 |
| TDA (Karmanov et al., 2024) | $44.6_{\pm 2.4}$ | **1414** |
| DMN (Zhang et al., 2024b) | $5.7_{\pm 0.06}$ | 5760 |
| BATCLIP (Maharana et al., 2025) | $86.72_{\pm 0.3}$ | 17456 |
| Mint (Bao et al., 2025) | $\mathbf{161.16_{\pm 1.64}}$ | 3112 |
| UnInfo (ours) | $73.7_{\pm 0.5}$ | 11736 |

because the image embeddings are already spread in a broader range than for the other corruptions, and their distribution is already aligned with that of the text embeddings before TTA.

### 5.4.5 Computational Efficiency

Tab. 7 shows each method's throughput (images per second) and GPU memory usage (MiB). The baseline methods based on marginal confidence over augmented views of an input (TPT (Shu et al., 2022), ZERO (Farina et al., 2024), and MTA (Zanella & Ben Ayed, 2024b)) had very low throughput because they needed to run forward passes for 64 augmented views per image. TPT and MTA had the lowest throughput because they require a further backward pass and solve an optimization problem for each image, respectively. Especially, TPT had a high GPU memory usage because of backward pass for prompt optimization. In contrast, UnInfo had higher throughput than them since it does not require data augmentation and only requires one forward and backward pass per image, though it had higher GPU memory usage. TDA had a higher throughput than TPT, ZERO, MTA, and DMN because it does not rely on either backpropagation or augmented views, but UnInfo had a higher throughput than TDA. This is because UnInfo can leverage GPUs' parallelization by processing test samples in batches and utilizing LoRA, whereas UnInfo requires backpropagation. In contrast, TDA requires one-by-one data processing to update the cache, which hinders GPU acceleration. BATCLIP and Mint had higher throughput than UnInfo because they update layer normalization parameters, which have fewer parameters than LoRA. Mint had less GPU memory than UnInfo and BATCLIP due to the batch size. Although UnInfo's throughput is not higher than BATCLIP's or Mint's, its efficiency is also reasonable. Moreover, UnInfo can further speed up inference and save GPU memory as much as No-adapt because the knowledge of the test distribution is accumulated in the LoRA adapters; one may stop adaptation and merge LoRA to the stem model when the test distribution is stable. In contrast, TPT, ZERO, and MTA are episodic methods, i.e., they do not update the model or accumulate any information. Thus, they always have to perform adaptation and inference together, unlike UnInfo.

## 6 Related Work

### 6.1 Contrastive Language-image Pre-training

CLIP (Radford et al., 2021) is a multimodal foundation model training paradigm, especially between image and text modalities. In CLIP, two encoders (an image encoder and a text encoder), are trained to map image and text inputs into a unified embedding space so that semantically corresponding inputs are mapped to close embeddings and vice versa. Although the training strategy is simple, CLIP has demonstrated remarkable generalization on downstream tasks by being trained on a huge dataset (e.g., hundreds of millions of image-text pairs). However, CLIP degrades downstream performance when faced with datasets with a large gap from the training dataset (Zhang et al., 2022b; Huang et al., 2024; Chen et al., 2023; Shu et al., 2022; Zhou et al., 2024; Karmanov et al., 2024; Zhang et al., 2024b; Zanella & Ben Ayed, 2024b; Wang et al., 2024b; Qian & Hu, 2024). Re-training is often infeasible for adapting CLIP to a new dataset because it incurs a substantial computational cost, as described above. To address this challenge, TTA of CLIP has been actively studied.

## 6.2 Test-time Adaptation of Vision-language Models

To adapt instantly to test distributions without incurring high computational costs, TTA for zero-shot classification with CLIP has been studied. TTA aims to adapt a zero-shot CLIP classifier to the test distribution with only unlabeled test data. Existing CLIP TTA methods can be classified along two axes: episodic/cumulative and frozen/updating encoders.

**Episodic / frozen encoders.** The representative approach of CLIP TTA is to update the image and/or text embeddings. MTA (Zanella & Ben Ayed, 2024b) selects reliable image embeddings and updates the feature centroid. ZERO (Farina et al., 2024) dynamically correct the prediction probabilities for each test input.

**Cumulative / frozen encoders.** Test-time prompt tuning approaches (Shu et al., 2022; Yoon et al., 2024; Wang et al., 2025) updates the text token embeddings (e.g., four embedding vectors corresponding to words of a prompt template "`a photo of a`") during testing to minimize the prediction entropy marginalized with augmented views of an input image. The text embedding corresponding to each class is updated to be more appropriate to the current domain by updating the text token embeddings. Existing method also directly update the embeddings or logits computed from the similarity of image and text embeddings. TDA (Karmanov et al., 2024) and DMN (Zhang et al., 2024b) accumulate test inputs and construct the image embedding caches to modify subsequent inputs' predictions. OnZeta (Qian & Hu, 2024), DPE (Zhang et al., 2024a) and TPS (Sui et al., 2025) dynamically update class weights and/or visual prototypes. STS (Dafnis & Metaxas, 2026) updates text prototypes within a spectral subspace.

The above frozen encoder methods can adapt well to domain shifts, such as changes in environments, rendition, or out-of-distribution (Hendrycks et al., 2021a; Recht et al., 2019; Hendrycks et al., 2021b; Wang et al., 2019), using fixed image and text encoders. This is because a pre-trained CLIP generalizes to a wide range of domains enough to encode the current domain's semantics properly.

**Cumulative / updating encoders.** More recently, TTA methods for CLIP targeting image corruption have been developed (Bao et al., 2025; Maharana et al., 2025; A Vargas Hakim et al., 2025; Osowiechi et al., 2024). These methods update the image and text encoders to more effectively mitigate embedding collapse caused by image corruption. In particular, Mint (Bao et al., 2025) is similar to UnInfo in that it maintains inter-class variance in image embeddings, which is close to uniformity. While these methods are effective to image corruption, we found that some of them are prone to overfitting to the ImageNet-C corruption types (Sec. 5.4.1).

Our contributions include the finding that uniformity (Wang & Isola, 2020; Wang et al., 2023a) especially matters in the embedding space under image corruption (Sec. 3).

## 7 Conclusion

We proposed UnInfo, a novel test-time adaptation (TTA) method for zero-shot classification with vision-language models under image corruption. Unlike existing methods, UnInfo updates the image encoder to address the specific challenge of image corruption, where loss input information is retained in the image embeddings unlike other natural distribution shifts. In the experiments, UnInfo achieved higher classification performance than baselines by refining the uniformity along with the entropy, and the information-aware loss balancing further improved the performance. One limitation of UnInfo is that it requires test data to be mini-batched and sampled i.i.d. to compute the uniformity loss and information-aware balancing weight. Our future work is to extend UnInfo to a fully online setting and to broader types of realistic distribution shifts, e.g., real-world sensor shifts (Baek et al., 2024), mixed severity/corruption types and non-i.i.d. test streams, which have been studied recently (Yuan et al., 2023; Zanella et al., 2025).

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

**Algorithm 1** Procedure of UnInfo.

**Input:** Pre-trained image encoder $f_{\theta_{\text{img}}}^{\text{img}}$, class text embeddings $\{\mathbf{t}_c\}_{c=1}^C$, initialized LoRA $\phi_{\text{img}}$, target dataset $\mathcal{D}$

**Output:** Adapted LoRA parameter (EMA) $\bar{\phi}_{\text{img}}$

    **for** mini-batch $\{\mathbf{x}_i\}_{i=1}^B$ in $\mathcal{D}$ **do**

        Compute image embeddings with the current LoRA $\{\mathbf{z}_i = f_{\theta_{\text{img}}, \phi_{\text{img}}}^{\text{img}}(\mathbf{x}_i)\}_{i=1}^B$.

        Compute zero-shot prediction probabilities $\{\hat{p}_{i,c}\}_{i=1}^B$ in accordance with Eq. (1).

        Compute teacher zero-shot prediction probabilities $\{\hat{q}_{i,c}\}_{i=1}^B$ using the EMA LoRA $\bar{\phi}_{\text{img}}$.

        Compute the loss in accordance with Eq. (14).

        Update $\phi_{\text{img}}$.

        Update EMA teacher $\bar{\phi}_{\text{img}}$ in accordance with Eq. (12).

    **end for**

Table 8: Text prompt templates used for ensemble in the preliminary experiment (Sec. 3). These templates are proposed by Radford et al. (2021)[4].

| | | | |
|---|---|---|---|
| a bad photo of a {}. | a photo of many {}. | a sculpture of a {}. | a photo of the hard to see {}. |
| a low resolution photo of the {}. | a rendering of a {}. | graffiti of a {}. | a bad photo of the {}. |
| a cropped photo of the {}. | a tattoo of a {}. | the embroidered {}. | a photo of a hard to see {}. |
| a bright photo of a {}. | a photo of a clean {}. | a photo of a dirty {}. | a dark photo of the {}. |
| a drawing of a {}. | a photo of my {}. | the plastic {}. | a photo of the cool {}. |
| a close-up photo of a {}. | a black and white photo of the {}. | a painting of the {}. | a painting of a {}. |
| a pixelated photo of the {}. | a sculpture of the {}. | a bright photo of the {}. | a cropped photo of a {}. |
| a plastic {}. | a photo of the dirty {}. | a jpeg corrupted photo of a {}. | a blurry photo of the {}. |
| a photo of the {}. | a good photo of the {}. | a rendering of the {}. | a {} in a video game. |
| a photo of one {}. | a doodle of a {}. | a close-up photo of the {}. | a photo of a {}. |
| the origami {}. | the {} in a video game. | a sketch of a {}. | a doodle of the {}. |
| a origami {}. | a low resolution photo of a {}. | the toy {}. | a rendition of the {}. |
| a photo of the clean {}. | a photo of a large {}. | a rendition of a {}. | a photo of a nice {}. |
| a photo of a weird {}. | a blurry photo of a {}. | a cartoon {}. | art of a {}. |
| a sketch of the {}. | a embroidered {}. | a pixelated photo of a {}. | itap of the {}. |
| a jpeg corrupted photo of the {}. | a good photo of a {}. | a plushie {}. | a photo of the nice {}. |
| a photo of the small {}. | a photo of the weird {}. | the cartoon {}. | art of the {}. |
| a drawing of the {}. | a photo of the large {}. | a black and white photo of a {}. | the plushie {}. |
| a dark photo of a {}. | itap of a {}. | graffiti of the {}. | a toy {}. |
| itap of my {}. | a photo of a cool {}. | a photo of a small {}. | a tattoo of the {}. |

# A  Broader Impact Concerns

While UnInfo enables test-time adaptation for zero-shot VLM classification in a fully unsupervised manner, it may also adapt to biased data, which could affect model behavior, such as fairness. In sensitive applications, model behavior should be carefully monitored and appropriately intervened in as needed.

# B  Details of UnInfo

Algorithm 1 lists the procedure of UnInfo.

# C  Text Prompt Ensemble

Here, we describe the details of the prompt ensemble in the preliminary experiment (Sec. 3). We used template texts listed in Tab. 8 for the ensemble of multiple templates (denoted by "Ensemble" in Tab. 1). We generated the prompts using the templates for each class and encoded them with the text encoder. Then, we calculated the mean of the text embeddings. We normalized the embedding and used it for the class prototype.

For the ensemble of corruption synonyms (denoted by "Corruption prompt" in Tab. 1), we ensembled the corruption synonyms listed in Tab. 9, which are generated by GPT-4o (Hurst et al., 2024) with the instruction "`You are an expert of image processing. List the synonyms of the word "[corruption name]," which represents image quality.`"

Table 9: Synonyms of the corruption names of ImageNet-C used in the preliminary experiment in Sec. 3.

| Defocus blur | Glass blur | Motion blur | Zoom blur | Contrast |
|---|---|---|---|---|
| defocus blur | glass blur | motion blur | zoom blur | contrast |
| out-of-focus blur | frosted blur | directional blur | radial blur | tonal contrast |
| soft focus | glazing blur | linear blur | zooming effect | brightness difference |
| bokeh | diffuse blur | dynamic blur | dynamic zoom blur | clarity |
| lens blur | smudged blur | streaking | burst blur | definition |
| gaussian blur | hazy blur | trail blur | focus expansion blur | distinction |
| depth blur | translucent blur | speed blur | depth blur | sharpness |
| background blur | refractive blur | panning blur | lens zoom blur | intensity diffenrece |
| field blur | distortion blur | motion streak | outward motion blur | dynamic range |
| focus softness | veiled blur | kinetic blur | radian streak blur | separation |

| Elastic transform | Jpeg compression | Pixelate | Gaussian noise | Impulse noise |
|---|---|---|---|---|
| elastic transform | jpeg compression | pixelate | Gaussian noise | impulse noise |
| warping | image compression | blockify | normal noise | salt-and-pepper noise |
| distortion | lossy compression | rasterize | additive noise | spiky noise |
| deformation | JPEG encoding | mosaic | white Gaussian noise | outlier noise |
| stretching | file compression | chunkify | statistical noise | random noise |
| bending | quantization artifacting | grid effect | random noise | shot noise |
| geometric transform | data compression | quantization | luminance noise | transitional noise |
| morphing | image encoding | low-resolution effect | stochastic interference | burst noise |
| image warping | compression artifacts | bitmapping | signal perturbation | pulsed noise |
| spatial transform | JPEG artifacts | aliased effect | normal distribution noise | point noise |

| Shot noise | Brightness | Fog | Frost | Snow |
|---|---|---|---|---|
| shot noise | brightness | fog | frost | snow |
| photon noise | luminance | haze | frosting | noise |
| Poisson noise | illumination | mist | glare | grain |
| quantum noise | lightness | obscuration | haze | salt-and-pepper noise |
| statistical noise | intensity | cloudiness | mist | static |
| random noise | radiance | smog | veiling | visual noise |
| electronic noise | glow | blur | soft-focus | pixel noise |
| counting noise | shininess | glare | diffusion | random noise |
| current noise | exposure | veiling | cloudiness | white noise |
| flicker noise | highlighting | dimming | blur | dither |

# D  Earth Mover's Distance

In Sec. 3, we evaluated the modality gap between image and text embeddings using the earth mover's distance (EMD). Here, we describe the details of EMD (Villani, 2008; Peyré & Cuturi, 2019).

EMD is computed as the minimum cost of transporting image embeddings $\{\mathbf{z}_1, \ldots, \mathbf{z}_N\}$ to match text embeddings $\{\mathbf{t}_1, \ldots, \mathbf{t}_C\}$. More formally, we computed EMD between two distributions $p(\mathbf{z}) = (1/N) \sum_{n=1}^{N} \delta(\mathbf{z} - \mathbf{z}_n)$ and $q(\mathbf{z}) = (1/C) \sum_{c=1}^{C} \delta(\mathbf{z} - \mathbf{t}_c)$ over $\mathbb{R}^d$, where $\delta(\cdot)$ is the Dirac function. Specifically, since the embedding vectors are normalized and on the unit hypersphere, we adopted the spherical distance $d(\mathbf{z}_n, \mathbf{t}_c) := \arccos(\mathbf{z}_n^\top \mathbf{t}_c)$ for the transportation cost. We implemented the computation with the Python Optimal Transport (POT) library (Flamary et al., 2021; 2024).

# E  Additional Experimental Results

## E.1  Adaptation Performance

**Other CLIP architectures.**  Similar to Sec. 5.4.1, we conducted experiments with OpenAI ViT-B/32 CLIP[5] and SigLIP (Zhai et al., 2023). Tabs. 10 to 13 show the results. The trend aligns with the cases of the ViT-B/16 CLIP in Sec. 5.4.1. Similar to the results shown in Sec. 5.4.1, UnInfo consistently improved the performance compared to No-adapt. In contrast, although the average accuracies of Mint and BATCLIP are sometimes higher than UnInfo, they are also sometimes lower than No-adapt, suggesting that UnInfo is more stable across corruption types and model architectures.

---

[4]https://github.com/openai/CLIP/blob/main/notebooks/Prompt_Engineering_for_ImageNet.ipynb
[5]https://github.com/openai/CLIP

Table 10: Test accuracy (%) of OpenAI ViT-B/32 on ImageNet-C. The numbers (1), (5), and (10) presented with the method names are the shot numbers per class $n$ used for the few-shot adaptation methods.

| Method | Defocus blur | Glass blur | Motion blur | Zoom blur | Contrast | Elastic transform | Jpeg compression | Pixelate | Gaussian noise | Impulse noise | Shot noise | Brightness | Fog | Frost | Snow | Mean |
|---|---|---|---|---|---|---|---|---|---|---|---|---|---|---|---|---|
| No-adapt | 22.31 | 11.39 | 20.07 | 17.50 | 17.30 | 18.13 | 29.36 | 30.24 | 13.01 | 13.54 | 13.14 | 48.38 | 28.40 | 24.84 | 23.81 | 22.09 |
| Linear probing (1) | $7.34_{\pm0.16}$ | $3.98_{\pm0.04}$ | $6.93_{\pm0.05}$ | $6.25_{\pm0.09}$ | $5.60_{\pm0.02}$ | $7.53_{\pm0.33}$ | $10.39_{\pm0.07}$ | $11.38_{\pm0.31}$ | $5.05_{\pm0.03}$ | $5.03_{\pm0.09}$ | $5.24_{\pm0.06}$ | $20.45_{\pm0.30}$ | $10.80_{\pm0.35}$ | $8.10_{\pm0.13}$ | $7.59_{\pm0.10}$ | $8.11_{\pm0.07}$ |
| Linear probing (5) | $12.27_{\pm0.18}$ | $6.91_{\pm0.23}$ | $11.11_{\pm0.05}$ | $9.82_{\pm0.20}$ | $8.93_{\pm0.11}$ | $12.29_{\pm0.21}$ | $16.34_{\pm0.13}$ | $18.54_{\pm0.24}$ | $7.63_{\pm0.06}$ | $7.88_{\pm0.20}$ | $8.22_{\pm0.15}$ | $31.84_{\pm0.08}$ | $17.36_{\pm0.27}$ | $13.49_{\pm0.11}$ | $12.55_{\pm0.53}$ | $13.01_{\pm0.02}$ |
| Linear probing (10) | $14.52_{\pm0.03}$ | $8.41_{\pm0.20}$ | $13.08_{\pm0.11}$ | $11.73_{\pm0.21}$ | $10.55_{\pm0.28}$ | $15.38_{\pm0.54}$ | $19.38_{\pm0.24}$ | $22.34_{\pm0.20}$ | $9.55_{\pm0.36}$ | $9.44_{\pm0.29}$ | $9.72_{\pm0.14}$ | $37.52_{\pm0.32}$ | $20.71_{\pm0.37}$ | $16.38_{\pm0.14}$ | $15.21_{\pm0.28}$ | $15.59_{\pm0.08}$ |
| Tip-adapter (Zhang et al., 2022b) (1) | $10.40_{\pm0.16}$ | $5.67_{\pm0.10}$ | $9.49_{\pm0.08}$ | $8.94_{\pm0.07}$ | $7.45_{\pm0.03}$ | $11.22_{\pm0.14}$ | $15.07_{\pm0.15}$ | $16.36_{\pm0.24}$ | $6.53_{\pm0.16}$ | $6.73_{\pm0.10}$ | $6.97_{\pm0.17}$ | $29.89_{\pm0.18}$ | $15.36_{\pm0.20}$ | $11.98_{\pm0.17}$ | $10.97_{\pm0.26}$ | $11.53_{\pm0.07}$ |
| Tip-adapter (5) | $14.07_{\pm0.10}$ | $7.94_{\pm0.07}$ | $12.34_{\pm0.12}$ | $11.24_{\pm0.13}$ | $9.39_{\pm0.11}$ | $14.91_{\pm0.28}$ | $18.77_{\pm0.39}$ | $21.68_{\pm0.25}$ | $8.72_{\pm0.08}$ | $9.08_{\pm0.21}$ | $9.12_{\pm0.26}$ | $36.48_{\pm0.21}$ | $19.45_{\pm0.21}$ | $15.83_{\pm0.09}$ | $14.00_{\pm0.23}$ | $14.87_{\pm0.05}$ |
| Tip-adapter (10) | $16.14_{\pm0.25}$ | $9.40_{\pm0.27}$ | $14.30_{\pm0.18}$ | $12.69_{\pm0.29}$ | $11.23_{\pm0.02}$ | $17.21_{\pm0.26}$ | $21.25_{\pm0.20}$ | $24.76_{\pm0.18}$ | $9.92_{\pm0.04}$ | $10.47_{\pm0.12}$ | $10.48_{\pm0.25}$ | $40.20_{\pm0.28}$ | $22.15_{\pm0.23}$ | $18.04_{\pm0.20}$ | $15.93_{\pm0.05}$ | $16.94_{\pm0.02}$ |
| TPT (Shu et al., 2022) | $23.29_{\pm0.06}$ | $12.16_{\pm0.03}$ | $21.01_{\pm0.03}$ | $19.62_{\pm0.04}$ | $18.62_{\pm0.02}$ | $19.50_{\pm0.08}$ | $31.74_{\pm0.09}$ | $32.57_{\pm0.08}$ | $12.81_{\pm0.05}$ | $13.24_{\pm0.02}$ | $13.13_{\pm0.03}$ | $50.47_{\pm0.03}$ | $30.06_{\pm0.04}$ | $26.89_{\pm0.09}$ | $25.68_{\pm0.02}$ | $23.39_{\pm0.02}$ |
| ZERO (Farina et al., 2024) | $21.48_{\pm0.04}$ | $10.17_{\pm0.03}$ | $18.93_{\pm0.02}$ | $20.12_{\pm0.08}$ | $14.33_{\pm0.03}$ | $17.27_{\pm0.03}$ | $30.04_{\pm0.05}$ | $31.22_{\pm0.04}$ | $9.19_{\pm0.05}$ | $9.80_{\pm0.01}$ | $9.31_{\pm0.06}$ | $48.44_{\pm0.07}$ | $29.21_{\pm0.03}$ | $25.33_{\pm0.08}$ | $25.02_{\pm0.03}$ | $22.33_{\pm0.00}$ |
| MTA (Zanella & Ben Ayed, 2024b) | $20.76_{\pm0.05}$ | $11.19_{\pm0.06}$ | $18.60_{\pm0.06}$ | $18.56_{\pm0.04}$ | $19.42_{\pm0.07}$ | $19.29_{\pm0.03}$ | $30.25_{\pm0.11}$ | $32.09_{\pm0.05}$ | $10.86_{\pm0.01}$ | $11.40_{\pm0.04}$ | $11.43_{\pm0.02}$ | $49.33_{\pm0.06}$ | $29.33_{\pm0.06}$ | $26.22_{\pm0.05}$ | $24.65_{\pm0.08}$ | $22.23_{\pm0.02}$ |
| TDA (Karmanov et al., 2024) | $23.93_{\pm0.08}$ | $12.97_{\pm0.06}$ | $22.29_{\pm0.03}$ | $19.49_{\pm0.04}$ | $18.22_{\pm0.05}$ | $21.10_{\pm0.08}$ | $31.41_{\pm0.02}$ | $32.89_{\pm0.06}$ | $14.65_{\pm0.04}$ | $15.32_{\pm0.02}$ | $15.24_{\pm0.05}$ | $50.49_{\pm0.04}$ | $31.66_{\pm0.13}$ | $27.33_{\pm0.03}$ | $26.02_{\pm0.14}$ | $24.20_{\pm0.02}$ |
| DMN (Zhang et al., 2024b) | $23.30_{\pm0.00}$ | $12.02_{\pm0.00}$ | $21.43_{\pm0.00}$ | $18.85_{\pm0.00}$ | $16.41_{\pm0.00}$ | $18.93_{\pm0.00}$ | $30.75_{\pm0.00}$ | $31.67_{\pm0.00}$ | $13.83_{\pm0.00}$ | $14.53_{\pm0.00}$ | $14.13_{\pm0.00}$ | $50.84_{\pm0.00}$ | $29.28_{\pm0.00}$ | $26.24_{\pm0.00}$ | $24.89_{\pm0.00}$ | $23.14_{\pm0.00}$ |
| Mint (Bao et al., 2025) | $19.89_{\pm0.16}$ | $\mathbf{20.67_{\pm0.08}}$ | $23.57_{\pm0.17}$ | $\mathbf{21.59_{\pm0.13}}$ | $15.99_{\pm0.03}$ | $\mathbf{25.99_{\pm0.09}}$ | $\mathbf{36.56_{\pm0.04}}$ | $\mathbf{36.88_{\pm0.09}}$ | $\mathbf{19.39_{\pm0.10}}$ | $\mathbf{19.00_{\pm0.12}}$ | $\mathbf{19.88_{\pm0.11}}$ | $45.06_{\pm0.16}$ | $\mathbf{36.02_{\pm0.11}}$ | $21.18_{\pm0.06}$ | $22.88_{\pm0.10}$ | $25.64_{\pm0.03}$ |
| BATCLIP (Maharana et al., 2025) | $6.47_{\pm0.19}$ | $9.42_{\pm0.61}$ | $18.08_{\pm0.72}$ | $19.42_{\pm0.43}$ | $3.74_{\pm0.36}$ | $24.24_{\pm0.32}$ | $28.11_{\pm1.36}$ | $26.54_{\pm0.62}$ | $9.39_{\pm0.05}$ | $8.18_{\pm0.39}$ | $8.77_{\pm0.09}$ | $43.25_{\pm0.33}$ | $29.35_{\pm0.09}$ | $14.91_{\pm0.49}$ | $20.49_{\pm0.21}$ | $18.03_{\pm0.25}$ |
| UnInfo (ours) | $\mathbf{26.80_{\pm0.28}}$ | $14.77_{\pm1.00}$ | $\mathbf{26.10_{\pm0.74}}$ | $21.41_{\pm0.57}$ | $\mathbf{21.51_{\pm0.27}}$ | $20.94_{\pm0.71}$ | $34.73_{\pm0.10}$ | $36.87_{\pm0.50}$ | $16.71_{\pm0.15}$ | $18.88_{\pm0.64}$ | $17.14_{\pm0.47}$ | $\mathbf{51.16_{\pm0.25}}$ | $33.09_{\pm0.44}$ | $\mathbf{27.44_{\pm0.13}}$ | $\mathbf{27.39_{\pm0.34}}$ | $\mathbf{26.33_{\pm0.04}}$ |

Table 11: Test accuracy (%) of SigLIP on ImageNet-C. The numbers (1), (5), and (10) presented with the method names are the shot numbers per class $n$ used for the few-shot adaptation methods.

| Method | Defocus blur | Glass blur | Motion blur | Zoom blur | Contrast | Elastic transform | Jpeg compression | Pixelate | Gaussian noise | Impulse noise | Shot noise | Brightness | Fog | Frost | Snow | Mean |
|---|---|---|---|---|---|---|---|---|---|---|---|---|---|---|---|---|
| No-adapt | 26.49 | 12.33 | 16.60 | 20.38 | 16.48 | 16.09 | 36.61 | 45.29 | 6.18 | 8.09 | 8.03 | 62.21 | 42.49 | 32.75 | 31.94 | 25.46 |
| Linear probing (1) | $9.79_{\pm1.48}$ | $5.80_{\pm0.10}$ | $6.56_{\pm1.32}$ | $9.75_{\pm0.23}$ | $6.76_{\pm0.26}$ | $8.33_{\pm0.20}$ | $14.46_{\pm0.24}$ | $20.78_{\pm0.41}$ | $3.13_{\pm0.29}$ | $3.23_{\pm0.03}$ | $3.34_{\pm0.14}$ | $31.11_{\pm2.40}$ | $19.56_{\pm0.34}$ | $10.68_{\pm0.23}$ | $13.48_{\pm0.38}$ | $11.12_{\pm0.19}$ |
| Linear probing (5) | $18.15_{\pm0.34}$ | $9.83_{\pm0.17}$ | $12.28_{\pm0.17}$ | $15.72_{\pm0.28}$ | $10.50_{\pm0.18}$ | $14.17_{\pm0.21}$ | $23.23_{\pm0.08}$ | $32.55_{\pm0.03}$ | $1.61_{\pm2.13}$ | $0.10_{\pm0.00}$ | $2.26_{\pm3.06}$ | $49.34_{\pm0.13}$ | $30.55_{\pm0.26}$ | $18.31_{\pm0.38}$ | $21.20_{\pm0.35}$ | $17.32_{\pm0.21}$ |
| Linear probing (10) | $21.64_{\pm0.05}$ | $12.08_{\pm0.12}$ | $14.54_{\pm0.23}$ | $18.97_{\pm0.28}$ | $12.62_{\pm0.26}$ | $17.40_{\pm0.36}$ | $27.96_{\pm0.15}$ | $37.78_{\pm0.37}$ | $0.10_{\pm0.00}$ | $0.10_{\pm0.00}$ | $2.85_{\pm3.88}$ | $55.23_{\pm0.10}$ | $35.82_{\pm0.43}$ | $22.93_{\pm0.23}$ | $24.72_{\pm0.04}$ | $20.31_{\pm0.29}$ |
| Tip-adapter (Zhang et al., 2022b) (1) | $15.16_{\pm0.24}$ | $7.86_{\pm0.17}$ | $10.10_{\pm0.27}$ | $13.73_{\pm0.24}$ | $9.25_{\pm0.19}$ | $11.93_{\pm0.15}$ | $21.88_{\pm0.23}$ | $29.03_{\pm0.37}$ | $4.49_{\pm0.07}$ | $5.51_{\pm0.13}$ | $5.53_{\pm0.09}$ | $44.67_{\pm0.12}$ | $27.45_{\pm0.41}$ | $16.71_{\pm0.20}$ | $18.62_{\pm0.11}$ | $16.13_{\pm0.10}$ |
| Tip-adapter (5) | $20.04_{\pm0.13}$ | $10.99_{\pm0.16}$ | $13.81_{\pm0.18}$ | $17.90_{\pm0.17}$ | $11.88_{\pm0.06}$ | $16.26_{\pm0.16}$ | $26.36_{\pm0.24}$ | $36.26_{\pm0.41}$ | $5.72_{\pm0.18}$ | $7.01_{\pm0.04}$ | $6.95_{\pm0.07}$ | $52.24_{\pm0.12}$ | $33.39_{\pm0.20}$ | $20.28_{\pm0.46}$ | $23.22_{\pm0.10}$ | $20.15_{\pm0.02}$ |
| Tip-adapter (10) | $22.86_{\pm0.14}$ | $12.91_{\pm0.14}$ | $15.80_{\pm0.28}$ | $20.27_{\pm0.22}$ | $13.44_{\pm0.14}$ | $18.60_{\pm0.08}$ | $29.41_{\pm0.20}$ | $39.62_{\pm0.15}$ | $6.47_{\pm0.08}$ | $7.88_{\pm0.18}$ | $8.06_{\pm0.16}$ | $55.65_{\pm0.16}$ | $36.67_{\pm0.29}$ | $23.37_{\pm0.16}$ | $25.94_{\pm0.19}$ | $22.46_{\pm0.05}$ |
| TPT (Shu et al., 2022) | $11.32_{\pm0.01}$ | $4.87_{\pm0.00}$ | $6.56_{\pm0.02}$ | $8.94_{\pm0.01}$ | $22.77_{\pm0.02}$ | $2.68_{\pm0.01}$ | $9.08_{\pm0.01}$ | $18.02_{\pm0.01}$ | $0.90_{\pm0.00}$ | $0.97_{\pm0.00}$ | $0.88_{\pm0.00}$ | $2.81_{\pm0.02}$ | $33.48_{\pm0.02}$ | $3.44_{\pm0.01}$ | $0.44_{\pm0.00}$ | $8.48_{\pm0.00}$ |
| ZERO (Farina et al., 2024) | $10.20_{\pm0.00}$ | $3.74_{\pm0.05}$ | $5.63_{\pm0.01}$ | $9.09_{\pm0.03}$ | $21.11_{\pm0.04}$ | $3.65_{\pm0.00}$ | $8.10_{\pm0.03}$ | $19.41_{\pm0.02}$ | $0.81_{\pm0.01}$ | $0.89_{\pm0.02}$ | $0.69_{\pm0.01}$ | $2.65_{\pm0.02}$ | $36.91_{\pm0.06}$ | $3.86_{\pm0.02}$ | $0.36_{\pm0.00}$ | $8.47_{\pm0.01}$ |
| MTA (Zanella & Ben Ayed, 2024b) | $11.35_{\pm0.05}$ | $5.04_{\pm0.02}$ | $6.27_{\pm0.03}$ | $9.04_{\pm0.06}$ | $\mathbf{26.00_{\pm0.06}}$ | $4.33_{\pm0.01}$ | $11.18_{\pm0.03}$ | $21.69_{\pm0.02}$ | $1.24_{\pm0.00}$ | $1.26_{\pm0.02}$ | $1.13_{\pm0.01}$ | $5.10_{\pm0.02}$ | $38.29_{\pm0.07}$ | $4.93_{\pm0.03}$ | $0.70_{\pm0.01}$ | $9.84_{\pm0.01}$ |
| TDA (Karmanov et al., 2024) | $29.07_{\pm0.02}$ | $14.75_{\pm0.05}$ | $19.09_{\pm0.02}$ | $23.57_{\pm0.02}$ | $18.48_{\pm0.03}$ | $19.95_{\pm0.04}$ | $38.35_{\pm0.06}$ | $47.45_{\pm0.02}$ | $7.87_{\pm0.04}$ | $9.76_{\pm0.01}$ | $9.71_{\pm0.03}$ | $64.13_{\pm0.05}$ | $45.62_{\pm0.10}$ | $35.45_{\pm0.05}$ | $34.81_{\pm0.11}$ | $27.87_{\pm0.01}$ |
| DMN (Zhang et al., 2024b) | $13.31_{\pm0.00}$ | $5.92_{\pm0.00}$ | $7.36_{\pm0.00}$ | $9.48_{\pm0.00}$ | $27.58_{\pm0.00}$ | $3.78_{\pm0.00}$ | $10.75_{\pm0.00}$ | $21.66_{\pm0.00}$ | $0.90_{\pm0.00}$ | $0.92_{\pm0.00}$ | $0.77_{\pm0.00}$ | $2.90_{\pm0.00}$ | $41.07_{\pm0.00}$ | $4.39_{\pm0.00}$ | $0.36_{\pm0.00}$ | $10.08_{\pm0.00}$ |
| Mint (Bao et al., 2025) | $21.41_{\pm0.06}$ | $\mathbf{24.13_{\pm0.07}}$ | $19.50_{\pm0.41}$ | $\mathbf{26.06_{\pm0.10}}$ | $19.19_{\pm0.21}$ | $28.46_{\pm0.10}$ | $39.42_{\pm0.03}$ | $46.80_{\pm0.13}$ | $15.56_{\pm0.05}$ | $\mathbf{19.26_{\pm0.11}}$ | $15.13_{\pm0.10}$ | $63.29_{\pm0.10}$ | $47.13_{\pm0.08}$ | $34.29_{\pm0.12}$ | $\mathbf{39.43_{\pm0.05}}$ | $\mathbf{30.19_{\pm0.02}}$ |
| BATCLIP (Maharana et al., 2025) | $19.44_{\pm1.60}$ | $18.85_{\pm0.32}$ | $\mathbf{22.28_{\pm0.16}}$ | $11.78_{\pm3.17}$ | $4.67_{\pm0.17}$ | $\mathbf{33.33_{\pm0.17}}$ | $33.97_{\pm0.08}$ | $37.88_{\pm0.28}$ | $\mathbf{16.80_{\pm0.06}}$ | $17.33_{\pm0.16}$ | $\mathbf{17.98_{\pm0.15}}$ | $59.99_{\pm0.09}$ | $42.60_{\pm0.32}$ | $11.07_{\pm0.31}$ | $35.82_{\pm0.07}$ | $25.59_{\pm0.31}$ |
| UnInfo (ours) | $\mathbf{30.82_{\pm0.05}}$ | $15.28_{\pm0.37}$ | $20.41_{\pm0.38}$ | $24.48_{\pm0.46}$ | $20.80_{\pm0.43}$ | $19.89_{\pm0.14}$ | $\mathbf{41.73_{\pm0.30}}$ | $\mathbf{49.98_{\pm0.13}}$ | $9.72_{\pm0.13}$ | $11.41_{\pm0.15}$ | $9.09_{\pm2.74}$ | $\mathbf{64.96_{\pm0.23}}$ | $\mathbf{48.22_{\pm0.19}}$ | $\mathbf{36.73_{\pm0.23}}$ | $37.11_{\pm0.30}$ | $29.38_{\pm0.26}$ |

**CIFAR-C datasets.** We conducted experiments on CIFAR10/100-C (Hendrycks & Dietterich, 2019), which are the corrupted versions of CIFAR10/100 (Krizhevsky, 2009). Tabs. 14 to 16 and Tabs. 17 to 19 show the results on CIFAR10-C and CIFAR100-C, respectively. Although Mint and BATCLIP achieved higher accuracy in most cases, UnInfo also improved accuracy compared to the other baselines, suggesting the efficacy against image corruption.

**Domain Shifts.** We also tested TTA baselines and our UnInfo on domain shift datasets: ImageNet (Deng et al., 2009), ImageNet-A (Hendrycks et al., 2021b), and R (Hendrycks et al., 2021a). Tab. 20 shows the results. Although UnInfo did not outperform the existing TTA baselines since they are specifically designed

Table 12: Test accuracy (%) of OpenAI ViT-B/32 on ImageNet-C-bar. The numbers (1), (5), and (10) presented with the method names are the shot numbers per class $n$ used for the few-shot adaptation methods.

| Method | Blue noise sample | Brownish noise | Caustic refraction | Checkerboard cutout | Cocentric sine waves | Inverse sparkles | Perlin noise | Plasma noise | Single frequency greyscale | Sparkles | Mean |
|---|---|---|---|---|---|---|---|---|---|---|---|
| No-adapt | 29.44 | 40.46 | 32.98 | 35.76 | 10.59 | 14.09 | 44.83 | 16.58 | 26.53 | 44.70 | 29.60 |
| Linear probing (1) | $11.85_{\pm0.40}$ | $17.61_{\pm0.17}$ | $12.22_{\pm0.16}$ | $14.22_{\pm0.02}$ | $3.17_{\pm0.15}$ | $4.76_{\pm0.04}$ | $17.89_{\pm0.39}$ | $6.70_{\pm0.28}$ | $6.94_{\pm0.17}$ | $19.51_{\pm0.47}$ | $11.49_{\pm0.04}$ |
| Linear probing (5) | $18.91_{\pm0.24}$ | $26.94_{\pm0.11}$ | $19.19_{\pm0.05}$ | $23.10_{\pm0.10}$ | $5.08_{\pm0.17}$ | $7.71_{\pm0.29}$ | $27.78_{\pm0.10}$ | $9.97_{\pm0.24}$ | $10.80_{\pm0.26}$ | $30.39_{\pm0.17}$ | $17.99_{\pm0.03}$ |
| Linear probing (10) | $22.96_{\pm0.33}$ | $30.90_{\pm0.17}$ | $23.29_{\pm0.29}$ | $27.44_{\pm0.03}$ | $6.63_{\pm0.05}$ | $9.32_{\pm0.09}$ | $33.01_{\pm0.41}$ | $11.52_{\pm0.18}$ | $13.51_{\pm0.31}$ | $35.42_{\pm0.57}$ | $21.40_{\pm0.04}$ |
| Tip-adapter (Zhang et al., 2022b) (1) | $17.16_{\pm0.37}$ | $24.37_{\pm0.08}$ | $18.03_{\pm0.17}$ | $20.65_{\pm0.30}$ | $4.94_{\pm0.17}$ | $6.83_{\pm0.10}$ | $26.05_{\pm0.17}$ | $9.00_{\pm0.16}$ | $11.84_{\pm0.11}$ | $27.31_{\pm0.30}$ | $16.62_{\pm0.03}$ |
| Tip-adapter (5) | $21.75_{\pm0.21}$ | $30.07_{\pm0.17}$ | $22.59_{\pm0.20}$ | $26.24_{\pm0.50}$ | $6.02_{\pm0.11}$ | $8.96_{\pm0.15}$ | $32.16_{\pm0.26}$ | $11.05_{\pm0.03}$ | $13.95_{\pm0.21}$ | $33.69_{\pm0.31}$ | $20.65_{\pm0.08}$ |
| Tip-adapter (10) | $24.19_{\pm0.35}$ | $32.85_{\pm0.09}$ | $25.15_{\pm0.36}$ | $29.52_{\pm0.19}$ | $7.10_{\pm0.02}$ | $10.42_{\pm0.04}$ | $35.13_{\pm0.11}$ | $12.59_{\pm0.14}$ | $16.01_{\pm0.31}$ | $36.78_{\pm0.23}$ | $22.97_{\pm0.06}$ |
| TPT (Shu et al., 2022) | $30.43_{\pm0.09}$ | $42.77_{\pm0.04}$ | $35.87_{\pm0.03}$ | $38.12_{\pm0.04}$ | $11.32_{\pm0.04}$ | $16.35_{\pm0.05}$ | $46.75_{\pm0.04}$ | $18.03_{\pm0.04}$ | $28.87_{\pm0.09}$ | $47.69_{\pm0.04}$ | $31.62_{\pm0.02}$ |
| ZERO (Farina et al., 2024) | $23.87_{\pm0.02}$ | $40.90_{\pm0.06}$ | $\mathbf{36.66_{\pm0.07}}$ | $39.51_{\pm0.04}$ | $9.68_{\pm0.08}$ | $\mathbf{19.63_{\pm0.01}}$ | $44.81_{\pm0.08}$ | $17.72_{\pm0.04}$ | $28.03_{\pm0.09}$ | $46.27_{\pm0.02}$ | $30.71_{\pm0.02}$ |
| MTA (Zanella & Ben Ayed, 2024b) | $28.94_{\pm0.05}$ | $40.85_{\pm0.01}$ | $35.61_{\pm0.01}$ | $37.84_{\pm0.05}$ | $11.17_{\pm0.03}$ | $16.68_{\pm0.01}$ | $45.65_{\pm0.03}$ | $16.61_{\pm0.02}$ | $29.65_{\pm0.06}$ | $46.46_{\pm0.02}$ | $30.95_{\pm0.01}$ |
| TDA (Karmanov et al., 2024) | $33.49_{\pm0.12}$ | $43.47_{\pm0.07}$ | $35.45_{\pm0.04}$ | $39.97_{\pm0.06}$ | $12.42_{\pm0.05}$ | $15.61_{\pm0.06}$ | $46.94_{\pm0.05}$ | $18.96_{\pm0.07}$ | $27.94_{\pm0.07}$ | $\mathbf{48.50_{\pm0.06}}$ | $32.27_{\pm0.01}$ |
| DMN (Zhang et al., 2024b) | $30.85_{\pm0.00}$ | $42.99_{\pm0.00}$ | $34.55_{\pm0.00}$ | $37.23_{\pm0.00}$ | $11.03_{\pm0.00}$ | $14.94_{\pm0.00}$ | $46.81_{\pm0.00}$ | $17.75_{\pm0.00}$ | $27.99_{\pm0.00}$ | $47.35_{\pm0.00}$ | $31.15_{\pm0.00}$ |
| Mint (Bao et al., 2025) | $34.85_{\pm0.25}$ | $42.73_{\pm0.05}$ | $29.31_{\pm0.07}$ | $39.76_{\pm0.09}$ | $\mathbf{14.05_{\pm0.07}}$ | $14.36_{\pm0.10}$ | $45.23_{\pm0.08}$ | $\mathbf{24.07_{\pm0.07}}$ | $27.43_{\pm0.08}$ | $46.48_{\pm0.11}$ | $31.83_{\pm0.04}$ |
| BATCLIP (Maharana et al., 2025) | $31.13_{\pm0.05}$ | $36.48_{\pm0.06}$ | $25.79_{\pm0.18}$ | $31.04_{\pm0.07}$ | $9.43_{\pm0.51}$ | $11.51_{\pm0.19}$ | $39.30_{\pm0.30}$ | $3.70_{\pm0.43}$ | $17.83_{\pm0.25}$ | $38.68_{\pm0.19}$ | $24.49_{\pm0.19}$ |
| UnInfo (ours) | $\mathbf{35.88_{\pm0.31}}$ | $\mathbf{43.86_{\pm0.17}}$ | $36.34_{\pm0.31}$ | $\mathbf{40.03_{\pm0.21}}$ | $12.34_{\pm0.10}$ | $17.09_{\pm0.30}$ | $\mathbf{48.46_{\pm0.17}}$ | $18.82_{\pm0.21}$ | $\mathbf{29.67_{\pm0.19}}$ | $48.03_{\pm0.30}$ | $\mathbf{33.05_{\pm0.09}}$ |

Table 13: Test accuracy (%) of SigLIP on ImageNet-C-bar. The numbers (1), (5), and (10) presented with the method names are the shot numbers per class $n$ used for the few-shot adaptation methods.

| Method | Blue noise sample | Brownish noise | Caustic refraction | Checkerboard cutout | Cocentric sine waves | Inverse sparkles | Perlin noise | Plasma noise | Single frequency greyscale | Sparkles | Mean |
|---|---|---|---|---|---|---|---|---|---|---|---|
| No-adapt | 32.20 | 55.95 | 46.04 | 56.68 | 13.33 | 24.72 | 59.48 | 27.46 | 21.30 | 60.34 | 39.75 |
| Linear probing (1) | $14.39_{\pm0.18}$ | $28.10_{\pm0.35}$ | $20.19_{\pm0.20}$ | $26.71_{\pm2.33}$ | $4.46_{\pm0.10}$ | $9.99_{\pm0.37}$ | $29.67_{\pm0.14}$ | $10.57_{\pm0.26}$ | $6.95_{\pm0.34}$ | $30.44_{\pm2.83}$ | $18.15_{\pm0.50}$ |
| Linear probing (5) | $22.61_{\pm0.10}$ | $43.51_{\pm0.08}$ | $32.23_{\pm0.31}$ | $43.69_{\pm0.23}$ | $7.74_{\pm0.11}$ | $16.64_{\pm0.42}$ | $45.60_{\pm0.23}$ | $17.11_{\pm0.18}$ | $11.80_{\pm0.28}$ | $47.98_{\pm0.16}$ | $28.89_{\pm0.03}$ |
| Linear probing (10) | $26.64_{\pm0.40}$ | $49.88_{\pm0.33}$ | $38.19_{\pm0.18}$ | $49.51_{\pm0.16}$ | $9.89_{\pm0.10}$ | $19.82_{\pm0.20}$ | $51.88_{\pm0.30}$ | $20.63_{\pm0.40}$ | $15.22_{\pm0.25}$ | $54.09_{\pm0.12}$ | $33.57_{\pm0.01}$ |
| Tip-adapter (Zhang et al., 2022b) (1) | $19.91_{\pm0.25}$ | $39.06_{\pm0.15}$ | $29.38_{\pm0.08}$ | $39.06_{\pm0.14}$ | $6.93_{\pm0.21}$ | $14.89_{\pm0.30}$ | $41.90_{\pm0.07}$ | $15.63_{\pm0.27}$ | $11.58_{\pm0.20}$ | $42.54_{\pm0.27}$ | $26.09_{\pm0.03}$ |
| Tip-adapter (5) | $25.00_{\pm0.31}$ | $46.29_{\pm0.10}$ | $35.35_{\pm0.23}$ | $46.78_{\pm0.37}$ | $8.57_{\pm0.09}$ | $18.50_{\pm0.22}$ | $48.91_{\pm0.28}$ | $18.72_{\pm0.37}$ | $13.56_{\pm0.08}$ | $50.05_{\pm0.11}$ | $31.17_{\pm0.06}$ |
| Tip-adapter (10) | $27.78_{\pm0.14}$ | $49.58_{\pm0.26}$ | $38.62_{\pm0.47}$ | $50.29_{\pm0.21}$ | $10.08_{\pm0.04}$ | $20.43_{\pm0.14}$ | $52.48_{\pm0.18}$ | $20.86_{\pm0.23}$ | $15.78_{\pm0.13}$ | $53.58_{\pm0.19}$ | $33.95_{\pm0.04}$ |
| TPT (Shu et al., 2022) | $4.86_{\pm0.00}$ | $17.57_{\pm0.00}$ | $16.77_{\pm0.01}$ | $22.19_{\pm0.02}$ | $2.40_{\pm0.01}$ | $0.66_{\pm0.00}$ | $37.78_{\pm0.01}$ | $8.47_{\pm0.01}$ | $2.23_{\pm0.00}$ | $31.88_{\pm0.02}$ | $14.48_{\pm0.00}$ |
| ZERO (Farina et al., 2024) | $6.32_{\pm0.03}$ | $19.31_{\pm0.01}$ | $22.32_{\pm0.02}$ | $27.12_{\pm0.03}$ | $2.09_{\pm0.01}$ | $4.64_{\pm0.02}$ | $43.11_{\pm0.09}$ | $8.54_{\pm0.01}$ | $5.53_{\pm0.02}$ | $35.37_{\pm0.06}$ | $17.44_{\pm0.00}$ |
| MTA Zanella & Ben Ayed (2024b) | $8.42_{\pm0.01}$ | $20.05_{\pm0.07}$ | $21.69_{\pm0.03}$ | $27.69_{\pm0.05}$ | $2.29_{\pm0.02}$ | $2.12_{\pm0.00}$ | $44.93_{\pm0.07}$ | $8.62_{\pm0.01}$ | $6.92_{\pm0.04}$ | $36.89_{\pm0.04}$ | $17.96_{\pm0.02}$ |
| TDA (Karmanov et al., 2024) | $35.11_{\pm0.04}$ | $59.13_{\pm0.11}$ | $48.80_{\pm0.05}$ | $58.94_{\pm0.05}$ | $16.30_{\pm0.06}$ | $28.05_{\pm0.02}$ | $61.59_{\pm0.06}$ | $30.90_{\pm0.05}$ | $23.55_{\pm0.08}$ | $63.03_{\pm0.01}$ | $42.54_{\pm0.02}$ |
| DMN (Zhang et al., 2024b) | $5.85_{\pm0.00}$ | $20.02_{\pm0.00}$ | $19.82_{\pm0.00}$ | $27.43_{\pm0.00}$ | $2.29_{\pm0.00}$ | $0.64_{\pm0.00}$ | $46.16_{\pm0.00}$ | $9.36_{\pm0.00}$ | $2.39_{\pm0.00}$ | $37.91_{\pm0.00}$ | $17.19_{\pm0.00}$ |
| Mint (Bao et al., 2025) | $\mathbf{44.16_{\pm0.05}}$ | $59.33_{\pm0.02}$ | $45.66_{\pm0.13}$ | $59.43_{\pm0.12}$ | $\mathbf{20.53_{\pm0.19}}$ | $27.13_{\pm0.10}$ | $59.74_{\pm0.10}$ | $\mathbf{33.77_{\pm0.10}}$ | $\mathbf{31.82_{\pm0.08}}$ | $\mathbf{63.92_{\pm0.04}}$ | $\mathbf{44.55_{\pm0.04}}$ |
| BATCLIP (Maharana et al., 2025) | $36.87_{\pm0.27}$ | $56.55_{\pm0.06}$ | $45.37_{\pm0.03}$ | $55.64_{\pm0.13}$ | $2.73_{\pm0.11}$ | $3.18_{\pm0.08}$ | $57.18_{\pm0.16}$ | $8.48_{\pm0.07}$ | $6.21_{\pm0.49}$ | $58.93_{\pm0.04}$ | $33.12_{\pm0.07}$ |
| UnInfo (ours) | $38.76_{\pm0.19}$ | $\mathbf{60.79_{\pm0.18}}$ | $\mathbf{50.93_{\pm0.30}}$ | $\mathbf{60.92_{\pm0.04}}$ | $15.74_{\pm2.17}$ | $\mathbf{28.82_{\pm0.60}}$ | $\mathbf{63.03_{\pm0.13}}$ | $31.70_{\pm0.27}$ | $25.57_{\pm0.12}$ | $63.57_{\pm0.53}$ | $43.98_{\pm0.15}$ |

Table 14: Test accuracy (%) of OpenCLIP ViT-B/16 on CIFAR10-C. The numbers (1), (5), and (10) presented with the method names are the shot numbers per class $n$ used for the few-shot adaptation methods.

| Method | Defocus blur | Glass blur | Motion blur | Zoom blur | Contrast | Elastic transform | Jpeg compression | Pixelate | Gaussian noise | Impulse noise | Shot noise | Brightness | Fog | Frost | Snow | Mean |
|---|---|---|---|---|---|---|---|---|---|---|---|---|---|---|---|---|
| No-adapt | 80.68 | 43.23 | 73.91 | 82.09 | 69.40 | 64.51 | 62.81 | 51.05 | 47.96 | 60.98 | 53.50 | 90.82 | 81.27 | 83.88 | 83.09 | 68.61 |
| Linear probing (1) | 53.69 | 26.94 | 51.24 | 57.45 | 42.31 | 36.93 | 38.12 | 32.38 | 24.32 | 39.26 | 29.25 | 61.41 | 49.18 | 52.74 | 50.62 | 43.06 |
| Linear probing (5) | 74.12 | 40.37 | 65.64 | 76.32 | 55.91 | 54.34 | 52.03 | 45.85 | 40.75 | 51.52 | 44.11 | 84.77 | 69.70 | 73.14 | 73.54 | 60.14 |
| Linear probing (10) | 78.78 | 47.01 | 71.92 | 81.04 | 62.99 | 61.28 | 58.08 | 54.03 | 45.24 | 57.00 | 50.13 | 89.39 | 75.67 | 78.23 | 79.83 | 66.04 |
| Tip-adapter (Zhang et al., 2022b) (1) | 69.86 | 32.67 | 65.07 | 72.66 | 53.94 | 50.66 | 48.39 | 38.79 | 32.07 | 47.32 | 36.41 | 83.47 | 66.36 | 72.51 | 69.93 | 56.01 |
| Tip-adapter (5) | 73.58 | 33.02 | 64.96 | 76.02 | 55.98 | 53.84 | 49.47 | 41.24 | 35.93 | 49.40 | 40.12 | 85.68 | 68.93 | 74.06 | 72.11 | 58.29 |
| Tip-adapter (10) | 75.68 | 40.47 | 69.63 | 78.08 | 60.27 | 58.66 | 53.87 | 46.30 | 36.47 | 51.62 | 41.95 | 87.54 | 72.00 | 76.81 | 77.46 | 61.79 |
| TPT (Shu et al., 2022) | 80.68 | 41.67 | 74.50 | 82.05 | 72.07 | 68.55 | 63.03 | 53.16 | 48.32 | 59.44 | 53.85 | 90.30 | 81.45 | 83.27 | 83.07 | 69.03 |
| ZERO (Farina et al., 2024) | 74.02 | 43.08 | 69.43 | 76.45 | 57.25 | 67.88 | 57.40 | 55.60 | 43.25 | 52.99 | 47.80 | 87.01 | 72.48 | 77.17 | 79.01 | 64.05 |
| MTA (Zanella & Ben Ayed, 2024b) | 78.68 | 44.91 | 73.96 | 80.73 | 78.06 | 68.11 | 61.96 | 55.13 | 49.10 | 59.14 | 54.58 | 89.42 | 80.34 | 81.99 | 82.17 | 69.22 |
| TDA (Karmanov et al., 2024) | 81.25 | 44.76 | 75.15 | 82.40 | 70.41 | 65.43 | 63.55 | 54.75 | 50.41 | 62.24 | 53.54 | 91.02 | 81.40 | 83.98 | 83.44 | 69.58 |
| Mint (Bao et al., 2025) | **84.25** | **62.09** | **81.96** | 85.64 | 84.25 | 76.12 | **70.21** | 74.59 | 64.86 | **74.56** | 69.76 | **92.72** | **87.04** | 86.71 | 86.31 | 78.74 |
| BATCLIP (Maharana et al., 2025) | 83.41 | **65.51** | 81.62 | **86.00** | **85.02** | **76.97** | 69.26 | **75.60** | **66.99** | 73.63 | **69.84** | 91.90 | 85.68 | **87.25** | **86.74** | **79.03** |
| UnInfo (ours) | 82.87 | 47.62 | 77.11 | 84.49 | 77.75 | 67.01 | 66.30 | 58.22 | 53.63 | 65.82 | 58.77 | 92.01 | 84.62 | 86.01 | 85.46 | 72.51 |

for domain shifts, UnInfo had slight improvements and no negative effects compared to No-adapt. This implies that the applicability of our method is not limited within image corruption by adaptively controlling the balance between entropy and uniformity. Moreover, one may incorporate UnInfo with other methods.

Table 15: Test accuracy (%) of OpenAI ViT-B/32 CLIP on CIFAR10-C. The numbers (1), (5), and (10) presented with the method names are the shot numbers per class $n$ used for the few-shot adaptation methods.

| Method | Defocus blur | Glass blur | Motion blur | Zoom blur | Contrast | Elastic transform | Jpeg compression | Pixelate | Gaussian noise | Impulse noise | Shot noise | Brightness | Fog | Frost | Snow | Mean |
|---|---|---|---|---|---|---|---|---|---|---|---|---|---|---|---|---|
| No-adapt | 69.72 | 42.40 | 63.95 | 69.84 | 64.47 | 60.87 | 55.40 | 50.53 | 35.49 | 42.80 | 39.77 | 81.92 | 67.03 | 72.92 | 71.78 | 59.26 |
| Linear probing (1) | 41.98 | 25.94 | 38.93 | 42.06 | 34.42 | 32.59 | 30.10 | 30.23 | 19.53 | 26.66 | 25.02 | 53.88 | 37.30 | 44.90 | 38.23 | 34.79 |
| Linear probing (5) | 59.37 | 36.53 | 55.51 | 62.61 | 48.55 | 46.27 | 44.32 | 40.37 | 34.40 | 35.54 | 38.12 | 70.92 | 51.36 | 60.83 | 58.77 | 49.56 |
| Linear probing (10) | 65.05 | 44.26 | 60.59 | 68.91 | 55.48 | 53.07 | 50.42 | 47.46 | 39.81 | 41.85 | 42.40 | 76.35 | 57.34 | 66.65 | 64.74 | 55.62 |
| Tip-adapter (Zhang et al., 2022b) (1) | 50.65 | 30.22 | 44.85 | 48.18 | 39.18 | 35.79 | 35.79 | 33.42 | 22.60 | 30.28 | 28.37 | 62.64 | 48.03 | 52.04 | 46.42 | 40.57 |
| Tip-adapter (5) | 56.75 | 31.63 | 52.15 | 58.88 | 46.01 | 42.09 | 40.04 | 36.31 | 29.82 | 31.61 | 31.61 | 69.18 | 50.64 | 61.38 | 56.75 | 46.32 |
| Tip-adapter (10) | 60.69 | 36.03 | 55.48 | 64.10 | 48.88 | 49.15 | 44.36 | 40.55 | 31.02 | 34.71 | 31.50 | 71.31 | 54.71 | 64.03 | 61.34 | 49.86 |
| TPT (Shu et al., 2022) | 71.43 | 46.43 | 68.17 | 72.72 | 73.63 | 62.82 | 57.79 | 50.17 | 43.11 | 46.77 | 46.04 | 83.98 | 68.99 | 75.88 | 73.67 | 62.77 |
| ZERO (Farina et al., 2024) | 67.65 | 51.16 | 64.37 | 69.73 | 54.14 | 65.17 | 56.14 | 54.06 | 41.65 | 47.44 | 45.59 | 82.41 | 66.87 | 74.20 | 73.25 | 60.92 |
| MTA (Zanella & Ben Ayed, 2024b) | 71.09 | 46.29 | 67.05 | 72.58 | 74.10 | 64.06 | 57.11 | 52.21 | 37.84 | 46.30 | 42.45 | 84.45 | 70.56 | 76.47 | 74.89 | 62.50 |
| TDA (Karmanov et al., 2024) | 72.51 | 45.95 | 66.10 | 73.10 | 66.05 | 61.50 | 56.83 | 54.68 | 39.58 | 43.98 | 43.01 | 82.88 | 69.18 | 74.95 | 73.03 | 61.56 |
| Mint (Bao et al., 2025) | 75.56 | **60.62** | **76.28** | 78.61 | 76.61 | **69.63** | **62.80** | **65.43** | **56.32** | 52.24 | **60.32** | 86.58 | 74.64 | **78.27** | **78.41** | **70.15** |
| BATCLIP (Maharana et al., 2025) | **76.23** | 53.60 | 75.03 | 76.16 | **78.73** | 67.05 | 61.33 | 55.90 | 50.40 | 52.78 | 54.52 | 86.32 | **74.80** | 78.22 | 76.88 | 67.86 |
| UnInfo (ours) | 75.55 | 50.34 | 71.97 | 76.45 | 73.95 | 65.80 | 59.79 | 54.81 | 41.72 | 45.69 | 43.85 | 86.53 | 74.48 | 77.56 | 76.76 | 65.02 |

Table 16: Test accuracy (%) of SigLIP on CIFAR10-C. The numbers (1), (5), and (10) presented with the method names are the shot numbers per class $n$ used for the few-shot adaptation methods.

| Method | Defocus blur | Glass blur | Motion blur | Zoom blur | Contrast | Elastic transform | Jpeg compression | Pixelate | Gaussian noise | Impulse noise | Shot noise | Brightness | Fog | Frost | Snow | Mean |
|---|---|---|---|---|---|---|---|---|---|---|---|---|---|---|---|---|
| No-adapt | 65.65 | 39.06 | 65.55 | 66.51 | 55.36 | 50.81 | 43.96 | 57.18 | 28.35 | 41.48 | 32.61 | 86.60 | 73.28 | 74.88 | 76.80 | 57.21 |
| Linear probing (1) | 37.04 | 25.77 | 41.69 | 39.88 | 29.49 | 31.71 | 25.06 | 29.08 | 16.82 | 27.46 | 22.22 | 56.13 | 42.90 | 43.71 | 43.98 | 34.20 |
| Linear probing (5) | 59.14 | 38.04 | 57.83 | 62.53 | 45.62 | 46.28 | 38.10 | 46.59 | 27.36 | 41.35 | 31.53 | 79.33 | 60.58 | 61.66 | 65.63 | 50.77 |
| Linear probing (10) | 65.04 | 44.06 | 63.90 | 69.78 | 51.95 | 52.19 | 44.06 | 54.37 | 31.64 | 47.51 | 34.74 | 83.76 | 69.09 | 69.64 | 73.06 | 56.99 |
| Tip-adapter (Zhang et al., 2022b) (1) | 48.14 | 29.99 | 50.66 | 49.97 | 36.63 | 38.87 | 29.57 | 34.67 | 19.93 | 33.93 | 24.04 | 70.59 | 52.00 | 51.39 | 54.95 | 41.69 |
| Tip-adapter (5) | 54.54 | 33.24 | 54.65 | 58.79 | 46.31 | 45.41 | 33.76 | 40.57 | 24.97 | 38.06 | 29.63 | 76.68 | 59.38 | 59.34 | 62.45 | 47.85 |
| Tip-adapter (10) | 57.60 | 40.62 | 59.49 | 64.34 | 49.09 | 50.54 | 37.49 | 45.00 | 26.96 | 40.65 | 30.00 | 81.24 | 66.00 | 68.45 | 69.43 | 52.46 |
| TPT (Shu et al., 2022) | 43.67 | 25.82 | 41.68 | 44.43 | 65.41 | 36.49 | 24.20 | 34.42 | 11.30 | 20.44 | 12.41 | 26.11 | 59.76 | 30.42 | 25.45 | 33.47 |
| ZERO (Farina et al., 2024) | 46.85 | 31.71 | 43.53 | 47.70 | 64.35 | 42.45 | 29.64 | 42.78 | 12.36 | 22.79 | 13.36 | 28.66 | 56.20 | 31.27 | 25.71 | 35.96 |
| MTA (Zanella & Ben Ayed, 2024b) | 50.72 | 30.36 | 49.55 | 52.40 | 74.46 | 42.37 | 31.38 | 42.82 | 11.35 | 21.43 | 12.30 | 32.52 | 69.17 | 35.66 | 27.59 | 38.94 |
| TDA (Karmanov et al., 2024) | 69.41 | 44.24 | 67.93 | 70.13 | 57.40 | 55.76 | 45.52 | 57.79 | 30.79 | 45.98 | 34.85 | 87.40 | 75.34 | 76.18 | 78.47 | 59.81 |
| Mint (Bao et al., 2025) | **80.41** | **58.94** | **81.53** | **81.91** | **78.13** | **69.87** | 50.25 | **71.41** | 50.48 | 63.37 | 56.39 | **91.54** | **84.90** | 81.89 | 82.17 | **72.21** |
| BATCLIP (Maharana et al., 2025) | 79.70 | 53.62 | 78.76 | 81.68 | 76.67 | 69.07 | **53.89** | 65.63 | 50.49 | 68.27 | **58.57** | 90.75 | 83.20 | **83.30** | **83.56** | 71.81 |
| UnInfo (ours) | 70.45 | 42.21 | 70.73 | 72.85 | 63.50 | 55.37 | 48.45 | 65.53 | 31.90 | 48.95 | 36.96 | 88.90 | 77.54 | 78.07 | 80.02 | 62.09 |

Table 17: Test accuracy (%) of OpenCLIP ViT-B/16 on CIFAR100-C. The numbers (1), (5), and (10) presented with the method names are the shot numbers per class $n$ used for the few-shot adaptation methods.

| Method | Defocus blur | Glass blur | Motion blur | Zoom blur | Contrast | Elastic transform | Jpeg compression | Pixelate | Gaussian noise | Impulse noise | Shot noise | Brightness | Fog | Frost | Snow | Mean |
|---|---|---|---|---|---|---|---|---|---|---|---|---|---|---|---|---|
| No-adapt | 53.49 | 16.15 | 48.43 | 56.98 | 39.75 | 32.74 | 34.52 | 25.62 | 22.08 | 26.82 | 24.83 | 65.20 | 51.83 | 55.53 | 56.89 | 40.72 |
| Linear probing (1) | 24.72 | 12.78 | 23.63 | 27.82 | 18.01 | 17.37 | 15.27 | 15.65 | 12.15 | 14.47 | 13.24 | 35.68 | 25.38 | 26.34 | 26.02 | 20.57 |
| Linear probing (5) | 42.81 | 21.34 | 38.85 | 46.94 | 30.84 | 29.29 | 26.55 | 25.53 | 21.31 | 25.43 | 23.52 | 57.72 | 41.68 | 45.07 | 44.18 | 34.74 |
| Linear probing (10) | 49.96 | 26.18 | 46.05 | 53.77 | 36.11 | 35.45 | 31.20 | 30.20 | 25.01 | 30.01 | 27.17 | 63.93 | 48.39 | 51.08 | 51.55 | 40.40 |
| Tip-adapter (Zhang et al., 2022b) (1) | 38.68 | 15.78 | 34.99 | 41.14 | 26.40 | 24.54 | 23.05 | 20.95 | 16.61 | 19.84 | 18.14 | 51.93 | 36.26 | 39.21 | 39.66 | 29.81 |
| Tip-adapter (5) | 43.94 | 21.59 | 39.98 | 48.26 | 31.69 | 30.59 | 27.78 | 26.04 | 20.13 | 24.52 | 22.17 | 59.48 | 42.73 | 46.30 | 45.72 | 35.39 |
| Tip-adapter (10) | 47.07 | 24.18 | 43.23 | 50.98 | 34.11 | 34.09 | 30.20 | 28.19 | 22.14 | 28.01 | 24.61 | 62.49 | 45.22 | 49.37 | 49.36 | 38.22 |
| TPT (Shu et al., 2022) | 50.46 | 15.45 | 46.21 | 54.80 | 35.89 | 31.62 | 31.30 | 22.19 | 20.95 | 24.90 | 23.37 | 62.46 | 50.18 | 53.16 | 53.71 | 38.44 |
| ZERO (Farina et al., 2024) | 45.36 | 14.47 | 42.04 | 48.80 | 27.68 | 33.63 | 28.57 | 23.48 | 13.70 | 24.37 | 14.90 | 55.40 | 43.05 | 44.76 | 48.14 | 33.89 |
| MTA (Zanella & Ben Ayed, 2024b) | 48.24 | 14.24 | 43.97 | 52.46 | 29.26 | 32.38 | 31.50 | 24.16 | 13.70 | 27.36 | 16.05 | 59.03 | 49.23 | 51.32 | 51.95 | 36.32 |
| TDA (Karmanov et al., 2024) | 55.74 | 21.26 | 51.05 | 59.60 | 42.52 | 36.08 | 37.35 | 30.93 | 26.47 | 32.89 | 29.24 | 67.77 | 53.81 | 57.92 | 58.68 | 44.09 |
| Mint (Bao et al., 2025) | 58.44 | **31.86** | **55.64** | 60.46 | **54.11** | **45.08** | **43.28** | **40.33** | 34.06 | 40.29 | 35.47 | **69.72** | **57.97** | **59.41** | **60.13** | **49.75** |
| BATCLIP (Maharana et al., 2025) | 54.16 | 30.35 | 50.26 | 56.38 | 51.64 | 39.62 | 35.54 | 14.77 | **34.25** | 31.78 | **36.34** | 65.62 | 52.18 | 54.51 | 54.10 | 44.10 |
| UnInfo (ours) | 56.82 | 18.73 | 50.73 | 59.66 | 43.80 | 36.33 | 36.17 | 29.40 | 25.14 | 29.20 | 28.74 | 67.32 | 54.29 | 57.79 | 59.53 | 43.58 |

Table 18: Test accuracy (%) of OpenAI ViT-B/32 CLIP on CIFAR100-C. The numbers (1), (5), and (10) presented with the method names are the shot numbers per class $n$ used for the few-shot adaptation methods.

| Method | Defocus blur | Glass blur | Motion blur | Zoom blur | Contrast | Elastic transform | Jpeg compression | Pixelate | Gaussian noise | Impulse noise | Shot noise | Brightness | Fog | Frost | Snow | Mean |
|---|---|---|---|---|---|---|---|---|---|---|---|---|---|---|---|---|
| No-adapt | 36.76 | 14.21 | 36.10 | 40.31 | 29.54 | 26.36 | 25.96 | 22.02 | 14.78 | 13.83 | 15.99 | 48.18 | 38.00 | 40.57 | 38.92 | 29.44 |
| Linear probing (1) | 15.89 | 9.16 | 15.71 | 17.27 | 11.89 | 11.50 | 11.02 | 10.19 | 8.26 | 8.55 | 9.42 | 20.62 | 15.12 | 17.15 | 15.41 | 13.14 |
| Linear probing (5) | 27.66 | 16.79 | 25.97 | 30.32 | 19.60 | 20.20 | 19.25 | 17.91 | 13.56 | 14.83 | 15.76 | 37.09 | 25.87 | 30.47 | 28.89 | 22.94 |
| Linear probing (10) | 32.98 | 20.29 | 30.97 | 35.82 | 24.22 | 24.90 | 23.67 | 22.06 | 15.43 | 17.69 | 18.06 | 42.73 | 30.93 | 35.97 | 34.05 | 27.32 |
| Tip-adapter (Zhang et al., 2022b) (1) | 19.55 | 11.28 | 19.11 | 22.05 | 15.12 | 14.45 | 14.70 | 13.14 | 10.50 | 10.05 | 11.35 | 27.36 | 19.07 | 21.28 | 19.82 | 16.59 |
| Tip-adapter (5) | 27.65 | 15.57 | 26.05 | 31.35 | 20.37 | 20.21 | 19.14 | 18.21 | 13.35 | 14.31 | 14.58 | 36.09 | 25.39 | 29.54 | 28.80 | 22.71 |
| Tip-adapter (10) | 30.48 | 18.33 | 29.72 | 33.97 | 22.63 | 23.42 | 22.33 | 19.47 | 14.46 | 15.82 | 16.03 | 40.46 | 29.14 | 33.67 | 31.63 | 25.44 |
| TPT (Shu et al., 2022) | 37.58 | 18.48 | 37.49 | 42.78 | 31.73 | 29.51 | 28.94 | 21.68 | 14.70 | 15.39 | 16.02 | 51.14 | 39.14 | 44.12 | 43.11 | 31.45 |
| ZERO (Farina et al., 2024) | 35.81 | 20.35 | 35.76 | 39.47 | 24.29 | 32.51 | 26.44 | 25.27 | 14.43 | 15.55 | 15.57 | 46.81 | 36.69 | 40.59 | 40.50 | 30.00 |
| MTA (Zanella & Ben Ayed, 2024b) | 37.64 | 15.14 | 37.73 | 41.71 | 34.58 | 29.22 | 26.81 | 22.07 | 14.30 | 14.62 | 15.33 | 49.87 | 39.67 | 43.27 | 41.73 | 30.91 |
| TDA (Karmanov et al., 2024) | 41.59 | 19.18 | 40.82 | 45.59 | 31.99 | 30.52 | 31.04 | 26.57 | 18.30 | 19.38 | 19.76 | 53.17 | 39.94 | 45.64 | 44.46 | 33.86 |
| Mint (Bao et al., 2025) | **47.68** | **27.05** | 44.83 | **50.37** | **42.78** | **34.99** | **33.65** | **29.21** | **25.94** | **23.20** | **28.68** | **59.83** | **45.00** | **47.11** | **47.60** | **39.19** |
| BATCLIP (Maharana et al., 2025) | 42.73 | 19.97 | 40.43 | 45.95 | 32.07 | 31.92 | 28.67 | 23.87 | 10.04 | 19.67 | 15.45 | 54.62 | 41.15 | 41.09 | 41.96 | 32.64 |
| UnInfo (ours) | 43.40 | 18.38 | 40.96 | 46.02 | 35.66 | 28.53 | 27.59 | 24.75 | 16.42 | 16.53 | 18.26 | 53.30 | 42.11 | 43.54 | 42.51 | 33.20 |

Table 19: Test accuracy (%) of SigLIP on CIFAR100-C. The numbers (1), (5), and (10) presented with the method names are the shot numbers per class $n$ used for the few-shot adaptation methods.

| Method | Defocus blur | Glass blur | Motion blur | Zoom blur | Contrast | Elastic transform | Jpeg compression | Pixelate | Gaussian noise | Impulse noise | Shot noise | Brightness | Fog | Frost | Snow | Mean |
|---|---|---|---|---|---|---|---|---|---|---|---|---|---|---|---|---|
| No-adapt | 40.19 | 15.63 | 40.63 | 44.75 | 29.20 | 24.75 | 21.60 | 24.48 | 14.00 | 25.16 | 15.59 | 60.89 | 47.84 | 47.56 | 49.88 | 33.48 |
| Linear probing (1) | 20.53 | 12.21 | 20.85 | 22.20 | 15.43 | 14.01 | 9.89 | 13.65 | 6.80 | 12.81 | 7.87 | 32.83 | 23.52 | 22.87 | 22.18 | 17.18 |
| Linear probing (5) | 33.54 | 19.29 | 31.88 | 35.74 | 25.62 | 22.75 | 17.14 | 22.23 | 11.34 | 21.07 | 13.01 | 51.42 | 37.53 | 36.20 | 36.83 | 27.71 |
| Linear probing (10) | 38.31 | 21.96 | 37.24 | 40.84 | 28.99 | 27.36 | 20.28 | 25.86 | 13.31 | 25.03 | 16.16 | 57.11 | 42.13 | 41.14 | 42.92 | 31.91 |
| Tip-adapter (Zhang et al., 2022b) (1) | 25.76 | 13.60 | 25.69 | 28.34 | 19.76 | 17.96 | 12.99 | 16.66 | 9.60 | 17.01 | 10.33 | 41.83 | 29.34 | 27.90 | 28.60 | 21.69 |
| Tip-adapter (5) | 31.85 | 18.91 | 32.08 | 35.47 | 24.85 | 23.39 | 17.52 | 21.44 | 11.02 | 20.05 | 12.19 | 50.71 | 36.86 | 35.38 | 36.33 | 27.20 |
| Tip-adapter (10) | 35.24 | 21.21 | 35.76 | 38.30 | 27.13 | 26.36 | 19.53 | 24.14 | 11.96 | 23.09 | 13.76 | 54.41 | 38.75 | 39.33 | 40.03 | 29.93 |
| TPT (Shu et al., 2022) | 20.34 | 6.88 | 18.20 | 21.84 | 36.35 | 11.38 | 8.54 | 10.10 | 2.26 | 5.27 | 2.74 | 8.61 | 34.41 | 11.18 | 8.51 | 13.77 |
| ZERO (Farina et al., 2024) | 15.25 | 7.20 | 13.15 | 16.72 | 25.03 | 10.51 | 7.53 | 12.37 | 4.25 | 7.09 | 4.52 | 7.33 | 22.64 | 8.39 | 6.41 | 11.23 |
| MTA (Zanella & Ben Ayed, 2024b) | 20.48 | 6.85 | 16.76 | 23.08 | 33.07 | 11.25 | 9.21 | 12.69 | 4.40 | 8.32 | 4.88 | 9.96 | 29.63 | 11.60 | 8.72 | 14.06 |
| TDA (Karmanov et al., 2024) | 42.97 | 18.38 | 43.33 | 47.27 | 31.96 | 28.97 | 24.24 | 27.91 | 14.71 | 28.05 | 16.90 | 62.97 | 50.07 | 49.16 | 51.21 | 35.87 |
| Mint (Bao et al., 2025) | **49.68** | **28.04** | **53.91** | **55.22** | **44.07** | **36.87** | **26.24** | **37.25** | **17.18** | **33.94** | **19.58** | **66.84** | **51.88** | **51.13** | **53.16** | **41.67** |
| BATCLIP (Maharana et al., 2025) | 47.11 | 22.44 | 46.27 | 51.43 | 37.74 | 34.03 | 24.79 | 30.08 | 15.51 | 30.89 | 17.95 | 64.05 | 50.13 | 51.08 | 50.92 | 38.30 |
| UnInfo (ours) | 43.81 | 19.53 | 44.05 | 47.92 | 31.60 | 27.88 | 23.59 | 26.50 | 14.64 | 27.91 | 16.38 | 63.43 | 49.90 | 50.51 | 52.74 | 36.03 |

Table 20: Test accuracy (%) on domain shifts. The numbers (1), (5), and (10) presented with the method names are the shot numbers per class $n$ used for the few-shot adaptation methods. '-' represents that the setting is infeasible due to the dataset size.

| Method | ImageNet | ImageNet-A | ImageNet-R | Mean |
|---|---|---|---|---|
| No-adapt | 66.97 | 32.91 | 74.36 | 58.08 |
| Linear probing (1) | 37.86 | 14.07 | 23.48 | 25.13 |
| Linear probing (5) | 55.95 | - | 52.91 | - |
| Linear probing (10) | 62.50 | - | 62.51 | - |
| Tip-adapter (Zhang et al., 2022b) (1) | 57.55 | 30.80 | 63.70 | 50.68 |
| Tip-adapter (5) | 61.76 | - | 63.82 | - |
| Tip-adapter (10) | 64.00 | - | 66.08 | - |
| TPT (Shu et al., 2022) | 70.70 | 19.92 | 80.26 | 56.96 |
| ZERO (Farina et al., 2024) | 69.02 | 43.23 | 75.53 | 62.59 |
| MTA (Zanella & Ben Ayed, 2024b) | 68.25 | 35.93 | 74.72 | 59.63 |
| TDA (Karmanov et al., 2024) | 70.33 | 33.24 | 76.81 | 60.13 |
| UnInfo (ours) | 67.82 | 33.13 | 76.53 | 59.16 |

## E.2 Ablation Study

**Efficacy of updating the image encoder.** To verify the efficacy of updating the image encoder rather than image embeddings, we compared a variant of UnInfo that freezes the image encoder and updates image embeddings as optimized parameters. The loss is the same as Eq. (8), but the knowledge distillation loss $\mathcal{L}_{pl}$ is omitted because the EMA teacher is not available in this setting. Tab. 21 shows the result. UnInfo (embedding) had only a slight effect on accuracy compared to No-adapt, suggesting that updating the image encoder is efficient. This is because UnInfo can accumulate knowledge of the target distribution by updating the image encoder, whereas updating image embeddings is episodic and cannot leverage knowledge from past inputs.

Table 21: Test accuracy (%) of UnInfo variants on ImageNet-C with ViT-B/16 CLIP.

| Method | Defocus blur | Glass blur | Motion blur | Zoom blur | Contrast | Elastic transform | Jpeg compression | Pixelate | Gaussian noise | Impulse noise | Shot noise | Brightness | Fog | Frost | Snow | Mean |
|---|---|---|---|---|---|---|---|---|---|---|---|---|---|---|---|---|
| No-adapt | 28.31 | 11.89 | 19.16 | 17.61 | 17.87 | 13.22 | 36.79 | 37.01 | 6.08 | 6.17 | 7.85 | 54.89 | 34.55 | 27.35 | 27.67 | 23.09 |
| UnInfo (ours) | 31.51 | 16.76 | 23.47 | 20.40 | 22.81 | 16.59 | 42.03 | 42.38 | 7.56 | 10.60 | 11.36 | 57.75 | 39.16 | 31.65 | 32.40 | 27.10 |
| UnInfo (embedding) | 28.19 | 11.81 | 19.13 | 17.65 | 17.89 | 13.37 | 36.76 | 36.92 | 6.02 | 6.13 | 7.87 | 54.74 | 34.39 | 27.31 | 27.78 | 23.06 |

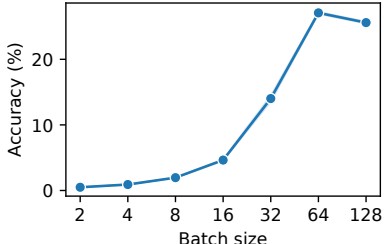

Figure 6: Sensitivity analysis of the batch size. The mean and standard deviation of the test accuracy calculated over the ImageNet-C corruptions are plotted.

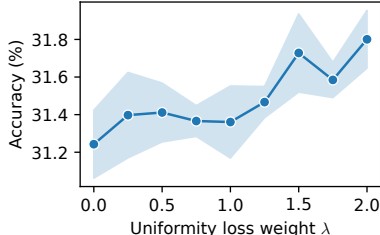

Figure 7: Hyperparameter sensitivity on only defocus blur.

**Ablation on the batch size.** Fig. 6 shows the mean accuracy of UnInfo over ImageNet-C corruptions with ViT-B/16 CLIP. When the batch size $\leq 32$, the accuracy dropped significantly. This is because the uniformity estimation becomes poor with small batch sizes, which is one limitation of UnInfo, as mentioned in Sec. 7. One possible modification is to accumulate batch statistics, such as a moving average of uniformity, to compensate for the estimation when using a single small batch size.

### E.3   Hyperparameter Selection

In Sec. 5.2, we selected the weight of uniformity loss $\lambda$ based on brightness, defocus blur, elastic transform, and Gaussian noise. Here, we searched $\lambda$ based on only defocus blur from $\{0.0, 0.25, \ldots, 2.0\}$ and checked the selected $\lambda$'s performance to examine whether the hyperparameter setting does not overfit. Fig. 7 shows the result. The best values based on defocus blur were 1.5 and 2.0. From the sensitivity analysis in Fig. 2, these values do not have a large effect on the average accuracy across the other corruption types, suggesting the reasonableness of the $\lambda$ used in the experiments.

Moreover, the hyperparameter setting we used in the experiments also works on ImageNet-C-bar. The corruption types in ImageNet-C-bar are selected to be dissimilar to ImageNet-C (Mintun et al., 2021), suggesting that our hyperparameter setting does not overfit to the selected corruption types.

### E.4   Continual Adaptation

We tested UnInfo in the continual TTA setting, where adaptation continues over all corruption types sequentially without resetting the model. Here, we used the same hyperparameter setting as in the main experiments, except setting the learning rate to $10^{-4}$ to avoid catastrophic forgetting. Tab. 22 shows the result on ImageNet-C with ViT-B/16 CLIP. Although UnInfo improved accuracy compared to No-adapt on most corruption types, the accuracy gains for the latter half of corruption types are smaller than those for the former half. This is because the EMA teacher retains the LoRA parameters adapted to previous corruptions, which affect current adaptation. One possible modification would be to adaptively reset LoRA using change detection, which is one direction for future work.

Table 22: Test accuracy (%) of UnInfo on ImageNet-C under the continual adaptation setting.

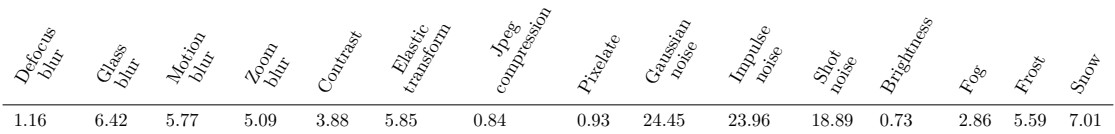

| Method | Defocus blur | Glass blur | Motion blur | Zoom blur | Contrast | Elastic transform | Jpeg compression | Pixelate | Gaussian noise | Impulse noise | Shot noise | Brightness | Fog | Frost | Snow | Mean |
|---|---|---|---|---|---|---|---|---|---|---|---|---|---|---|---|---|
| No-adapt | 28.31 | 11.89 | 19.16 | 17.61 | 17.87 | 13.22 | 36.79 | 37.01 | 6.08 | 6.17 | 7.85 | 54.89 | 34.55 | 27.35 | 27.67 | 23.09 |
| UnInfo | $29.21_{\pm0.05}$ | $13.62_{\pm0.06}$ | $22.14_{\pm0.36}$ | $20.70_{\pm0.31}$ | $20.21_{\pm0.32}$ | $17.24_{\pm0.70}$ | $36.33_{\pm0.33}$ | $36.64_{\pm0.50}$ | $6.06_{\pm0.07}$ | $6.41_{\pm0.10}$ | $8.23_{\pm0.11}$ | $52.08_{\pm0.22}$ | $34.87_{\pm0.37}$ | $28.71_{\pm0.03}$ | $27.88_{\pm0.34}$ | $24.02_{\pm0.09}$ |

Table 23: The ratio of the class to which an unadapted ViT-B/16 CLIP classified the largest number of ImageNet-C samples.

| Defocus blur | Glass blur | Motion blur | Zoom blur | Contrast | Elastic transform | Jpeg compression | Pixelate | Gaussian noise | Impulse noise | Shot noise | Brightness | Fog | Frost | Snow |
|---|---|---|---|---|---|---|---|---|---|---|---|---|---|---|
| 1.16 | 6.42 | 5.77 | 5.09 | 3.88 | 5.85 | 0.84 | 0.93 | 24.45 | 23.96 | 18.89 | 0.73 | 2.86 | 5.59 | 7.01 |

### E.5 Analysis of Model Collapse with Entropy Loss

In Tabs. 5 and 6, we observed that the entropy-only loss ($\mathcal{L}_{\text{ent}}$) resulted in model collapse. Here, we investigated the statistics of the model prediction to understand this phenomenon. Tab. 23 shows the ratio of the class to which an unadapted ViT-B/16 CLIP classified the largest number of ImageNet-C samples. We can observe a significant imbalance due to image corruption. A large number of samples ($> 1/1000 = 0.1\%$), especially around 20% on noise corruptions, were classified to a single class, where the entropy-only loss exacerbates the imbalance. This is due to the degradation of uniformity observed in Sec. 3; By image corruption, a large number of image embeddings are distributed within the region of a single class in the embedding space.

The instability of simple entropy minimization is also addressed in classification TTA studies, such as SAR (Niu et al., 2023) and ReCAP (Hu et al., 2025). Combining these techniques is one direction of our future work.

### E.6 Optimized Parameters

To validate the effect of using LoRA in UnInfo, we changed the optimized parameter set. Here, we compared two variants of UnInfo:

**LayerNorm:** updates the affine parameters in LayerNorm layers (Ba et al., 2016) in the image encoder instead of LoRA, as existing studies (e.g., BATCLIP (Maharana et al., 2025) and Mint (Bao et al., 2025))

**LayerNorm (w/o EMA teacher):** also updates the affine parameters in LayerNorm layers, but the EMA teacher regularization term $\mathcal{L}_{\text{pl}}$ is omitted. We compared this variant to verify whether model collapse can be avoided by limiting the number of optimized parameters, as we introduced $\mathcal{L}_{\text{pl}}$ to avoid model collapse.

Tab. 24 shows the result. Although LayerNorm (w/o EMA teacher) outperformed LoRA (original UnInfo) on several corruption types, LoRA achieved the best average accuracy. While LayerNorm sometimes had significantly higher accuracy than LayerNorm (w/o EMA teacher), e.g., defocus blur, it also sometimes had lower accuracy. When optimizing LayerNorm parameters, model collapse can be avoided without $\mathcal{L}_{\text{pl}}$ because LayerNorm has fewer degrees of freedom than LoRA. However, UnInfo (LoRA with $\mathcal{L}_{\text{pl}}$) performed the best because it hits a better balance between adaptation flexibility and regularization.

### E.7 Consistency across Severity

In real-world applications, whether image corruption occurs cannot be known in advance. To examine whether UnInfo negatively affects the performance on clean or low-severity corruption, we changed the severity from 0 (clean) to 5. Tab. 25 shows the result. Although the accuracy gain is higher at higher

Table 24: LoRA vs. LayerNorm affine parameters in UnInfo on ImageNet-C.

| Method | Defocus blur | Glass blur | Motion blur | Zoom blur | Contrast | Elastic transform | Jpeg compression | Pixelate | Gaussian noise | Impulse noise | Shot noise | Brightness | Fog | Frost | Snow | Mean |
|---|---|---|---|---|---|---|---|---|---|---|---|---|---|---|---|---|
| No-adapt | 28.31 | 11.89 | 19.16 | 17.61 | 17.87 | 13.22 | 36.79 | 37.01 | 6.08 | 6.17 | 7.85 | 54.89 | 34.55 | 27.35 | 27.67 | 23.09 |
| LayerNorm | $29.22_{\pm 0.04}$ | $12.22_{\pm 0.00}$ | $19.78_{\pm 0.02}$ | $18.31_{\pm 0.03}$ | $18.62_{\pm 0.06}$ | $13.90_{\pm 0.01}$ | $38.12_{\pm 0.03}$ | $38.56_{\pm 0.01}$ | $6.29_{\pm 0.01}$ | $6.40_{\pm 0.01}$ | $8.17_{\pm 0.01}$ | $56.23_{\pm 0.02}$ | $35.84_{\pm 0.01}$ | $28.20_{\pm 0.03}$ | $28.69_{\pm 0.04}$ | $23.90_{\pm 0.01}$ |
| LayerNorm (w/o EMA teacher) | $20.74_{\pm 0.88}$ | $15.84_{\pm 2.13}$ | $15.24_{\pm 0.87}$ | $18.87_{\pm 3.05}$ | $22.22_{\pm 0.69}$ | $\mathbf{21.74_{\pm 1.07}}$ | $\mathbf{42.08_{\pm 1.85}}$ | $\mathbf{45.51_{\pm 0.30}}$ | $6.97_{\pm 0.15}$ | $7.54_{\pm 0.50}$ | $7.47_{\pm 0.39}$ | $\mathbf{57.83_{\pm 0.19}}$ | $31.65_{\pm 4.15}$ | $23.47_{\pm 0.72}$ | $23.47_{\pm 2.39}$ | $24.04_{\pm 0.61}$ |
| LoRA (ours) | $\mathbf{31.51_{\pm 0.19}}$ | $\mathbf{16.76_{\pm 1.62}}$ | $\mathbf{23.47_{\pm 0.74}}$ | $\mathbf{20.40_{\pm 0.80}}$ | $\mathbf{22.81_{\pm 0.27}}$ | $16.59_{\pm 0.40}$ | $42.03_{\pm 0.23}$ | $42.38_{\pm 0.26}$ | $\mathbf{7.56_{\pm 3.50}}$ | $\mathbf{10.60_{\pm 0.41}}$ | $\mathbf{11.36_{\pm 0.86}}$ | $57.75_{\pm 0.11}$ | $\mathbf{39.16_{\pm 0.34}}$ | $\mathbf{31.65_{\pm 0.48}}$ | $\mathbf{32.40_{\pm 0.13}}$ | $\mathbf{27.10_{\pm 0.12}}$ |

Table 25: Average test accuracy (%) of ViT-B/16 CLIP on ImageNet-C across severity.

| Method | Severity | | | | | | Mean |
|---|---|---|---|---|---|---|---|
| | 0 (clean) | 1 | 2 | 3 | 4 | 5 | |
| No-adapt | 66.97 | 56.09 | 48.54 | 42.14 | 32.85 | 23.06 | 44.94 |
| UnInfo (ours) | $67.82_{\pm 0.05}$ | $58.83_{\pm 0.08}$ | $52.13_{\pm 0.01}$ | $46.64_{\pm 0.08}$ | $37.81_{\pm 0.03}$ | $27.10_{\pm 0.30}$ | $48.39_{\pm 0.04}$ |

severities, UnInfo showed consistent improvement regardless of corruption severity, including clean images, suggesting that UnInfo does not have a negative effect on low-severity images.

