# OpenReview forum: "Uniformity First: Uniformity-aware Test-time Adaptation of Vision-language Models against Image Corruption"
_TMLR — Under review for TMLR_

### Review · Reviewer_ZVoC · 2026-06-13

**Summary Of Contributions:**

The paper studies test-time adaptation of vision-language models, such as CLIP, under common corruptions. The authors first observe that common corruptions produce higher prediction entropy and uniformity loss than standard domain shifts. Based on this observation, they propose Uniformity-aware Information-balanced Test-time Adaptation (UnInfo).

UnInfo optimizes LoRA modules in the image encoder using three components: (1) entropy minimization, (2) a uniformity loss, and (3) a prediction-distillation loss from an exponential-moving-average teacher. UnInfo also dynamically reweights the entropy and uniformity losses based on the mutual information of the batch predictions.

The paper evaluates UnInfo on ImageNet-C and ImageNet-C-bar. The authors compare it against no adaptation, few-shot linear probing, Tip-Adapter, and several test-time adaptation methods, including TPT, ZERO, MTA, and TDA. UnInfo improves the mean accuracy over the strongest included baseline on both datasets. The paper also includes component ablations, sensitivity analyses, visualizations, and computational-efficiency measurements.

Strengths: The problem is relevant, the proposed method is relatively simple, and the experiments indicate that adapting the image encoder with a stabilized objective can improve CLIP robustness to corruptions.

Weaknesses: The paper lacks comparisons with recent state-of-the-art methods, does not sufficiently support its interpretation of uniformity as preserved information, provides insufficient ablation of encoder-level versus embedding-level adaptation, and does not adequately explain the efficiency comparison with TDA.

**Audience:**

Yes

**Audience Explanation:**

The finding that common corruptions are associated with concentrated CLIP image representations can be potentially useful, even though the paper currently overinterprets this as direct information loss. Additionally, dynamical balancing confidence maximization and feature dispersion may be useful for designing more stable adaptation methods. I believe the research direction and some of the empirical findings are relevant to a portion of the TMLR audience.

**Claims And Evidence:**

No

**Claims Explanation:**

1. Uniformity is not equivalent to preserved information.

The paper states that a low value of uniformity loss suggests that more information is preserved. The uniformity loss used measures pairwise distances between normalized embeddings. Minimizing it encourages samples within a batch to spread across the hypersphere. This can reduce representational collapse and increase feature diversity, but it does not by itself measure semantic information.

2. Important related methods are missing.

The paper does not compare against closely related works such as BATCLIP[1], CLIPArTT[2], WATT[3], and Mint[4]. These methods also study CLIP adaptation under common corruptions. This weakens both the novelty claim and the experimental evaluation.

3. Missing controlled ablation on encoder and embedding adaptation.

The paper argues that adapting embeddings is insufficient and the image encoder must be updated. A controlled ablation is needed to verify this using the same loss and optimization settings while adapting different components.

4. The entropy-only result requires verification.

Entropy minimization achieves exactly 0.10% accuracy on every corruption, this is equivalent to chance-level accuracy. The authors should verify whether this is caused by model collapse, the learning rate, temperature, adaptation protocol, or an implementation error.

5. The efficiency comparison with TDA is unclear.

UnInfo requires both forward and backward propagation, whereas TDA is backpropagation-free. The authors should explain why UnInfo is reported to be computationally more efficient than TDA.

[1] S. Maharana, B. Zhang, L. Karlinsky, R. Feris, and Y. Guo, “BATCLIP: Bimodal Online Test-Time Adaptation for CLIP,” in Proceedings of the IEEE/CVF International Conference on Computer Vision (ICCV), 2025, pp. 1569–1579.

[2] G. A. Vargas Hakim, D. Osowiechi, M. Noori, M. Cheraghalikhani, A. Bahri, M. Yazdanpanah, I. Ben Ayed, and C. Desrosiers, “CLIPArTT: Adaptation of CLIP to New Domains at Test Time,” in Proceedings of the Winter Conference on Applications of Computer Vision (WACV), 2025, pp. 7092–7101.

[3] D. Osowiechi, M. Noori, G. A. Vargas Hakim, M. Yazdanpanah, A. Bahri, M. Cheraghalikhani, S. Dastani, F. Beizaee, I. Ben Ayed, and C. Desrosiers, “WATT: Weight Average Test Time Adaptation of CLIP,” in Advances in Neural Information Processing Systems, vol. 37, 2024.

[4] W. Bao, R. Deng, and J. He, “Mint: A Simple Test-Time Adaptation of Vision-Language Models against Common Corruptions,” in Advances in Neural Information Processing Systems, vol. 38, 2025.

**Requested Changes:**

1. Clarify the main claim: Revise the interpretation of uniformity as preserved information, or provide direct evidence that links uniformity with semantic information.

2. Add missing baselines: Compare against closely related methods, including BATCLIP, CLIPArTT, WATT, and Mint. The introduction should also be revised, as it currently claims that existing methods do not evaluate on corruptions.

3. Add a controlled ablation: Compare encoder-level and embedding-level adaptation using the same loss, optimizer, and adaptation protocol.

4. Verify entropy-only collapse: Explain the consistent 0.10% accuracy and rule out implementation or hyperparameter issues.

5. Clarify the TDA efficiency comparison: Report a controlled comparison and explain why a backpropagation-based method is faster than TDA.

6. Clarify the shift definitions: Clearly distinguish domain shift from sensor degradation and justify why the evaluated corruptions are categorized as sensor degradation.

---

> ### Author Response · Authors · 2026-07-15
>
> Dear reviewer `ZVoC`,
>
> We appreciate your careful reading and insightful comments.
>
> > Uniformity is not equivalent to preserved information.
> > The paper states that a low value of uniformity loss suggests that more information is preserved. The uniformity loss used measures pairwise distances between normalized embeddings. Minimizing it encourages samples within a batch to spread across the hypersphere. This can reduce representational collapse and increase feature diversity, but it does not by itself measure semantic information.
>
> > Requested change 1: Clarify the main claim: Revise the interpretation of uniformity as preserved information, or provide direct evidence that links uniformity with semantic information.
>
> Thank you for your clarification.
> Indeed, uniformity itself does not directly measure the amount of semantic information, but, theoretically, it is closely related to mutual information [Wang & Isola, 2020].
> The expectation of uniformity is equivalent to mutual information, which measures the amount of information preserved.
> The preserved information includes semantic information, i.e., alignment with text embeddings.
> Thus, maintaining uniformity is necessary to preserve semantic information, as it cannot be recovered if uniformity is collapsed (i.e., all image embeddings are concentrated in a single region).
> Then, semantic information is enhanced by the entropy loss in **Eq. (5)**.
>
>
>
> > Important related methods are missing.
> > The paper does not compare against closely related works such as BATCLIP[1], CLIPArTT[2], WATT[3], and Mint[4]. These methods also study CLIP adaptation under common corruptions. This weakens both the novelty claim and the experimental evaluation.
>
> > Requested change 2: Add missing baselines: Compare against closely related methods, including BATCLIP, CLIPArTT, WATT, and Mint. The introduction should also be revised, as it currently claims that existing methods do not evaluate on corruptions.
>
> Thank you for pointing it out.
> We added discussion on these methods to the related work section (**Sec. 6**) in the revised paper.
> In addition, we added Mint and BATCLIP to the baselines in the experiments (**Tabs. 3, 4, and 10-13**).
> Although these methods outperformed UnInfo on average (**Tabs. 3 and 4**), they perform worse than No-adapt on some corruption types.
> Moreover, on ImageNet-C-bar and different CLIP architectures, their average accuracy is sometimes worse than No-adapt (**Tabs. 12 and 13**).
> On the other hand, UnInfo is more stable across corruption types and model architectures, highlighting its strong generality for image corruption.
>
>
>
> > Missing controlled ablation on encoder and embedding adaptation.
> > The paper argues that adapting embeddings is insufficient and the image encoder must be updated. A controlled ablation is needed to verify this using the same loss and optimization settings while adapting different components.
>
> > Requested change 3: Add a controlled ablation: Compare encoder-level and embedding-level adaptation using the same loss, optimizer, and adaptation protocol.
>
> Thank you for your suggestion.
> We conducted an additional experiment of the suggested UnInfo variant, which freezes the image encoder and updates image embeddings as optimized parameters.
> We added the results to **Sec. E.2** (**Tab. 21**) in the appendix.
> UnInfo (embedding) had only a slight effect on accuracy compared to No-adapt, suggesting that updating the image encoder is efficient.
> This is because UnInfo can accumulate knowledge of the target distribution by updating the image encoder, whereas updating image embeddings is episodic and cannot leverage knowledge from past inputs.

---

> ### Author Response · Authors · 2026-07-15
>
> > The entropy-only result requires verification.
> > Entropy minimization achieves exactly 0.10% accuracy on every corruption, this is equivalent to chance-level accuracy. The authors should verify whether this is caused by model collapse, the learning rate, temperature, adaptation protocol, or an implementation error.
>
> > Requested change 4: Verify entropy-only collapse: Explain the consistent 0.10% accuracy and rule out implementation or hyperparameter issues.
>
>
> Thank you for pointing it out.
> First of all, we found a mistake in our accuracy aggregation script and modified the accuracies of $\mathcal{L}\_\text{ent}$ in **Tabs. 5 and 6**.
> We apologize for the mistake.
> However, it does not affect the conclusion; $\mathcal{L}\_\text{ent}$ had only $<1\\%$ accuracy on all corruption types.
> The significant performance drop in the entropy-only loss is due to model collapse.
> The entropy-only loss can be minimized by just assigning a high probability to an arbitrary class.
> When updating the image encoder, although we adopt LoRA, entropy minimization can overfit the target distribution due to the high degree of freedom.
> Moreover, adding $\mathcal{L}\_\text{pl}$ significantly improved performance, implying that this is not an implementation error.
>
> In CLIP adaptation to image corruption, model collapse becomes severe because of uniformity collapse.
> We observed that CLIP classifies a large portion of the test data into a single class under image corruption.
> **Tab. A** shows the ratio of the class (%) to which an unadapted CLIP classified the largest number of ImageNet-C samples.
>
>
> **Tab. A**
> | Defocus blur | Glass blur | Motion blur | Zoom blur | Contrast | Elastic transform | Jpeg compression | Pixelate | Gaussian noise | Impulse noise | Shot noise | Brightness | Fog | Frost | Snow |
> |---|---|---|---|---|---|---|---|---|---|---|---|---|---|---|
> | 1.16 | 6.42 | 5.77 | 5.09 | 3.88 | 5.85 | 0.84 | 0.93 | 24.45 | 23.96 | 18.89 | 0.73 | 2.86 | 5.59 | 7.01 |
>
> We can see that a significantly larger ($>1/1000=0.1\\%$) number of samples were classified to a single class, where entropy-only loss exacerbates this imbalance.
>
> We added this discussion to **Sec. E.5** in the appendix.
>
>
>
> > The efficiency comparison with TDA is unclear.
> UnInfo requires both forward and backward propagation, whereas TDA is backpropagation-free. The authors should explain why UnInfo is reported to be computationally more efficient than TDA.
>
> > Requested change 5: Clarify the TDA efficiency comparison: Report a controlled comparison and explain why a backpropagation-based method is faster than TDA.
>
>
> Thank you for your clarification.
> This is because UnInfo can leverage GPUs' parallelization by processing test samples in batches, even though backpropagation is required.
> TDA requires one-by-one data processing to update the cache, which hinders GPU acceleration.
> On the other hand, UnInfo achieved higher throughput by processing data in batches and using LoRA.
> In contrast, UnInfo's GPU memory usage was higher than TDA, MTA, and ZERO, suggesting a trade-off between throughput and memory usage.
>
> We added the explanation to **Sec. 5.4.5**.
>
>
>
> > Requested change 6: Clarify the shift definitions: Clearly distinguish domain shift from sensor degradation and justify why the evaluated corruptions are categorized as sensor degradation.
>
> Thank you for pointing it out.
> The difference between sensor degradation and other distribution shifts (e.g., domain shifts) is that sensor degradation data have their clean counterparts.
> Image corruption is a form of sensor degradation, since clean images exist before being corrupted.
> That is, sensor degradation can be viewed as a Markov chain $X \rightarrow X' \rightarrow Z$, where $X$ and $X'$ are the original image and its corrupted one, and $Z$ is the embedding.
> By the data processing inequality, $\mathcal{I}(X;X') \geq \mathcal{I}(X;Z)$, where $\mathcal{I}(\cdot; \cdot)$ is mutual information.
> By rewriting mutual information with entropy $\mathcal{H}(\cdot)$, we have $\mathcal{H}(X|X') + \mathcal{H}(Z) \leq \mathcal{H}(Z|X)$.
> When corruption becomes severe, it becomes more difficult to recover the original image $X$ from the corrupted one $X'$, i.e., $\mathcal{H}(X|X')$ increases.
> Thus, assuming that the fluctuation of $\mathcal{H}(Z|X)$ is small, $\mathcal{H}(Z)$ is likely to decrease, i.e., $Z$ becomes less diverse and distributes less uniformly.
>
>
> We clarified the difference between sensor degradation (image corruption) and other distribution shifts and added the above discussion to **Sec. 3**.

---

### Review · Reviewer_2i4L · 2026-06-16

**Summary Of Contributions:**

This paper studies TTA of CLIP-style VLMs under sensor degradation, as distinct from domain shift. Through preliminary experiments, the authors show that under corruption, CLIP's zero-shot accuracy collapses and, critically, that existing TTA methods fail to recover performance. They trace this failure to a drop in uniformity of image embeddings on the unit hypersphere. Based on this, they propose UnInfo, which fine-tunes the image encoder via LoRA at test time using their proposed losses. Experiments on ImageNet-C and ImageNet-C-bar show consistent improvements over other baselines.

**Audience:**

Yes

**Audience Explanation:**

The topic and the paper are well within TMLR's scope and would also be of interest to a broader audience.

**Broader Impact Concerns:**

The draft doesn't include Broader Impact Concerns. Even though it is not mandatory, it would be nice to have one since the paper discusses and promotes the deployment of self-adapting vision-language models.

**Claims And Evidence:**

Yes

**Claims Explanation:**

*Strengths*
1. Clear and well-motivated problem framing. The preliminary experiment in Section III, especially the contrast between domain-shift and corruption statistics in Tables 1 and 2, is a genuinely useful diagnostic contribution.
2. The ablations individually well justify the method components.
3. Comparisons include a few zero-shot TTA and few-shot adaptation baselines, which is a useful sanity check.

*Weaknesses*
1. The major weakness is the lack of relevant baselines. A large bunch of published works on TTA for CLIP models (with the same TTA setting) already exist, and none of them have been cited or benchmarked against. Please see the cited works, with some of them outperforming on the same datasets. I'm sure there are more works to compare against. In fact, a similar analysis as in Section III has been done in [1].
2. Lack of analysis on CIFAR-10C and CIFAR-100C.
3. Hyperparameters were tuned on "a few corruption types" (Section 5.2). This does not seem to be a good practice, especially in test-time settings.
4. Why is there a new term, "sensor degradation" and not "corruptions" that is generally used in the TTA literature?


[1] Sarthak Kumar Maharana, Baoming Zhang, Leonid Karlinsky, Rogerio Feris, and Yunhui Guo. Batclip: Bimodal online test-time adaptation for clip. In ICCV, 2025b.

[2] Maxime Zanella, Clément Fuchs, Christophe De Vleeschouwer, and Ismail Ben Ayed. Realistic test-time adaptation of vision-language models. arXiv preprint arXiv:2501.03729, 2025.

[3] David Osowiechi, Mehrdad Noori, Gustavo Adolfo Vargas Hakim, Moslem Yazdanpanah, Ali Bahri, Milad Cheraghalikhani, Sahar Dastani, Farzad Beizaee, Ismail Ben Ayed, and Christian Desrosiers. Watt: Weight average test-time adaption of clip. arXiv preprint arXiv:2406.13875, 2024

[4] Shambhavi Mishra, Julio Silva-Rodriguez, Ismail Ben Ayed, Marco Pedersoli, and Jose Dolz. Semantic anchor transport: Robust test-time adaptation for vision-language models, 2024.

[5] Mario Döbler, Robert A Marsden, Tobias Raichle, and Bin Yang. A lost opportunity for vision-language models: A
comparative study of online test-time adaptation for vision-language models. ECCV Workshops, 2024.

**Requested Changes:**

1. Add the missing baselines and compare against them under the same TTA settings.
2. Add CIFAR-10-C and CIFAR-100-C experiments.
3. Revise the hyperparameter selection procedure and reporting. Please:
  - Report which specific corruption types were used for selection,
  - Add an experiment showing sensitivity when hyperparameters are selected on one corruption family (e.g., noise) and evaluated on a disjoint family (e.g., weather/blur), to demonstrate that the method is not simply overfit to the selection corruptions.
4. Given the expanded baseline set, please also re-run or extend the computational-efficiency comparison.
5. It would also be nice to see continual TTA results, if possible.

---

> ### Author Response · Authors · 2026-07-15
>
> Dear reviewer `2i4L`,
>
> We appreciate your thoughtful comments and suggestions.
>
> > The major weakness is the lack of relevant baselines. A large bunch of published works on TTA for CLIP models (with the same TTA setting) already exist, and none of them have been cited or benchmarked against. Please see the cited works, with some of them outperforming on the same datasets. I'm sure there are more works to compare against. In fact, a similar analysis as in Section III has been done in [1]. (BATCLIP)
>
> > Requested change 1: Add the missing baselines and compare against them under the same TTA settings.
>
> > Requested change 4: Given the expanded baseline set, please also re-run or extend the computational-efficiency comparison.
>
>
> Thank you for pointing it out.
> We added a discussion on existing TTA methods targeting image corruption to **Sec. 6**.
> Also, we added experiments on Mint [Bao et al., 2025] and BATCLIP [Maharana et al., 2025] (**Tabs. 3, 4, and 10-13**).
> Although these methods outperformed UnInfo on average (**Tabs. 3 and 4**), they perform worse than No-adapt on some corruption types.
> Moreover, on ImageNet-C-bar and different CLIP architectures, their average accuracy was sometimes worse than No-adapt (**Tabs. 12 and 13**).
> This is likely because the baselines are hyperparameter-sensitive and prone to overfitting to ImageNet-C corruption.
> On the other hand, UnInfo is more stable across corruption types and model architectures, with a single hyperparameter setting and only a few ImageNet-C corruption types, suggesting its generality for image corruption.
>
> As for computation, we added the two methods to **Tab. 7** (**Sec. 5.4.5**).
> Although BATCLIP and Mint had higher throughput than UnInfo because they update layer normalization parameters, whose number is smaller than LoRA, UnInfo's throughput is also reasonable.
>
>
> > Lack of analysis on CIFAR-10C and CIFAR-100C.
>
> > Requested change 2: Add CIFAR-10-C and CIFAR-100-C experiments.
>
>
> Thank you for your suggestion.
> We added experiments on CIFAR10C and CIFAR100C to **Sec. E.1** (**Tabs. 14-19**) in the appendix.
> UnInfo also consistently improved accuracy also on these datasets compared to the embedding-based baselines (TPT, ZERO, MTA, and TDA), although BATCLIP and Mint achieved higher accuracy.
> But UnInfo is more stable, as it consistently improved accuracy on more realistic datasets (ImageNet-C and ImageNet-C-bar) across model architectures.
>
>
>
> > Hyperparameters were tuned on "a few corruption types" (Section 5.2). This does not seem to be a good practice, especially in test-time settings.
>
> > Requested change 3: Revise the hyperparameter selection procedure and reporting. Please:
> > Report which specific corruption types were used for selection,
> > Add an experiment showing sensitivity when hyperparameters are selected on one corruption family (e.g., noise) and evaluated on a disjoint family (e.g., weather/blur), to demonstrate that the method is not simply overfit to the selection corruptions.
>
>
> Thank you for your clarification.
> We reported which types of corruption were used in **Sec. 5.2**.
>
> Also, we tried a selection strategy based on only defocus blur.
> As described in **Sec. 5.2**, the mutual information threshold $\mathcal{I}_0$ is selected based on clean ImageNet's mutual information.
> Thus, we selected the uniformity loss weight $\lambda$ based on defocus blur from $\\{ 0.0, 0.25, \ldots, 2.0 \\}$.
> The best hyperparameters are $1.5$ and $2.0$.
> From the sensitivity analysis in **Fig. 2**, these values do not have a large effect on the average accuracy on the other corruption types, suggesting the reasonability of $\lambda$ used in the experiments.
>
> Moreover, the hyperparameter setting we used in the experiments (selected based on ImageNet-C corruption) also works on ImageNet-C-bar.
> The corruption types on ImageNet-C-bar are selected to be dissimilar to ImageNet-C [Mintun et al., 2021], suggesting that our hyperparameter setting does not overfit to the selected corruption types.
>
> We added this experiment to **Sec. E.3** in the appendix.
>
>
> > Why is there a new term, "sensor degradation" and not "corruptions" that is generally used in the TTA literature?
>
> Thank you for pointing it out.
> We rephrased the term "sensor degradation" as "image corruption", which is more familiar in the TTA literature.

---

> > ### Author Response · Authors · 2026-07-15
> >
> > > Requested change 5: It would also be nice to see continual TTA results, if possible.
> >
> > Thank you for your suggestion.
> > We tested UnInfo under the continual setting, where TTA is applied sequentially across all corruption types, in **Sec. E.4** in the appendix.
> > Here, we used the same hyperparameter settings except for the learning rate, being set to $10^{-4}$ to avoid catastrophic forgetting.
> > The result is shown in **Tab. 22**.
> > Although UnInfo improved accuracy compared to No-adapt on most corruption types, the accuracy gains for the latter half of corruptions are smaller than those for the former half.
> > This is because the EMA teacher retains the LoRA parameters adapted to previous corruptions, which affect current adaptation.
> > One possible modification would be to adaptively reset LoRA using change detection, which is one direction for future work.
> >
> >
> > > The draft doesn't include Broader Impact Concerns. Even though it is not mandatory, it would be nice to have one since the paper discusses and promotes the deployment of self-adapting vision-language models.
> >
> > Thank you for pointing it out.
> > We added the broader impact section to **Sec. A** of the appendix.

---

### Review · Reviewer_Hnxw · 2026-07-03

**Summary Of Contributions:**

This paper studies test-time adaptation (TTA) of CLIP-family zero-shot classifiers under sensor degradation, modeled with synthetic image corruptions, and argues that this regime differs qualitatively from the domain shifts (ImageNet-A/R style) targeted by existing CLIP TTA methods. The contributions are threefold. First, a diagnostic analysis: under ImageNet-C corruptions at the highest severity, prediction entropy and the uniformity loss of image embeddings rise sharply (uniformity 0.513 on clean data versus a corruption mean of 0.715; entropy 0.748 versus 2.395) while the EMD-based modality gap remains essentially flat (1.29 versus 1.31); prompt ensembling gains about one point, corruption-name prompts reduce accuracy, and CLIP classifies the corruption type itself at only 23.5% on average. The authors interpret this as corruption reducing the information retained in image embeddings rather than shifting image-text alignment. Second, the proposed method, UnInfo: an online, cumulative TTA procedure that updates the image encoder through LoRA adapters on the attention projections, combining (i) batch entropy minimization with uniformity maximization, (ii) a dynamic weight $w = exp(I(z; \hat{y}) − I_0)$ that trades the two losses based on batch mutual information, and (iii) soft distillation toward an EMA-of-LoRA teacher (m = 0.001), which also serves as the inference model. Third, experiments on ImageNet-C (15 types) and ImageNet-C-bar (10 types) at maximum severity with three backbones, against episodic methods (TPT, ZERO, MTA), one cumulative cache method (TDA), and few-shot baselines, together with ablations, sensitivity analyses, qualitative weight dynamics, and a throughput comparison (73.7 images/s for UnInfo versus 1.8 for TPT).

**Additional Comments:**

1. Since the LoRA B matrices are zero-initialized, $L_{pl}$ initially anchors the student to the source zero-shot predictions, so the teacher acts as a soft source anchor early in adaptation. This effect deserves explicit acknowledgment, given that $L_{ent} + L_{pl}$ already recovers 25.54 of the final 27.10 (Table 5) and may account for much of the stabilization attributed to the framework.

2. How reliable is the estimate of $I(z; \hat y)$ with B = 64 and C = 1000, where the marginal entropy $H(\bar y)$ is substantially biased when the batch size is far smaller than the number of classes? A short discussion or an empirical check of this estimator would be helpful.

3. Does the "Domain shift Mean" column in Table 1 intentionally include clean ImageNet? If so, please clarify the header or the caption.

**Audience:**

Yes

**Audience Explanation:**

TTA of vision-language models is an active area, and the paper offers value beyond the method itself: (a) an empirical demonstration that episodic augmentation-based CLIP TTA methods degrade or fail under corruption, including the notable result that TPT, ZERO, and MTA fall far below No-adapt on SigLIP, which is useful information for practitioners regardless of UnInfo; (b) the uniformity-versus-modality-gap diagnostic, which gives the community a measurable indicator of when frozen-feature methods are unlikely to help; and (c) a fast, competitive encoder-updating baseline for a benchmark combination (CLIP with ImageNet-C/C-bar TTA) that is currently under-explored.

**Broader Impact Concerns:**

No statement is included, and I do not consider one strictly required. Given the motivating applications (autonomous driving, surveillance), a short paragraph would be appropriate on: (a) the risk of silent failure, since continual unsupervised adaptation can degrade without any labeled signal to detect it, as the paper's own ablation illustrates; (b) the susceptibility of online TTA to adversarial or manipulated test streams; and (c) considerations around the surveillance use case itself.

**Claims And Evidence:**

No

**Claims Explanation:**

What I find convincing: the gains are consistent across 25 corruption types and three backbones; the separation of corruption from domain shift via uniformity and entropy is a genuinely useful observation; each ablation component adds accuracy; the balancing mechanism helps most exactly where the paper's reasoning predicts (noise corruptions: Gaussian 3.90 to 7.56, impulse 7.87 to 10.60, shot 7.17 to 11.36); the weight-evolution plots (Fig. 4) match the intended mechanism; and the SigLIP experiments are an informative stress test in which the augmentation-based episodic baselines fall far below the no-adaptation baseline (TPT at 8.48 versus 25.46).

Though my main concerns are:

1. UnInfo is the only cumulative and parameter-updating method in the comparison, while the cumulative frozen-encoder family is represented solely by TDA. Its strongest members are absent: DMN (Zhang et al., CVPR 2024) is cited in Section 6.2 but never benchmarked, and DPE (Zhang et al., NeurIPS 2024) is neither cited nor benchmarked. DPE evolves textual and visual prototypes online, using a cumulative average and a priority queue respectively, with per-sample learnable residuals optimized under alignment and self-entropy losses, and it reports outperforming TDA as well as TPS (Sui et al., WACV 2025) and DMN-ZS on standard benchmarks. These methods accumulate task-specific knowledge across the stream exactly as UnInfo does while keeping the encoders frozen, so they directly test the paper's central claim that frozen-feature methods cannot recover information lost to corruption. If UnInfo beats them on the high-uniformity-loss corruptions, the encoder-updating contribution is cleanly isolated; if they match UnInfo, the thesis weakens. Given that UnInfo's mean margin over TDA is 1.56 points on ImageNet-C and 0.71 on C-bar, a stronger cumulative frozen-encoder baseline could plausibly close much of that gap, so the question should be settled empirically.

2. Within the cumulative parameter-updating cell itself, no alternative objectives are evaluated under the same LoRA, optimizer, and stream protocol. TENT-style entropy on LayerNorm affines, a SHOT-style information-maximization objective, $L_{pl}$ alone, and $L_{unif} + L_{pl}$ are all natural comparisons, and WATT and CLIPArTT (encoder-updating CLIP TTA methods evaluated on corruption benchmarks such as CIFAR-10/100-C) already exist in this cell and are uncited.  Without these, the benefit of the specific proposed losses cannot be separated from the benefit of continually updating the encoder at all.

3. $I(z; \hat y) = H(\bar y) - H(\hat y | z) $ is computed in Eq. (9) but enters the method only as an exponential weight. Directly maximizing this quantity as a loss, that is, adding a prediction-diversity term as in the information-maximization literature the paper cites for the estimator, is the standard remedy for the degenerate solution observed in the ablation, and it is never compared. Demonstrating that the $exp(I − I_0)$ weighting outperforms this simpler objective is important for establishing the balancing scheme as a contribution rather than a reformulation.

4. A uniform 0.10 across all 25 corruption types and both datasets corresponds exactly to predicting a single class on a class-balanced 50,000-image set (50/50,000 = 0.1%), which is the global minimum of the entropy objective. This outcome reflects the interaction of the loss with the specific optimization configuration (expressive LoRA parameters, AdamW at lr 1e-3, continual updates without reset, severity-5 inputs) as much as the loss itself: Fig. 3 shows a fixed entropy weight of 20 also drives accuracy toward zero even with $L_{unif}$ and $L_{pl}$ active, while TPT, itself an entropy-minimization method, reaches 25.07 in Table 3. Prediction histograms or mutual-information trajectories confirming the collapse mode, a tuned or parameter-constrained entropy-only variant, and a connection to prior analyses of entropy-minimization instability (e.g., SAR, EATA-style filtering) would make this ablation considerably more informative than the current presentation, which reads as evidence that entropy is inherently unusable in the presence of corruption.

5. All evidence for the sensor-degradation claim comes from ImageNet-C and C-bar at the highest severity, i.e., synthetic digital perturbations. A real captured benchmark exists: ImageNet-ES (CVPR 2024) contains 202k photographs taken with a physical camera in a controlled testbed under varying light and camera-sensor settings, built precisely to complement digital-perturbation benchmarks. Evaluating on such data, or at lower severities, would substantially strengthen the claim; otherwise the framing should be softened toward "common corruptions". In addition, the whole experimental section is based on these two datasets and it is somewhat limited to draw safe conclusions about the proposed method.

6.  Section 3 states a LAION-pretrained ViT-B/16 while Section 5.2 states DataComp-1B. We need clarification on why the authors used a different pre-trained model for the preliminary experiment than for the main ones.

7. Three random seeds are used, and only means appear in Tables 3 to 6, without standard deviations or significance tests. Several per-type margins over TDA on C-bar are below one point, and a few are negative (Sparkles 53.74 versus 55.00; concentric sine waves 12.49 versus 12.69), so uncertainty estimates matter for the conclusions drawn.

**Requested Changes:**

Changes I consider critical to support the claims:

1. Add the strongest cumulative frozen-encoder methods as baselines: DMN (cited but not compared) and DPE (currently uncited). Report matched stream orderings, seeds, operating batch sizes (DMN and DPE run at batch size 1 while UnInfo requires 64), and computational cost, including DPE's per-sample augmentation cost in Table 7. These comparisons isolate whether encoder updating, rather than accumulation alone, produces the gains.

2. Add cumulative parameter-updating alternatives under the identical LoRA, optimizer, and stream protocol: TENT (LayerNorm-affine entropy), a SHOT-style information-maximization objective (entropy minus marginal entropy), $L_{pl}$ alone, and $L_{unif} + L_{pl}$. Report whether the $exp(I − I_0)$ weighting outperforms directly maximizing mutual information, and discuss WATT and CLIPArTT as existing members of this family.

3. Strengthen the entropy-only ablation with prediction histograms or mutual-information trajectories confirming the collapse mode, add a tuned or parameter-constrained entropy-only variant, and relate the finding to prior work on entropy-minimization instability rather than presenting it as evidence that entropy is inherently unusable under corruption.

4. Evaluate on at least one non-synthetic sensor-shift dataset (for example, ImageNet-ES), or adjust the sensor-degradation framing accordingly. The experimental section is limited to two datasets only.

5. Specify the evaluation protocol precisely: whether accuracy is computed online (predicting each batch before updating) or after a full adaptation pass over the same data, since Algorithm 1 leaves this ambiguous. Add accuracy-versus-iteration curves and a batch-size ablation (for example, B in {1, 8, 16, 32, 64}), given that $L_{unif}$ and $I(z; \hat y)$ are batch statistics and the fully online case is acknowledged as a limitation.

6. Resolve the LAION versus DataComp checkpoint discrepancy, and add standard deviations to the main tables.

7. Reorganize Section 6.2 along two axes, grouping TDA, DMN, DPE, and OnZeta as cumulative frozen-encoder methods rather than listing them with episodic approaches, and add the missing related work: DPE (Ce Zhang et al. NeurIPS 2024); TPS (WACV 2025), which learns per-class prototype shift vectors from unlabeled test inputs; STS (Konstantinos M. Dafnis et al. NeurIPS 2025) which extracts a spectral subspace from the textual embeddings to define principal semantic directions and learns to steer latent representations in a spectrum-aware manner by adapting a small number of per-sample shift parameters to minimize entropy across augmented views; "Feature Alignment and Uniformity for Test Time Adaptation" (Shuai Wang et al. CVPR 2023), which already uses the alignment and uniformity measures for TTA with experiments on corruption benchmarks and therefore affects the novelty framing; SHOT (Jian Liang et al. ICML 2020) as the canonical precedent for the mutual-information quantity in Eq. (9); SAR (Shuaicheng Niu et al. ICLR 2023) on entropy-minimization instability; WATT (David Osowiechi et al. NeurIPS 2024) and CLIPArTT (Gustavo Adolfo Vargas Hakim et al WACV 2025) as encoder-updating CLIP TTA methods evaluated on corruptions; ImageNet-ES (CVPR 2024); and please also correct the attribution of Eq. (6), which is the uniformity loss of Wang and Isola (2020) with t = 1 rather than Oord et al. (2018).

Changes that would strengthen the work but are not blocking:

8. Experiments with mixed or continually changing corruption streams, which are known challenges for batch-statistic TTA.

9. An analysis of why episodic augmentation-based methods perform so poorly on SigLIP

---

> ### Author Response · Authors · 2026-07-17
>
> Dear reviewer `Hnxw`,
>
> We appreciate your careful reading and insightful feedback.
>
>
> > stronger cumulative frozen-encoder baseline could plausibly close much of that gap, so the question should be settled empirically.
>
> > Requested change 1: Add the strongest cumulative frozen-encoder methods as baselines: DMN (cited but not compared) and DPE (currently uncited). Report matched stream orderings, seeds, operating batch sizes (DMN and DPE run at batch size 1 while UnInfo requires 64), and computational cost, including DPE's per-sample augmentation cost in Table 7.
>
> Thank you for your suggestion.
> We added DMN as a cumulative frozen-encoder baseline to **Tabs. 3-6 and 10-13**.
> As we expected, the improvement by DMN was small, or sometimes worse than No-adapt, suggesting that updating the encoder is crucial for handling image corruption.
>
> Also, we added DMN to the computational cost table (**Tab. 7**).
> DMN had lower throughput because it runs with a batch size of 1 and uses augmented views.
>
>
> > WATT and CLIPArTT (encoder-updating CLIP TTA methods evaluated on corruption benchmarks such as CIFAR-10/100-C) already exist in this cell and are uncited. Without these, the benefit of the specific proposed losses cannot be separated from the benefit of continually updating the encoder at all.
>
> > Requested change 2: Add cumulative parameter-updating alternatives under the identical LoRA, optimizer, and stream protocol: TENT (LayerNorm-affine entropy), a SHOT-style information-maximization objective (entropy minus marginal entropy), $L_{pl}$ alone, and $L_{pl}+L_{unif}$. Report whether the $exp(I-I_0)$ weighting outperforms directly maximizing mutual information, and discuss WATT and CLIPArTT as existing members of this family.
>
> > $I(z; \hat y) = H(\bar y) - H(\hat y | z)$ is computed in Eq. (9) but enters the method only as an exponential weight. Directly maximizing this quantity as a loss, that is, adding a prediction-diversity term as in the information-maximization literature the paper cites for the estimator, is the standard remedy for the degenerate solution observed in the ablation, and it is never compared.
>
> > weighting outperforms this simpler objective is important for establishing the balancing scheme as a contribution rather than a reformulation.
>
>
> Thank you for pointing it out.
> We added Mint [Bao et al., 2025] and BATCLIP [Maharana et al., 2025] to the baselines as cumulative encoder-updating methods in **Tabs. 3-6 and 10-13**.
> Although these methods outperformed UnInfo on average (**Tabs. 3 and 4**), they perform worse than No-adapt on some corruption types.
> Moreover, on ImageNet-C-bar and different CLIP architectures, their average accuracy is sometimes worse than No-adapt (**Tabs. 12 and 13**).
> On the other hand, UnInfo is more stable across corruption types and model architectures, highlighting its strong generality for image corruption.
>
> Also, to separate the benefit of our loss from that of continually updating the encoder, we added a controlled experiment of an UnInfo variant that freezes the image encoder and directly updates image embeddings.
> The result is shown in **Tab. 21 (Sec. E.2)** in the appendix.
> UnInfo (embedding) had only a slight effect on accuracy compared to No-adapt, suggesting that updating the image encoder is efficient.
>
> To verify the efficacy of the loss function itself, we added an ablation to **Tabs. 5 and 6 (Sec. 5.4.2)**.
> We compared the mutual information-only loss $\mathcal{L}\_\text{mi}$ and $\mathcal{L}\_\text{mi}+\mathcal{L}\_\text{pl}$.
> The mutual information-only loss resulted in model collapse, similar to the entropy-only loss.
> This is because it is optimal for $\mathcal{L}_\text{mi}$ to make each image embedding in the batch identical to a text embedding, regardless of the ground truth, making predictions diverse and confident, but suboptimal for image embedding diversity.
> Although combining the EMA teacher $\mathcal{L}\_\text{mi}+\mathcal{L}\_\text{pl}$ significantly improved the accuracy and sometimes outperformed UnInfo, enhancing uniformity and adaptive balancing had better results on average.

---

> > ### Author Response · Authors · 2026-07-17
> >
> > > Requested change 3: Strengthen the entropy-only ablation with prediction histograms or mutual-information trajectories confirming the collapse mode, add a tuned or parameter-constrained entropy-only variant, and relate the finding to prior work on entropy-minimization instability rather than presenting it as evidence that entropy is inherently unusable under corruption.
> >
> > > while TPT, itself an entropy-minimization method, reaches 25.07 in Table 3
> >
> > To analyze model collapse caused by the entropy-only loss, we investigated the statistics of the model's predictions.
> > **Tab. A** shows the ratio of the class (%) to which an unadapted CLIP classified the largest number of ImageNet-C samples.
> >
> > **Tab. A**
> > | Defocus blur | Glass blur | Motion blur | Zoom blur | Contrast | Elastic transform | Jpeg compression | Pixelate | Gaussian noise | Impulse noise | Shot noise | Brightness | Fog | Frost | Snow |
> > |---|---|---|---|---|---|---|---|---|---|---|---|---|---|---|
> > | 1.16 | 6.42 | 5.77 | 5.09 | 3.88 | 5.85 | 0.84 | 0.93 | 24.45 | 23.96 | 18.89 | 0.73 | 2.86 | 5.59 | 7.01 |
> >
> > We can see that a significantly larger ($>1/1000=0.1\\%$) number of samples were classified to a single class, where entropy-only loss exacerbates this imbalance.
> > The instability of simple entropy minimization is also addressed in classification TTA studies, such as SAR [Niu et al., 2023] and ReCAP [Hu et al., 2025].
> > Combining these techniques with our method would be future work.
> >
> > We added this discussion to **Sec. E.5** in the appendix.
> >
> > On the other hand, TPT did not collapse because it is an episodic method; the prompt embeddings are reset for every prediction.
> >
> >
> > > Requested change 4: Evaluate on at least one non-synthetic sensor-shift dataset (for example, ImageNet-ES), or adjust the sensor-degradation framing accordingly. The experimental section is limited to two datasets only.
> >
> > Thank you for pointing it out.
> > We rephrased the term "sensor degradation" as "image corruption."
> > We would like to evaluate on non-synthetic sensor-shift datasets, such as ImageNet-ES, in future work.
> > We also added a discussion in **Sec. 7** regarding the challenges of real-world sensor-shift data.
> >
> > > Section 3 states a LAION-pretrained ViT-B/16 while Section 5.2 states DataComp-1B. We need clarification on why the authors used a different pre-trained model for the preliminary experiment than for the main ones.
> >
> > > Requested change 6: Resolve the LAION versus DataComp checkpoint discrepancy, and add standard deviations to the main tables.
> >
> > Thank you for pointing it out.
> > The statement of DataComp in **Sec. 5.2** is our mistake.
> > Actually, we used the same OpenCLIP ViT-B/16 pre-trained on the LAION dataset in both the preliminary and main experiments.
> > We fixed the description in **Sec. 5.2**.
> > We apologize for the confusion.
> >
> >
> > > Three random seeds are used, and only means appear in Tables 3 to 6, without standard deviations or significance tests. Several per-type margins over TDA on C-bar are below one point, and a few are negative (Sparkles 53.74 versus 55.00; concentric sine waves 12.49 versus 12.69), so uncertainty estimates matter for the conclusions drawn.
> >
> > We added the standard deviations to the main results (**Tabs. 3-6 and 10-13**).
> >
> >
> >
> > > Requested change 5: Specify the evaluation protocol precisely: whether accuracy is computed online (predicting each batch before updating) or after a full adaptation pass over the same data, since Algorithm 1 leaves this ambiguous. Add accuracy-versus-iteration curves and a batch-size ablation (for example, B in {1, 8, 16, 32, 64})
> >
> > Thank you for pointing it out.
> > Our evaluation protocol is batched online (predicting each batch before updating).
> > We added the description to **Section 5.4.1**.
> >
> > We also added a batch size ablation to **Fig. 6 (Sec. E.2)**.
> > When the batch size is $\leq 32$, the accuracy dropped significantly.
> > This is because the uniformity estimation becomes poor with small batch sizes, which is one limitation of UnInfo discussed in **Sec. 7**.
> > One possible modification is to accumulate batch statistics, such as a moving average of uniformity, to compensate for the estimation when using a single small batch size.

---

> > > ### Author Response · Authors · 2026-07-17
> > >
> > > > Requested change 7: Reorganize Section 6.2 along two axes
> > >
> > > Thank you for your suggestion.
> > > We updated the related work section (**Sec. 6.2**) from the perspectives of episodic/cumulative and frozen/updating encoders, and added suggested papers.
> > > Also, we added citations to the SHOT and uniformity papers accordingly.
> > >
> > >
> > > > Experiments with mixed or continually changing corruption streams, which are known challenges for batch-statistic TTA.
> > >
> > > Thank you for your suggestion.
> > > We tested UnInfo under the continual setting, where TTA is applied sequentially across all corruption types, in **Sec. E.4** in the appendix.
> > > Here, we used the same hyperparameter settings except for the learning rate, being set to $10^{-4}$ to avoid catastrophic forgetting.
> > > The result is shown in **Tab. 22**.
> > > Although UnInfo improved accuracy compared to No-adapt on most corruption types, the accuracy gains for the latter half of corruptions are smaller than those for the former half.
> > > This is because the EMA teacher retains the LoRA parameters adapted to previous corruptions, which affect the current adaptation.
> > > One possible modification would be to adaptively reset LoRA using change detection, which is one direction for future work.
> > >
> > >
> > > > Since the LoRA B matrices are zero-initialized, $L_{pl}$ initially anchors the student to the source zero-shot predictions, so the teacher acts as a soft source anchor early in adaptation. This effect deserves explicit acknowledgment, given that  $L_{ent} + L_{pl}$ already recovers 25.54 of the final 27.10 (Table 5) and may account for much of the stabilization attributed to the framework.
> > >
> > > Thank you for pointing it out.
> > > Indeed, UnInfo owes its improvement to the $\mathcal{L}_\text{pl}$'s stabilization effect.
> > > However, the uniformity loss and balancing also play critical roles in UnInfo, especially when embedding collapse is severe, such as under noise corruption.
> > > In such cases, the teacher anchor is no longer reliable in the initial phase, where the uniformity loss recovers the collapse.
> > > For instance, in **Fig. 4 (c) and (d)**, the weight of the uniformity loss is larger than that of the entropy loss in the initial phase of adaptation.
> > > This suggests that the classification with the current CLIP (similar to the EMA teacher in the initial phase) does not work, e.g., classifying all samples to a single class.
> > > UnInfo shows significant improvements in such cases compared to $L\_\text{ent} + L\_\text{pl}$.
> > >
> > >
> > > > How reliable is the estimate of $I(z; \hat y)$ with B = 64 and C = 1000, where the marginal entropy $H(\bar{y})$ is substantially biased when the batch size is far smaller than the number of classes? A short discussion or an empirical check of this estimator would be helpful.
> > >
> > > Thank you for your clarification.
> > > As you pointed out, while it is difficult to precisely estimate mutual information with the given batch size and number of classes, our purpose is to verify whether the current classification goes well.
> > > Although $\mathcal{I}(\mathcal{z};\hat{y})$ does not necessarily align with the true mutual information, it works for this purpose in that it takes small values when prediction is collapsed (e.g., classifying all samples to a single class).
> > > **Fig. 4** also verifies our information-based weighting because the weight transitions differ depending on the corruption types.
> > >
> > > One limitation is that our weighting requires an i.i.d. assumption, i.e., the class of each sample is independent of the classes of other samples.
> > > We added the discussion and limitation to **Sec. 4.2** and **Sec. 7**.
> > >
> > >
> > >
> > >
> > > > Does the "Domain shift Mean" column in Table 1 intentionally include clean ImageNet? If so, please clarify the header or the caption.
> > >
> > > Thank you for pointing it out.
> > > We included the clean ImageNet in the mean.
> > > We added clarification to **Tab. 1**.

---

### Review · Reviewer_WrQk · 2026-07-08

**Summary Of Contributions:**

The paper studies test-time adaptation of CLIP for zero-shot classification on corrupted images. The authors show that CLIP is vulnerable to sensor degradation and that simple prompting strategies (e.g., ensembling text prompts or adding the corruption in the text prompt are not sufficient to recover performance).
The proposed method is composed of:
(i) an information-aware loss balancing to weight two different losses used in unsupervised adaptation, the entropy loss encouraging confident predictions and a uniformity loss encouraging diverse visual embeddings.
(ii) Low-Rank Adaptation: the model is adapted through low-rank (LoRA) updates on the visual encoder only.
(iii) Knowledge Distillation from EMA Teacher: with a cross-entropy between the prediction of the main model and a teacher model obtained by moving average.
The authors benchmark the proposed method on corrupted versions of the ImageNet dataset: ImageNet-C and ImageNet-C-bar and show consistent improvement over the selected baselines.

**Additional Comments:**

The experimental setting is not clearly stated, the continual adaptation across batches should be made explicit in the Setup paragraph.

**Audience:**

Yes

**Audience Explanation:**

Yes. The paper addresses a practical angle of CLIP test-time adaptation, i.e., robustness to sensor degradation. The central idea of automatically reweighting the adaptation objectives is interesting and, to my knowledge, quite novel in this setting; it could inspire further work on "adaptive" objectives for unsupervised adaptation.

**Claims And Evidence:**

No

**Claims Explanation:**

The claim that "the TTA of VLMs against distribution shifts outside domain shifts has not been explored or evaluated in existing works," together with the claim that "existing CLIP TTA methods for domain shifts address the modality gap by updating embedding
vectors in a post-hoc manner without updating encoders", might be overstated. See [1], which reports results on CIFAR10-C, CIFAR100-C and ImageNet-C while adapting LayerNorm of the visual encoder; [2], which reports results on ImageNet-C while adapting LayerNorm parameters of both CLIP encoders; and [3], which studies CLIP adaptation on corrupted versions of CIFAR while also updating LayerNorm parameters of the visual encoder.

The proposed method is the only one in the comparative results to directly adapt weights through LoRA. While showing that adapting weights on noisy samples can provide stable results is of interesting value, applying other common objectives with LoRA could strengthen the message by better isolating the effect of the proposed automatically-weighted objective UnInfo. Relatedly, why is entropy alone at 0.1 accuracy in Tables 5 and 6? Is it because LoRA is not stable enough compared to updating only LayerNorm (see the results of [2] obtained with LayerNorm, i.e., TENT)? Another way around would be to study UnInfo reweighting scheme with LayerNorm parameter update.

The improvement from the balancing component of UnInfo helps most where intended (i.e., "hard" corruptions) but overall gain seems quite meager. It is concentrated on the corruptions with very low accuracy (Gaussian noise and shot noise in Table 5). These accuracies are so low that they remain of limited practical value.

Results on clean ImageNet are missing in the main paper (only available in Table 14 of the Appendix). Stable performance on non-corrupted data streams is an important consideration, since a deployed system does not know in advance whether the incoming stream is degraded; while UnInfo does not harm performance on clean stream, which is a good property (+0.85 compared to no-adapt), it is important to note that it is still lower than other test-time adaptation methods like the training-free TDA (70.3 for TDA vs 67.8 for UnInfo).
Also studying at least one lower severity level could strengthen the results to show whether UnInfo is effective only under strong degradation or not.

If the model is not reset, how does the accuracy evolve? It could be of particular interest if accuracy of the first samples is crucial.

Why is TDA's throughput lower than UnInfo, given that UnInfo trains a LoRA (i.e., requires a backward pass) whereas TDA is training-free? Is the reported throughput measured in images/sec without counting the backward pass? And does it include the additional inference cost of the EMA teacher model? The same question applies to the GPU memory.

On a more general level concerning the overall setting: it is also unclear what happens when several severity levels and/or different noise types are mixed within the test stream (e.g., [4]). Finally, the overall setup assumes a balanced class distribution. A line of recent work has challenged this fairly "standard" assumption (e.g., [5]). Presenting and/or at least discussing these more challenging scenarios would be valuable.


[1] Bao, Wenxuan, et al. "Mint: A simple test-time adaptation of vision-language models against common corruptions." Advances in Neural Information Processing Systems 2025.

[2] Maharana, Sarthak, et al. "Batclip: Bimodal online test-time adaptation for clip." Proceedings of the IEEE/CVF International Conference on Computer Vision 2025.

[3] Osowiechi, David, et al. "Watt: Weight average test time adaptation of clip." Advances in neural information processing systems 2024.

[4] Yuan, Longhui, et al. "Robust test-time adaptation in dynamic scenarios." Proceedings of the IEEE/CVF Conference on Computer Vision and Pattern Recognition 2023.

[5] Zanella, Maxime, et al. "Realistic test-time adaptation of vision-language models." Proceedings of the IEEE/CVF Conference on Computer Vision and Pattern Recognition 2025.

**Requested Changes:**

Additional missing baselines (see above). Adding results on CIFAR10-C and CIFAR100-C could also help support the message.

Additional experiments with UnInfo+LayerNorm and/or other common test-time adaptation methods with LoRA.

Reporting clean ImageNet in the main paper, together with a discussion and at least one lower severity level, would strengthen the evaluation and make the accuracy trade-off between clean and corrupted regimes clearer.

Reporting the standard deviation over the 3 runs (rather than only the mean) would strengthen the main results and the ablation. For example, gains from the balancing component are small on several corruption and even slightly negative on Elastic transform (−0.08), making it hard to judge significance where seed variance may exceed the reported gains (e.g., Table 5: +0.07 Motion blur, +0.08 Zoom blur, +0.28 Glass blur, etc.).

---

> ### Author Response · Authors · 2026-07-17
>
> Dear reviewer `WrQk`,
>
> We appreciate your careful reading and insightful comments.
>
> > The claim that "the TTA of VLMs against distribution shifts outside domain shifts has not been explored or evaluated in existing works," together with the claim that "existing CLIP TTA methods for domain shifts address the modality gap by updating embedding vectors in a post-hoc manner without updating encoders", might be overstated. See [1], which reports results on CIFAR10-C, CIFAR100-C and ImageNet-C while adapting LayerNorm of the visual encoder; [2], which reports results on ImageNet-C while adapting LayerNorm parameters of both CLIP encoders; and [3], which studies CLIP adaptation on corrupted versions of CIFAR while also updating LayerNorm parameters of the visual encoder.
>
> Thank you for pointing it out.
> We modified our claim accordingly and updated the related work section (**Sec. 6.2**) to include recent CLIP TTA methods targeting image corruption.
>
> Also, we added experiments on Mint [Bao et al., 2025] and BATCLIP [Maharana et al., 2025] (**Tabs. 3, 4, and 10-13**).
> Although these methods outperformed UnInfo on average (**Tabs. 3 and 4**), they perform worse than No-adapt on some corruption types.
> Moreover, on ImageNet-C-bar and different CLIP architectures, their average accuracy is sometimes worse than No-adapt (**Tabs. 12 and 13**).
> On the other hand, UnInfo is more stable across corruption types and model architectures, suggesting its generality for image corruption.
>
>
> > why is entropy alone at 0.1 accuracy in Tables 5 and 6? Is it because LoRA is not stable enough compared to updating only LayerNorm (see the results of [2] obtained with LayerNorm, i.e., TENT)?
>
> Thank you for pointing it out.
> It is because LoRA's higher degree of freedom than LayerNorm.
> We investigated the statistics of the model's predictions.
> **Tab. A** shows the ratio of the class (%) to which an unadapted CLIP classified the largest number of ImageNet-C samples.
>
> **Tab. A**
> | Defocus blur | Glass blur | Motion blur | Zoom blur | Contrast | Elastic transform | Jpeg compression | Pixelate | Gaussian noise | Impulse noise | Shot noise | Brightness | Fog | Frost | Snow |
> |---|---|---|---|---|---|---|---|---|---|---|---|---|---|---|
> | 1.16 | 6.42 | 5.77 | 5.09 | 3.88 | 5.85 | 0.84 | 0.93 | 24.45 | 23.96 | 18.89 | 0.73 | 2.86 | 5.59 | 7.01 |
>
> We can see that a significantly larger ($>1/1000=0.1\\%$) number of samples were classified to a single class, where minimizing entropy-only loss with LoRA exacerbates this imbalance.
>
> We added the discussion in **Sec. E.5**.
>
>
> > Requested change 1: Additional missing baselines (see above). Adding results on CIFAR10-C and CIFAR100-C could also help support the message.
>
> Thank you for your suggestion.
> We added experiments on CIFAR10C and CIFAR100C to **Sec. E.1** (**Tabs. 14-19**) in the appendix.
> UnInfo also consistently improved accuracy on these datasets compared to the embedding-based baselines (TPT, ZERO, MTA, and TDA), although BATCLIP and Mint achieved higher accuracy.
> But UnInfo is more stable, consistently improving accuracy on more realistic datasets (ImageNet-C and ImageNet-C-bar) across model architectures.
>
>
> > Requested change 2: Additional experiments with UnInfo+LayerNorm and/or other common test-time adaptation methods with LoRA.
>
> Thank you for your suggestion.
> We compared two variants of UnInfo that update the affine parameters of LayerNorm instead of LoRA:
>
> - LayerNorm: updates the affine parameters in LayerNorm layers in the image encoder.
> - LayerNorm (w/o EMA teacher): also updates LayerNorm, but the EMA teacher regularization $\mathcal{L}_\text{pl}$ is omitted.
>
> The result is shown in **Tab. 24** (**Sec. E.6**) in the appendix.
> Although LayerNorm (w/o EMA teacher) outperformed LoRA (original UnInfo) on several corruption types, LoRA achieved the best average accuracy.
> While LayerNorm sometimes had significantly higher accuracy than LayerNorm (w/o EMA teacher) (e.g., defocus blur), it also sometimes had lower accuracy.
> When optimizing LayerNorm parameters, model collapse can be avoided without $\mathcal{L}\_\text{pl}$ because LayerNorm has fewer degrees of freedom than LoRA.
> However, UnInfo (LoRA with $\mathcal{L}\_\text{pl}$) performed the best because it hits a better balance between adaptation flexibility and regularization.

---

> > ### Author Response · Authors · 2026-07-17
> >
> > > The improvement from the balancing component of UnInfo helps most where intended (i.e., "hard" corruptions) but overall gain seems quite meager. It is concentrated on the corruptions with very low accuracy (Gaussian noise and shot noise in Table 5). These accuracies are so low that they remain of limited practical value.
> >
> > Although the overall gain from the balancing is small, it significantly improved accuracy on difficult corruption types, such as noise (e.g., $7.17\% \rightarrow 11.36 \%$ in **Tab. 5**), demonstrating that the balancing is a crucial component.
> > In addition, UnInfo consistently showed improved accuracy, whereas other methods sometimes had lower accuracy than no-adapt for some corruptions and model architectures.
> > The stability of UnInfo is an important practical value.
> >
> >
> > > Requested change 3: Reporting clean ImageNet in the main paper, together with a discussion and at least one lower severity level, would strengthen the evaluation and make the accuracy trade-off between clean and corrupted regimes clearer.
> >
> >
> > Thank you for your suggestion.
> > We added the experiment on lower severities in **Tab. 25** (**Sec. E.7**). (We will report it in the main paper in the camera-ready version, but leave it in the appendix in order not to change the numbering for discussion.)
> >
> > Although the accuracy gain is higher at higher severity levels, UnInfo showed consistent improvement regardless of corruption severity, including clean images, suggesting that UnInfo does not have a negative effect on low-severity images.
> >
> >
> > > Why is TDA's throughput lower than UnInfo, given that UnInfo trains a LoRA (i.e., requires a backward pass) whereas TDA is training-free? Is the reported throughput measured in images/sec without counting the backward pass? And does it include the additional inference cost of the EMA teacher model? The same question applies to the GPU memory.
> >
> > Thank you for your clarification.
> > We measured and reported all inference and backward pass including the EMA teacher in UnInfo.
> > The reason TDA's throughput was lower than UnInfo's is that UnInfo can leverage GPU parallelism by processing test samples in batches, even though backpropagation is required.
> > TDA requires one-by-one data processing to update the cache, which hinders GPU acceleration.
> > On the other hand, UnInfo achieved higher throughput by processing data in batches and using LoRA.
> > In contrast, UnInfo's GPU memory usage was higher than TDA, MTA, and ZERO, suggesting a trade-off between throughput and memory usage.
> >
> > We added the explanation to **Sec. 5.4.5**.
> >
> >
> >
> > > If the model is not reset, how does the accuracy evolve? It could be of particular interest if accuracy of the first samples is crucial.
> >
> > We tested UnInfo under the continual setting, where TTA is continued without reset over all corruption types sequentially, in **Sec. E.4** in the appendix.
> > Here, we used the same hyperparameter settings except for the learning rate, which was set to $10^{-4}$ to avoid catastrophic forgetting.
> > The result is shown in **Tab. 22**.
> > Although UnInfo improved accuracy compared to No-adapt on most corruption types, the accuracy gains for the latter half of corruptions are smaller than those for the former half.
> > This is because the EMA teacher retains the LoRA parameters adapted to the previous corruptions and affects the current adaptation.
> > One possible modification would be adaptively resetting LoRA using change detection, which is one direction of future work.
> >
> >
> >
> >
> > > On a more general level concerning the overall setting: it is also unclear what happens when several severity levels and/or different noise types are mixed within the test stream (e.g., [4]). Finally, the overall setup assumes a balanced class distribution. A line of recent work has challenged this fairly "standard" assumption (e.g., [5]). Presenting and/or at least discussing these more challenging scenarios would be valuable.
> >
> >
> > Thank you for pointing it out.
> > Indeed, our method assumes that the test samples are sampled i.i.d. for computing mutual information in **Eq. (9)**.
> > We would like to examine non-i.i.d. settings and mixed severity and corruption types in future work.
> > We updated **Sec. 7**.
> >
> >
> > > Requested change 4: Reporting the standard deviation over the 3 runs (rather than only the mean) would strengthen the main results and the ablation.
> >
> > Thank you for your suggestion.
> > We added the standard deviations to the main results (**Tabs. 3-6 and 10-13**).
> >
> >
> > > The experimental setting is not clearly stated, the continual adaptation across batches should be made explicit in the Setup paragraph.
> >
> > Thank you for pointing it out.
> > We updated the setting description in **Sec. 5.4.1**.